# *SuperGPQA*: Scaling LLM Evaluation across 285 Graduate Disciplines

**M-A-P & ByteDance Seed & 2077AI**
*Full author list in Contributions*

## Abstract

Large language models (LLMs) have demonstrated remarkable proficiency in mainstream academic disciplines such as mathematics, physics, and computer science. However, human knowledge encompasses over 200 specialized disciplines, far exceeding the scope of existing benchmarks. The capabilities of LLMs in many of these specialized fields—particularly in light industry, agriculture, and service-oriented disciplines—remain inadequately evaluated. To address this gap, we present ***SuperGPQA*** [1], a comprehensive benchmark that evaluates graduate-level knowledge and reasoning capabilities across 285 disciplines. Our benchmark employs a novel Human-LLM collaborative filtering mechanism to eliminate trivial or ambiguous questions through iterative refinement based on both LLM responses and expert feedback. Our experimental results reveal significant room for improvement in the performance of current state-of-the-art LLMs across diverse knowledge domains ( *e.g.*, the reasoning-focused model Gemini-2.5-Pro achieved the highest accuracy of 63.56% on *SuperGPQA*), highlighting the considerable gap between current model capabilities and artificial general intelligence. Additionally, we present comprehensive insights from our management of a large-scale annotation process, involving over 80 expert annotators and an interactive Human-LLM collaborative system, offering valuable methodological guidance for future research initiatives of comparable scope.

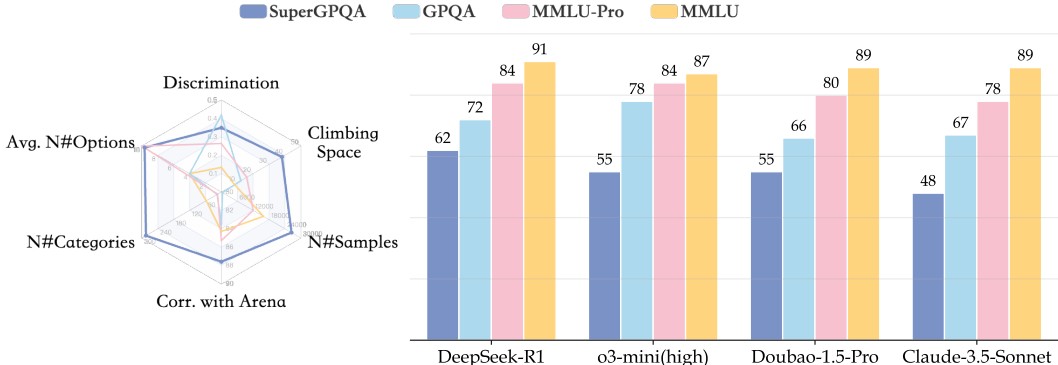

Figure 1: **Benchmark Comparison. Left**: Radar Chart. **Discrimination:** The degree of distinction between different models (detailed in Sec. D.2). **Climbing Space:** The remaining improvement space for the SOTA models. **Corr. with Arena:** Correlation with Chatbot Arena Elo scores. **Right**: Performance comparison of SOTA models across different benchmarks.

---

[1]https://supergpqa.github.io

39th Conference on Neural Information Processing Systems (NeurIPS 2025) Track on Datasets and Benchmarks.

# 1 Introduction

Large language models (**LLMs**) have greatly changed human life. LLMs are seen as the next technological singularity and proved to surpass human performance in many areas [Phan et al., 2025], significantly improving work efficiency. However, the measurement of LLMs' accessibility on various real-world professionalism remains an unresolved issue, especially in the long-tailed fields with less attention, such as light industry, agriculture, and various service-related disciplines. Many popular benchmarks, such as MMLU [Hendrycks et al., 2020], GPQA [Rein et al., 2023], and MMLU-pro [Wang et al., 2024b], evaluate LLMs' abilities across different fields, while mainly focus on common fields like mathematics, physics, chemistry, biology, and law, limiting these benchmarks' practical significance on many real-world professionalisms. These benchmarks fail to cover the diverse and long-tail knowledge accumulated by humans. Moreover, large language models have achieved very high scores on these benchmarks, making them lose their value as challenging frontiers.

To address the gap, we introduce *SuperGPQA*, a comprehensive evaluation at the boundaries of human knowledge covering the evaluation of 285 graduate-level disciplines' knowledge and reasoning capacities. *SuperGPQA* provides at least 50 questions for each graduate-level disciplines to guarantee its accessibility on various real-world professionalism. *SuperGPQA* is developed by a large-scale human-LLM collaboration system, with crowd-sourcing annotators, experts, and state-of-the-art (**SOTA**) LLMs participating in, and then verified by a rigorous 3-stage quality inspection process, to guarantee its reliability. Moreover, *SuperGPQA* is qualified as a challenging frontier for SOTA reasoning LLMs, instruct LLMs, and base LLMs, where the best LLMs (e.g., o1 and Deepseek-R1) only achieve a score of around 60.

For building *SuperGPQA*, we propose a large-scale human-LLM collaboration system and share the valuable lessons learned in the paper. We divide the annotation system of *SuperGPQA* into three major stages: **Source Screening**, **Transcription**, and **Quality Inspection**. **During the source screening stage**, expert annotators collect credible resources of different disciplines' questions to guarantee the reliability and difficulty of the raw questions. **During the transcription stage**, crowd-sourcing annotators are asked to revise or translate the raw questions to multiple-choice questions, generate complementary confusion options, and estimate the difficulty and reliability of these questions. Crowd-sourcing annotators estimate the difficulty and reliability of candidate questions based on both expert judgments and the accuracy of LLMs' responses during the annotation process. Crowd-sourcing annotators, rigorous real-time plagiarism checks with existing candidate questions, and a robust filtering system based on SOTA LLMs are adopted in the transcription stage to reduce the waste of funding and expert manpower. **During the quality inspection stage**, we adopt a rigorous three-stage quality inspection process: *i.* we select suspicious candidate questions based on a checklist of LLMs' responses; *ii.* expert annotators review the suspicious candidate questions with unrestricted access to the web and revise these questions; *iii.* the easy questions are further tailored based on the accuracy of LLMs' responses to guarantee the discrimination of *SuperGPQA*.

We share several major insights based on the evaluation results of *SuperGPQA*:

- **Reasoning capacities matter**. The reasoning models (e.g., DeepSeek-R1, o1-2024-12-17) achieve better performance than general non-reasoning models in **SuperGPQA**.
- **Instruction tuning is very helpful**. For example, the results (47.40, 40.75) of DeepSeek-V3 and Qwen2.5-72B-Instruct are better than the results (32.14, 34.33) of DeepSeek-V3-Base and Qwen2.5-72B a lot, respectively.
- **More powerful LLMs lead to more balanced results**. The results of simple, middle, and hard splits of DeepSeek-R1 are 63.59, 63.63, and 56.87. In contrast, the results of easy, middle, and hard splits of Qwen2.5-14B-Instruct are 44.82, 37.90, and 19.97.
- **Models are better in newer versions.** For example, the results of GPT-4o-2024-11-20, GPT-4o-2024-08-06, and GPT-4o-2024-05-13 are 44.40, 41.64, and 39.76, respectively.

# 2 Data Collection

We solicit difficult questions from well-educated experts and crowd-sourcing annotators, where we consider experts as individuals having or pursuing a PhD, as in GPQA [Rein et al., 2023], and crowd-sourcing annotators as undergraduates and master students from top-tier universities. Figure 2 shows the three major stages of *SuperGPQA*'s data collection pipeline: Source Screening, Transcription,

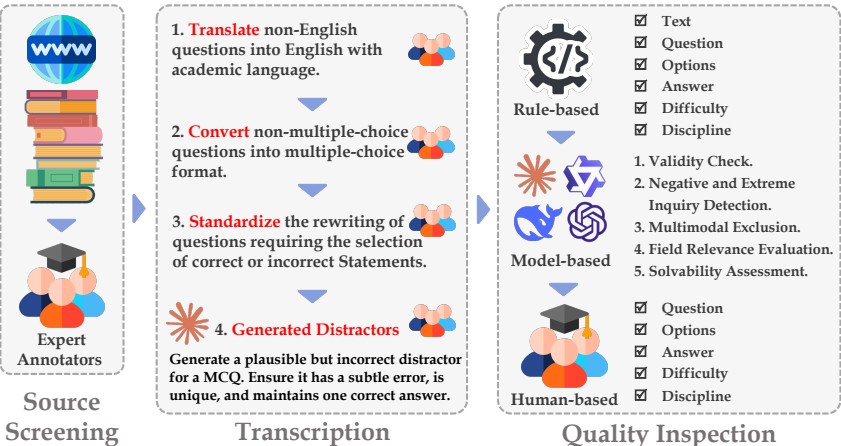

Figure 2: Data Collection Process of *SuperGPQA*.

and Quality Inspection. First, experts select credible resources of different disciplines' questions. Second, crowd-sourcing annotators revise the raw questions from credible resources to candidate questions. Finally, a rigorous human-LLM collaboration quality inspection process is adopted to select difficult and reliable questions from candidate questions.

> 💡 **Lessons of Source Screening:**
>
> - Crowd-Sourcing Annotators are not capable of collecting credible resources for multiple-choice question annotation with high expertise requirements.
> - The questions and answers (**QAs**) on exercise websites are not always reliable, even sometimes these QAs are claimed verified.
> - Multiple-choice questions modified from calculation and reasoning problems usually are more discriminatory than the original multi-choice questions available online.

During the source screening stage, only expert annotators are allowed to collect credible resources of different disciplines to guarantee the reliability and difficulty of the raw questions. In the early stage of collecting candidate questions of *SuperGPQA*, we trust crowd-sourcing annotators to collect credible resources themselves. However, the candidate questions based on the resources found by the crowd-sourcing annotators are always judged too easy or unreliable by expert annotators. As a result, a significant portion of early funding is wasted on ineffective questions annotated by crowd-sourcing annotators. Additionally, we point out that QAs on exercise websites are not always reliable.

In the early stage of *SuperGPQA* collection process, some annotators, even expert annotators, trust exercise websites that serve as corroboration of their reasoning process and answers. In the subsequent quality inspection stage, this proved to be a costly mistake, leading us to spend a significant amount of time and cost correcting erroneous answers derived from online exercise websites. Furthermore, we find that many SOTA LLMs, such as GPT-4o, o1-mini, and Gemini-flash, exhibit a high frequency of consistency in both process and answers with the erroneous processes and answers from several online exercise websites. The observations reveal that **the reliability of the solutions provided by online exercise websites is limited and there is a significant risk of data leakage**.

Expert annotators are asked to provide raw questions from credible resources with screenshots for further annotation in the source screening stage. The screenshot greatly eases the workload of quality inspection. We observe that the efficiency of quality inspection is greatly improved with the provided source screenshot. The priority order for selecting original questions is:

- Example problems with solutions from textbooks.
- Calculation and reasoning-needed questions with solutions from websites.
- Reasoning-needed multiple-choice questions with solutions from websites.
- General multiple-choice questions with solutions from websites.
- Questions only with answers but deemed correct by expert annotators.

A sampled list of credible resources certified by expert annotators is provided in Appendix M.

> 💡 **Lessons of Transcription:**
> - Crowd-sourcing annotators have low accuracy in judging generated distractors. For question types like selecting correct or incorrect options, it is easy to generate flawed distractors, requiring unified rewriting at this stage.

During the transcription stage, crowd-sourcing annotators are asked to revise the original questions into candidate questions. Specifically, the following operations are performed:

- Translate non-English questions into English with academic language.
- Convert non-multiple-choice questions into multiple-choice format.
- Standardize the rewriting of questions requiring the selection of correct or incorrect statements, as shown in Figure 7.
- Include region-specific information where necessary, such as specifying the country for laws mentioned in the questions, except for universally accepted rules.

The questions requiring the selection of correct or incorrect statements must be standardized as shown in Figure 7. Because we notice that even SOTA LLMs, e.g. Claude-3.5-Sonnet, GPT-4o-0806, suffer from generating correct suitable confounders for questions requiring the selection of correct or incorrect statements. The full transcription tutorial is provided in Appendix G.

> 💡 **Lessons of Quality Inspection:**
> - Questions where LLMs choose the same incorrect option are highly suspicious.
> - Cases where multiple or all SOTA LLMs make the same error often indicate that the LLMs have memorized explanations from incorrect exercise websites, based on SOTA LLMs' responses.

We refer to the data quality control methods in LIME and MMLU-Redux [Zhu et al., 2024, Gema et al., 2024, Wu et al., 2024], etc. The quality inspection process consists of three substages: **Rule-based Quality Inspection**, **LLM-based Quality Inspection**, and **Expert-based Quality Inspection**.

Candidate questions with clear formatting issues are identified and filtered out by rule-based quality inspection. The full checklist of rule-based quality inspection refers to subsection H.1. We then adopt several SOTA LLMs to generate responses and additional tags to these reserved candidate questions. LLM-based quality inspection includes validity checks, negative and extreme inquiry detection, multimodal exclusion, field relevance evaluation, completeness assessment, and discrimination tagging based on SOTA LLMs' responses. It provides an estimation of not only the correctness but also the discrimination of the candidate questions. Finally, we ask the expert annotators to re-annotate the suspicious candidate questions. The checklist for selecting suspicious candidate questions refer to subsection H.2. We adopt GPT-4o-2024-08-06, Gemini-2.0-flash, Doubao-1.5-pro-32k-250115, Claude-3.5-Sonnet, DeepSeek-R1, QwQ, Qwen-2.5-72B-Instruct as SOTA LLMs for selecting suspicious candidate questions. During the re-annotation process, following the rules stated in GPQA [Rein et al., 2023], expert annotators are asked to review and solve the given candidate questions with unrestricted access to the web. They spend over 30 minutes on each candidate question. The tutorial for manual quality review refers to subsection H.3.

# 3 Statistics

We propose *SuperGPQA*, designed as a comprehensive benchmark to probe the upper bounds of state-of-the-art Large Language Models' capabilities. With 26,529 questions spanning 13 disciplines, 72 fields, and 285 subfields, it substantially surpasses existing benchmarks in both scale and taxonomic depth. Compared to similar "hard" benchmarks such as GPQA (448 questions) and MMLU-Pro (12,032 questions), *SuperGPQA* not only contains a larger question pool but also features a more challenging format with an average of 9.67 options per question.

**Comprehensiveness and Discrimination.** In Table 1, the distribution of questions across disciplines reveals a notable concentration in STEM fields, with Science (9,838 questions), Engineering

Table 1: Statistics of *SuperGPQA*. Tokens are calculated with Tiktoken using `cl100k_base` encoding.

| Discipline | #Num. | Question #Tokens | | | Answer #Tokens | | | Options | |
|---|---|---|---|---|---|---|---|---|---|
| | | Max | Min | Avg | Max | Min | Avg | #Num. | #Tokens |
| Engineering | 7892 | 715 | 4 | 67.26 | 352 | 1 | 13.86 | 9.76 | 13.67 |
| Medicine | 2755 | 447 | 4 | 39.06 | 89 | 1 | 7.57 | 9.67 | 7.38 |
| Science | 9838 | 623 | 5 | 69.77 | 372 | 1 | 16.89 | 9.71 | 16.75 |
| Philosophy | 347 | 573 | 5 | 44.63 | 71 | 1 | 8.93 | 9.63 | 8.42 |
| Military Science | 205 | 336 | 7 | 30.03 | 50 | 1 | 6.76 | 9.29 | 6.40 |
| Economics | 314 | 314 | 5 | 36.15 | 98 | 1 | 7.85 | 9.86 | 7.12 |
| Management | 501 | 308 | 7 | 41.87 | 217 | 1 | 7.77 | 9.72 | 6.62 |
| Sociology | 143 | 107 | 6 | 25.31 | 75 | 1 | 7.34 | 9.87 | 6.91 |
| Literature | 1676 | 869 | 5 | 35.80 | 88 | 1 | 6.82 | 8.78 | 6.75 |
| History | 674 | 539 | 6 | 28.38 | 125 | 1 | 5.32 | 9.64 | 5.49 |
| Agriculture | 485 | 331 | 7 | 27.25 | 33 | 1 | 5.32 | 9.91 | 5.24 |
| Law | 656 | 560 | 6 | 66.38 | 89 | 1 | 12.17 | 9.73 | 11.45 |
| Education | 484 | 173 | 7 | 23.35 | 37 | 1 | 5.74 | 9.78 | 5.39 |
| Overall | 26529 | 869 | 4 | 58.42 | 372 | 1 | 12.86 | 9.67 | 12.64 |

(7,892 questions), and Medicine (2,755 questions) collectively accounting for 77.2% of the benchmark. While this distribution might appear uneven at first glance, it emerges from our rigorous question collection and validation process. During the data annotation phase, we source a comparable number of reference books for 285 subfields (the subfields are detailed in Table 10 and Table 11 in Appendix I). Note that we also use some open-source datasets for supplementation of *SuperGPQA* in Appendix M. However, the STEM disciplines yielded more questions meeting our stringent quality and difficulty criteria aforementioned in section 2, where the questions and options are filtered through rigorous rule-based, model-based and human-based pipeline. This natural emergence of STEM-heavy distribution aligns with the benchmark's goal of probing LLMs' upper-bound capabilities in complex reasoning tasks. Despite the relatively smaller representation of non-STEM disciplines (*e.g.*, Philosophy: 347, Literature: 1,676, History: 674 questions), our experiments demonstrate that these subsets effectively discriminate various SOTA LLMs' performance levels (detailed in subsection 4.2). This once again validates the discriminative power of our benchmark across all domains.

Table 2: Distribution of Difficulty Levels Across Disciplines. "#N" denotes the number of samples.

| Difficulty | Agro. | Econ. | Edu. | Eng. | Hist. | Law | Lit. & Arts | Mgmt. | Med. | Military Sci. | Phil. | Sci. | Socio. |
|---|---|---|---|---|---|---|---|---|---|---|---|---|---|
| Hard (#N) | 7 | 47 | 1 | 2458 | 3 | 57 | 12 | 6 | 217 | 4 | 27 | 4210 | 1 |
| Middle (#N) | 219 | 565 | 179 | 3462 | 180 | 343 | 496 | 236 | 1629 | 78 | 183 | 4133 | 45 |
| Easy (#N) | 259 | 261 | 304 | 1972 | 491 | 256 | 1168 | 259 | 909 | 123 | 137 | 1495 | 97 |
| Total | 485 | 873 | 484 | 7892 | 674 | 656 | 1676 | 501 | 2755 | 205 | 347 | 9838 | 143 |

**Difficulty.** The difficulty distribution across disciplines (Table 2) reveals varying levels of complexity. In STEM fields, we observe a more balanced distribution of difficulty levels. For instance, Engineering questions are distributed as 31.1% hard, 43.9% middle, and 25.0% easy, while Science shows 42.8% hard, 42.0% middle, and 15.2% easy. Non-STEM disciplines generally show a different pattern, with a higher proportion of easy and middle-difficulty questions. Notably, 42.33% of all questions require mathematical calculations or formal reasoning, with Science (66.34%) and Engineering (55.40%) showing the highest calculation rates.

**Sentence Length.** Question and answer length analysis reveals substantial variation across disciplines. The average question length is 58.42 tokens, with Literature questions showing the highest maximum length (869 tokens) and Engineering questions averaging 67.26 tokens. The "answer" options maintain a length pattern similar to the general options across different disciplines, averaging 12.86 tokens per option. Such a consistent length distribution across options aligns with one of the requirements of error option curation, which tries to confuse the models by providing confusing options in similar lengths.

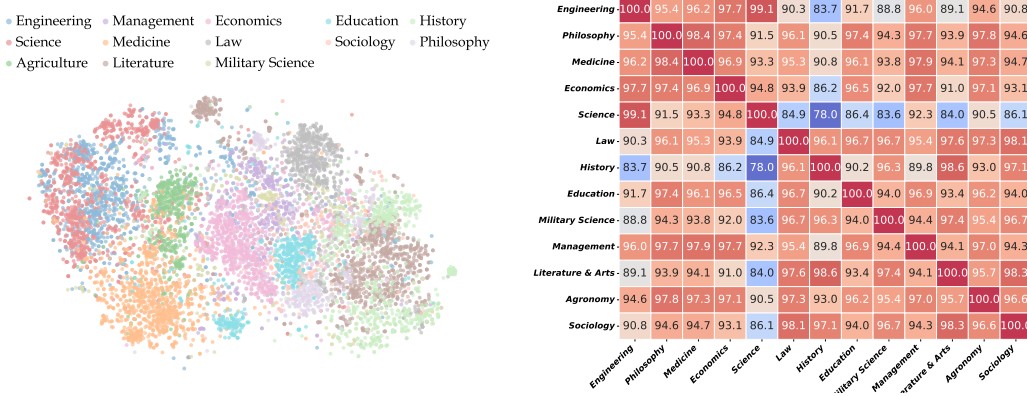

| | (a) The Visualization of the Text Embeddings. | (b) The Correlation Between Disciplines. |

Figure 3: Semantic Visualization.

**Semantic Visualization.** In Figure 3a, we employ t-SNE visualization of question-answer pair embeddings to visualize the distribution *SuperGPQA*[2] The resulting visualization demonstrates clear clustering patterns across disciplines while maintaining substantial overlap in conceptual spaces. The Engineering and Science demonstrate the highest degree of embedding overlap, suggesting strong semantic similarities in their Q&A patterns. The humanities cluster (Literature, Philosophy, History) shows diffuse boundaries but maintains distinct centroids. The Military Science exhibits relatively isolated embedding patterns, indicating unique domain-specific language. This analysis shows aligning conclusions from Figure 3b and reveals that the semantic structure of the *SuperGPQA* pairs reflects both the traditional organization of academic disciplines and their natural intellectual relationships, effectively capturing both domain-specific knowledge and cross-disciplinary connections.

## 4 Experiments

### 4.1 Baseline Models

We evaluate 6 reasoning models (o3-mini has three modes), 28 chat models and 17 base models on *SuperGPQA*, which includes closed-source models, open-source models, and fully open-source models. The full list of baselines could be found at subsection D.1. Reasoning models and chat models are evaluated using a zero-shot approach, while base models are assessed using a five-shot evaluation. Specifically, the five-shot evaluation for base models follow a similar methodology to MMLU-Pro. The specific prompts employed for both the zero-shot and five-shot evaluations are detailed in Appendix K. For all main results, the temperature is set to 0. The maximum number of new tokens is set to 32K for reasoning models, while for all other models, it is set to 4K.

### 4.2 Main Results

We present the results of the baselines (Table 3) and disciplines (Table 4). Due to the page limit, only part of the models are selected as representatives in the main texts, and the results for the comprehensive baselines are located in subsection D.2. Moreover, we provide detailed results across discipline-field-subfield for each model in the supplementary materials. In Table 3, we have the following observations. (1) The top-performed reasoning models (e.g., DeepSeek-R1, o1-2024-12-17) achieve better performance in *SuperGPQA*. (2) Instruction tuning can greatly improve the performance in *SuperGPQA*. For instance, the results (47.40, 40.75) of DeepSeek-V3 and Qwen2.5-72B-Instruct are significantly better than the results (32.14, 34.33) of DeepSeek-V3-Base and Qwen2.5-72B, respectively. (3) More powerful LLMs achieve more balanced results. For example, the results of simple, middle and hard splits of DeepSeek-R1 are 63.59, 63.63 and 56.87. In contrast, the results of simple, middle and hard splits of Qwen2.5-14B-Instruct are 44.82, 37.90 and 19.97.

---

[2]We use the `gte-large-en-v1.5` [Li et al., 2023, Zhang et al., 2024b] encoding model and set the t-SNE parameters as: perplexity 100, learning rate 500, and 1000 iterations.

**Difficulty-Specific Capabilities.** The difficulty stratification in *SuperGPQA* reveals distinct capability patterns between reasoning-focused and knowledge-oriented LLMs. In Table 3, the hard split specifically challenges models' reasoning capacities, while easy and middle splits better reflect factual knowledge mastery. For instance, the o3-mini series exhibits lower scores than Doubao-1.5-pro-32k-250115 on easy and middle splits, yet surpasses it significantly on hard questions. This dichotomy suggests that Chat-oriented LLMs (e.g., Doubao series) excel at knowledge recall for common professional questions but struggle with complex reasoning in long-tail domains. And the Reasoning-specialized models demonstrate superior performance on hard questions through enhanced logical processing, despite potential compromises in broad knowledge coverage.

**Observations for Reasoning Models.** Surprisingly, the performance gap between DeepSeek-R1 and DeepSeek-R1-Zero is relatively small. In Table 4, the DeepSeek-R1 only beats DeepSeek-R1-Zero in two disciplines (i.e., Science and Engineering). In the other disciplines, the DeepSeek-R1-Zero is slightly better than DeepSeek-R1, which leaves the optimal training paradigm of the reason models an open question. From the performances across different dimensions, compared to the o1-2024-12-17 model, the newer o1-mini and o3-mini models show decreasing scores except in science and engineering, suggesting a potential data leakage.

Table 3: **Performance on *SuperGPQA*.** The highest score is in $\boxed{box}$; the second-best score is in **bold**, and the third-best score is in underlined.

| Model | Overall | Easy | Middle | Hard |
|---|---|---|---|---|
| *Reasoning Models* | | | | |
| Gemini-2.5-Pro-Exp-0325 | $\boxed{63.56}$ | $\boxed{65.15}$ | **64.10** | $\boxed{60.91}$ |
| DeepSeek-R1 | **61.82** | 63.59 | 63.63 | 56.87 |
| Seed-Thinking-v1.5 | 61.34 | 59.14 | $\boxed{64.50}$ | **58.48** |
| o1-2024-12-17 | 60.24 | 64.40 | 61.44 | 53.67 |
| DeepSeek-R1-Zero | 60.24 | 62.61 | **65.06** | 50.99 |
| claude-3-7-sonnet-20250219-thinking | 58.31 | 60.91 | 59.27 | 53.87 |
| Qwen3-235B-A22B-thinking | 56.79 | 57.13 | 60.95 | 49.48 |
| o3-mini-2025-01-31-high | 55.22 | 53.05 | 56.09 | 56.16 |
| QwQ-32B | 54.43 | 54.12 | 57.65 | 49.42 |
| o3-mini-2025-01-31-medium | 52.69 | 51.30 | 53.79 | 52.37 |
| Qwen3-32B-thinking | 49.72 | 51.75 | 54.28 | 39.89 |
| o3-mini-2025-01-31-low | 48.03 | 48.80 | 50.21 | 43.53 |
| *Chat Models* | | | | |
| Doubao-1.5-pro-32k-250115 | $\boxed{55.09}$ | 57.70 | $\boxed{60.15}$ | **43.80** |
| Llama-4-Maverick-17B-128E-Instruct | **54.47** | 58.92 | **58.08** | 43.59 |
| gpt-4.1-2025-04-14 | 53.68 | **60.34** | 55.92 | 42.67 |
| grok-3-beta | 53.61 | 59.81 | 54.57 | $\boxed{45.21}$ |
| claude-3-7-sonnet-20250219 | 52.21 | $\boxed{60.64}$ | 55.36 | 37.70 |
| Qwen3-235B-A22B-nothink | 49.25 | 52.59 | 53.48 | 38.52 |
| claude-3-5-sonnet-20241022 | 48.16 | 59.04 | 51.91 | 29.99 |
| DeepSeek-V3 | 47.40 | 55.63 | 50.11 | 33.86 |
| MiniMax-Text-01 | 45.11 | 54.51 | 48.60 | 28.98 |
| Llama-4-Scout-17B-16E-Instruct | 44.82 | 50.47 | 49.06 | 31.56 |
| Qwen3-32B-nothink | 44.78 | 48.62 | 49.58 | 32.57 |
| gpt-4o-2024-11-20 | 44.40 | 56.84 | 48.75 | 23.50 |
| Llama-3.1-405B-Instruct | 43.14 | 56.06 | 46.31 | 23.70 |
| gpt-4o-2024-08-06 | 41.64 | 55.22 | 45.11 | 20.98 |
| Qwen2.5-72B-Instruct | 40.75 | 48.84 | 45.42 | 24.10 |
| gpt-4o-2024-05-13 | 39.76 | 53.37 | 42.38 | 20.45 |
| Qwen2.5-32B-Instruct | 38.76 | 47.42 | 43.05 | 22.13 |
| Llama-3.3-70B-Instruct | 37.69 | 49.68 | 40.68 | 19.55 |
| OLMo-2-1124-13B-Instruct | 18.66 | 27.10 | 17.85 | 10.74 |
| MAP-Neo-7B-Instruct-v0.1 | 17.05 | 23.26 | 16.62 | 10.95 |
| OLMo-2-1124-7B-Instruct | 16.81 | 22.80 | 15.82 | 11.90 |
| *Base Models* | | | | |
| Llama-4-Maverick-17B-128E | $\boxed{37.83}$ | $\boxed{46.98}$ | $\boxed{41.28}$ | $\boxed{22.06}$ |
| Qwen2.5-72B | **34.33** | **46.20** | **38.12** | 15.01 |
| Qwen2.5-32B | 33.16 | 45.12 | 36.58 | 14.34 |
| Qwen3-14B-Base | 33.02 | 42.75 | 36.42 | 16.70 |
| DeepSeek-V3-Base | 32.14 | 41.28 | 34.50 | **18.20** |
| Llama-4-Scout-17B-16E | 31.51 | 41.31 | 34.00 | 16.60 |
| Qwen3-30B-A3B-Base | 30.36 | 39.04 | 33.31 | 15.94 |
| Llama-3.1-405B | 25.23 | 37.58 | 25.12 | 11.86 |
| Mixtral-8x22B-v0.1 | 22.41 | 32.78 | 21.67 | 12.26 |
| OLMo-2-1124-13B | 16.07 | 27.24 | 14.41 | 6.57 |
| MAP-Neo-7B | 15.76 | 22.86 | 14.64 | 9.83 |
| OLMo-2-1124-7B | 15.15 | 24.43 | 13.83 | 7.15 |

**Effect of Subfield Information in Prompts.** We conduct zero-shot evaluations under two conditions: (1) zero-shot-with-subfield, where the prompt includes a description of the problem's subfield, and (2) zero-shot-without-subfield, where no such information is provided. The prompts of zero-shot-with-subfield are shown in subsection L.1. We evaluate Qwen2.5-Instruct models ranging from 0.5B to 72B parameters across these two settings. Figure 4a show that incorporating subfield information generally leads to improved performance, particularly for larger models. For example, Qwen2.5-72B-Instruct achieves an accuracy of 41.93% in the zero-shot-with-subfield setting, compared to 40.82% without subfield annotations. Similarly, Qwen2.5-32B-Instruct improves from 39.13% to 39.65%, and Qwen2.5-14B-Instruct sees a minor increase from 35.36% to 35.78%, suggesting that additional contextual information helps larger models refine their reasoning by narrowing down the relevant domain. However, for smaller models like Qwen2.5-0.5B-Instruct and Qwen2.5-1.5B-Instruct, the subfield information does not lead to a noticeable gain in accuracy, which indicates that smaller models may lack the capacity to leverage fine-grained domain-specific cues effectively.

**Robustness Analysis.** Benchmark evaluations can be significantly influenced by slight variations in prompts, leading to inconsistencies in model ranking and overall assessment reliability. To investigate this phenomenon, we conduct robustness experimentacross Qwen2.5-Instruct models (0.5B~72B), employing 24 distinct yet semantically equivalent prompts in a zero-shot setting, using the same evaluation parameters as in the main results. Specifically, we employ 4 types of initial prompts and 6 types of question formats, resulting in a combination of 24 different prompt styles to verify the robustness of our bench. The combination of initial prompt 1 and question format 1 is the default prompt for our evaluation. The detailed prompts are shown in the subsection L.2. Figure 4b shows that larger models exhibit increased robustness, as a steady improvement in accuracy from 9.80% of

Table 4: **Performance on *SuperGPQA*.** We present LLMs' performance on different disciplines. The highest, the second-best and the third-best scores are shown in $\boxed{box}$, **bold** and underlined, respectively.

| Model | Agr. | Econ. | Edu. | Eng. | Hist. | Law | Lit & Arts | Mgt. | Med. | Mil Sci. | Phil. | Sci. | Soc. |
|---|---|---|---|---|---|---|---|---|---|---|---|---|---|
| *Reasoning Models* | | | | | | | | | | | | | |
| Gemini-2.5-Pro-Exp-0325 | 55.05 | 63.12 | 51.86 | 64.47 | 66.62 | **65.24** | 61.52 | **57.49** | 58.22 | 63.90 | 66.86 | 65.56 | 63.64 |
| DeepSeek-R1 | **54.43** | 66.09 | 54.75 | 63.10 | 55.19 | 65.24 | 52.45 | 57.09 | 59.93 | 57.07 | 63.11 | 63.69 | 67.13 |
| DeepSeek-R1-Zero | 53.81 | **66.44** | 60.54 | 60.28 | 58.61 | 66.77 | **56.86** | 59.68 | **60.65** | 58.54 | 63.69 | 59.93 | **67.13** |
| Seed-Thinking-v1.5 | 52.99 | 67.70 | **56.40** | **63.53** | 42.28 | 62.65 | 44.63 | 56.49 | **61.05** | 55.61 | **65.13** | **64.08** | 59.44 |
| o1-2024-12-17 | 50.93 | 61.17 | 53.31 | 59.17 | **60.98** | 63.87 | 55.79 | **57.29** | 62.25 | **60.49** | 61.38 | 61.76 | 64.34 |
| Qwen3-32B-thinking | 49.07 | 57.73 | 50.00 | 52.10 | 31.75 | 49.24 | 36.04 | 49.10 | 51.94 | 49.76 | 54.18 | 49.88 | 55.24 |
| claude-3-7-sonnet-20250219-thinking | 47.22 | 60.25 | 55.17 | 58.38 | 57.12 | 61.43 | 53.28 | 51.50 | 53.65 | 56.59 | 60.23 | 61.06 | 64.34 |
| Qwen3-235B-A22B-thinking | 47.22 | 61.97 | 53.93 | 60.83 | 38.87 | 59.15 | 44.45 | 50.50 | 59.64 | 53.17 | 60.81 | 56.35 | 54.55 |
| QwQ-32B | 46.19 | 59.79 | 53.31 | 55.27 | 35.31 | 54.12 | 37.65 | 48.50 | 53.03 | 52.68 | 54.18 | 58.69 | 53.85 |
| o3-mini-2025-01-31-high | 41.03 | 54.07 | 46.28 | 56.41 | 36.80 | 45.73 | 39.44 | 51.50 | 51.72 | 46.34 | 51.01 | 61.72 | 46.15 |
| o3-mini-2025-01-31-medium | 40.62 | 51.32 | 41.74 | 53.83 | 35.01 | 43.60 | 37.11 | 48.50 | 50.34 | 45.85 | 48.13 | 58.79 | 44.06 |
| o3-mini-2025-01-31-low | 37.32 | 45.25 | 38.43 | 48.20 | 32.94 | 39.79 | 36.22 | 43.71 | 48.09 | 41.46 | 44.09 | 53.24 | 45.45 |
| *Chat Models* | | | | | | | | | | | | | |
| Doubao-1.5-pro-32k-250115 | 50.93 | 65.06 | **55.58** | **55.60** | 42.88 | **58.84** | 42.30 | 53.29 | 59.13 | 54.15 | 61.96 | 55.54 | 51.75 |
| Llama-4-Maverick-17B-128E-Instruct | **48.87** | 59.91 | 50.41 | 56.06 | 51.93 | 57.93 | 55.49 | 56.23 | 50.73 | 54.47 | 54.62 | 55.24 |
| claude-3-7-sonnet-20250219 | 47.84 | 56.82 | 53.51 | 51.67 | 57.86 | 58.84 | 51.55 | 52.50 | 53.28 | **58.54** | 57.93 | 50.82 | 65.03 |
| gpt-4.1-2025-04-14 | 47.22 | 55.78 | 54.55 | 51.93 | **55.79** | 58.08 | **51.37** | 53.09 | **57.17** | 56.10 | **58.21** | 53.87 | **61.54** |
| Qwen3-235B-A22B-nothink | 46.60 | 52.35 | 49.38 | 49.82 | 35.01 | 50.61 | 39.68 | 47.11 | 52.20 | 50.73 | 53.03 | 50.25 | 50.35 |
| grok-3-beta | 46.19 | 51.78 | 48.14 | 53.85 | 55.34 | 55.18 | 49.76 | 51.30 | 51.51 | 56.59 | 51.01 | **55.35** | 55.94 |
| Llama-3.1-405B-Instruct | 43.09 | 49.71 | 45.45 | 41.89 | 50.00 | 55.34 | 43.20 | 47.70 | 49.04 | 52.20 | 48.41 | 39.80 | 49.65 |
| gpt-4o-2024-11-20 | 42.27 | 46.74 | 50.00 | 42.83 | 52.52 | 53.81 | 46.72 | 47.31 | 52.52 | 52.20 | 52.74 | 40.67 | 54.55 |
| gpt-4o-2024-05-13 | 41.44 | 40.78 | 44.21 | 37.47 | 52.90 | 52.90 | 45.11 | 48.32 | 48.82 | 50.73 | 45.24 | 35.41 | 53.85 |
| DeepSeek-V3 | 41.24 | 49.48 | 43.80 | 47.21 | 47.18 | 51.07 | 45.23 | 48.50 | 46.10 | 48.29 | 43.23 | 48.30 | 55.94 |
| Qwen2.5-72B-Instruct | 41.24 | 46.62 | 44.42 | 41.12 | 30.71 | 45.88 | 34.90 | 42.32 | 45.74 | 45.37 | 43.52 | 39.33 | 46.15 |
| Qwen3-32B-nothink | 40.21 | 51.09 | 48.55 | 45.82 | 29.82 | 44.51 | 34.01 | 45.71 | 47.04 | 47.80 | 47.84 | 45.43 | 46.85 |
| MiniMax-Text-01 | 39.79 | 49.60 | 47.52 | 44.88 | 45.25 | 54.27 | 43.02 | 48.90 | 47.08 | 48.29 | 45.53 | 43.81 | 53.85 |
| gpt-4o-2024-08-06 | 39.59 | 40.78 | 44.42 | 38.84 | 50.89 | 54.12 | 46.30 | 45.51 | 50.49 | 50.73 | 47.84 | 38.41 | 53.85 |
| Mistral-Large-Instruct-2411 | 38.76 | 42.27 | 42.15 | 39.66 | 44.96 | 47.87 | 41.11 | 43.31 | 43.23 | 48.78 | 42.07 | 39.24 | 50.35 |
| Llama-4-Scout-17B-16E-Instruct | 37.73 | 49.60 | 45.04 | 45.78 | 32.05 | 45.73 | 36.04 | 44.51 | 47.22 | 48.78 | 43.23 | 45.58 | 46.15 |

(a) Impact of subfield information.

(b) Model robustness analysis.

Figure 4: (a) Accuracy comparison of Qwen2.5 models with and without subfield information in prompts. Larger models benefit more from additional context. (b) Robustness evaluation across 24 semantically equivalent prompts, indicating larger models exhibit higher stability with lower variance.

Qwen2.5-0.5B-Instruct to 41.15% of Qwen2.5-72B-Instruct while variance decreases ($\sigma = 1.37$ for the smallest model, dropping to $\sigma \approx 0.27 \sim 0.36$ in the largest models).

## 5 Discussion: From Benchmarking to Training

### 5.1 Reliability: The Foundation for Model Development

*SuperGPQA* demonstrates strong reliability through the following dimensions:

**Correlation with Representative Benchmarks** As shown in Figure 5, we calculate the correlation between *SuperGPQA* and existing mainstream benchmarks with Chatbot Arena. We select evaluation results from nearly 50 models to calculate the correlation. For other benchmarks, we select results reported on open-source leaderboards. *SuperGPQA* achieves the highest correlation coefficient (87.6) with the dynamic Chatbot Arena Elo scores compared to other static benchmarks like MMLU-Pro (83.1) and GPQA (85.2). This aligns with human preferences for model outputs, validating

*SuperGPQA*'s practical relevance. Furthermore, *SuperGPQA* exhibits strong alignment (95.6) with GPQA, indicating consistency in assessing reasoning capabilities across difficult specialized domains.

**Robustness Across Prompt Variations.** Our robustness analysis (Figure 4) reveals that *SuperGPQA* maintains stable model rankings under 24 semantically equivalent prompts. For example, most of the Qwen2.5 series chat models show accuracy variances under $0.5$ across different prompts. This stability stems from our iterative Human-LLM collaborative filtering mechanism, which eliminates prompt-sensitive questions during quality inspection.

## 5.2 Distinctiveness: Illuminating Model Differences for Model Training

*SuperGPQA* provides granular insights into model capabilities through its discriminative power as follows:

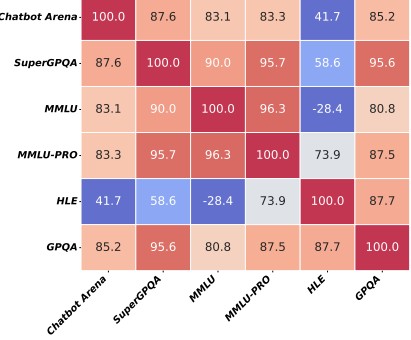

Figure 5: The Correlation Coefficients between Different Benchmarks. MMLU is tested under the 5-shot setting.

**Scaling Still Matters** Larger models exhibit higher performance. *SuperGPQA* reveals a significant gap for different sizes: Qwen2.5-72B-Instruct (40.75%) outperforms its 7B counterpart (28.78%) by 44.5%. This suggests *SuperGPQA* could reveal that existing smaller models may overfit to narrow domains rather than acquiring generalized reasoning.

**Humanities as Discriminative Frontier** We conduct a performance discrimination analysis between the top and bottom–performed models across disciplines, and find that current models mostly struggle disproportionately with non-STEM disciplines (subsection D.2). While models exhibit minimal differentiation in Management, Military Science, and Engineering, they show pronounced performance gaps in Law and History. This inverse relationship between STEM/non-STEM discriminative power – with humanities disciplines demonstrating greater model differentiation on average – underscores systemic underemphasis on linguistic nuance and contextual reasoning in current models.

**Reasoning vs. Chat** Reasoning models like DeepSeek-R1 (61.82%) outperform instruction-tuned counterparts (Doubao-1.5-Pro: 55.09%) by 7.73%, which shows the need for explicit reasoning capacity beyond standard instruction tuning. Moreover, for Qwen3 and Qwen2.5, the Qwen3-32B trained with a mixture of "thinking" (reasoning) and "no-thinking" data outperforms the Qwen2.5-72B-Instruct from last generation, where only "no-thinking" data has been used. This suggests that integrating reasoning traces during pre-training could largely bridge the capability gap.

**Reasoning Through Sampling** Pass@K analysis (Figure 6) reveals that Qwen2.5-72B-Instruct improves by 33% (from 41% to 74%) as $K$ increases from 1 to 32, compared to Doubao-1.5-pro's 17% gain. This demonstrates the untapped potential of incorporating reasoning-encouraged training pipeline like reinforcement learning with verifiable reward (RLVR) or supervised fine-tuning with distilled reasoning data.

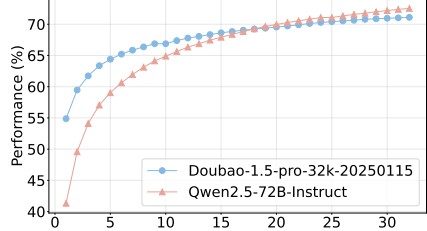

Figure 6: Performance comparison of Qwen2.5-72B-Instruct and Doubao-1.5-pro-32k-20250115 under Pass@K.

## 6 Conclusion

In this paper, we have identified a significant limitation in existing LLM evaluation benchmarks, particularly in assessing models' capabilities across long-tailed professional fields. To address this, we introduced *SuperGPQA*, a comprehensive evaluation framework covering 285 graduate-level disciplines, developed through a large-scale human-LLM collaboration and rigorous quality inspection. Extensive experiments reveal the importance of reasoning capacities, the effectiveness of instruction tuning, the relationship between model power and performance balance, and etc, which can provide insightful findings for further improvements.

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

# A Contributions and Acknowledgements

Multimodal Art Projection (M-A-P) is a non-profit open-source AI research community, ran by donation. The community members are working on research topics in a wide range of spectrum, including but not limited to the pre-training paradigm of foundation models, large-scale data collection and processing, and the derived applications on coding, reasoning and music generation.

Our team members contribute to the development of *SuperGPQA* from the following perspectives:

- Data Annotation Management
- Data Annotation
- Data Quality Inspection
- Model Evaluation
- Result Analysis
- Paper Writing

**Leading Authors**

- Xinrun Du, M-A-P
- Yifan Yao, M-A-P
- Kaijing Ma, M-A-P
- Bingli Wang, SAU
- Tianyu Zheng, M-A-P, Tiktok
- King Zhu, M-A-P, OPPO
- Minghao Liu, 2077.AI

**Outstanding Contributors**

- Yiming Liang, M-A-P, CASIA
- Xiaolong Jin, Purdue University
- Zhenlin Wei, HEU
- Chujie Zheng, Tsinghua University

**Core Contributors (Alphabet Order)**

- Kaixin Deng, CDUT
- Shawn Gavin, M-A-P
- Shian Jia, Zhejiang University
- Sichao Jiang, Zhejiang University
- Qinrui Li, UCSB
- Rui Li, Peking University
- Sirun Li, Peking University
- Yizhi Li, The University of Manchester
- Yunwen Li, CUHK-Shenzhen
- Yiyan Liao, Peking University
- David Ma, M-A-P
- Yuansheng Ni, M-A-P
- Haoran Que, Zhipu
- Qiyao Wang, DUT
- Zekun Moore Wang, M-A-P, Beihang University
- Zhoufutu Wen, ByteDance.Inc
- Siwei Wu, The University of Manchester
- Tyshawn Hsing, M-A-P
- Ming Xu, NJUPT
- Zhenzhu Yang, M-A-P, CUGB
- Junting Zhou, Peking University

**Contributors (Alphabet Order)**

- Yuelin Bai, M-A-P
- Xingyuan Bu, Alibaba.Inc
- Chenglin Cai, 01.AI
- Liang Chen, Peking University
- Yifan Chen, ByteDance.Inc
- Chengtuo Cheng, Abaka.AI
- Tianhao Cheng, Fudan University
- Keyi Ding, Hangzhou Dianzi University
- Siming Huang, The University of Melbourne
- Yun Huang, NUS
- Yaoru Li, Zhejiang University
- Yizhe Li, Zhejiang University
- Zhaoqun Li, Zhejiang University
- Tianhao Liang, Zhejiang University

- Chengdong Lin, Hangzhou Dianzi University
- Hongquan Lin, University of Science and Technology of China
- Yinghao Ma, Queen Mary University of London
- Tianyang Pang, ByteDance.Inc
- Zhongyuan Peng, Alibaba.Inc
- Zifan Peng, HKUST-Guangzhou
- Qige Qi, ByteDance.Inc
- Shi Qiu, Peking University
- Xingwei Qu, The University of Manchester
- Shanghaoran Quan, Beihang University
- Yizhou Tan, Harvard University
- Chenqing Wang, 2077.AI
- Hao Wang, Beihang University
- Yiya Wang, Peking University
- Yubo Wang, University of Waterloo
- Zili Wang
- Jiajun Xu, Meta
- Kexin Yang
- Ruibin Yuan, HKUST
- Yuanhao Yue, Fudan University
- Tianyang Zhan, ByteDance.Inc
- Chun Zhang, ByteDance.Inc
- Jinyang Zhang, Zhejiang University
- Xingjian Zhang, Princeton University
- Xiyue Zhang, Peking University
- Yue Zhang, ByteDance.Inc
- Yongchi Zhao, Alibaba.Inc
- Xiangyu Zheng, Fudan University
- Chenghua Zhong, USTB

**Organization and Sponsor Committee (Alphabet Order)**

- Meng Cao, MBZUAI
- Yang Gao, Nanjing University
- Zhoujun Li, Beihang University
- Dayiheng Liu
- Qian Liu, Tiktok
- Tianyu Liu
- Shiwen Ni, SIAT-CAS
- Junran Peng, USTB
- Yujia Qin, ByteDance.Inc
- Wenbo Su
- Guoyin Wang, ByteDance.Inc
- Shi Wang, ICT-CAS
- Jian Yang, Beihang University
- Min Yang, SIAT-CAS
- Xiang Yue, M-A-P
- Zhaoxiang Zhang, CASIA
- Wangchunshu Zhou, OPPO

**Corresponding Authors**

- Jiaheng Liu, M-A-P, Nanjing University
- Qunshu Lin, Abaka.AI
- Wenhao Huang, M-A-P, ByteDance.Inc
- Ge Zhang, M-A-P, ByteDance.Inc

## B  Limitation

While *SuperGPQA* advances LLM evaluation across specialized disciplines, several limitations warrant discussion. First, despite covering 285 graduate-level disciplines, the benchmark exhibits a STEM-heavy distribution (e.g., 9,838 Science questions vs. 143 Sociology questions), reflecting inherent challenges in sourcing expert-validated questions for long-tailed fields. Second, the benchmark currently focuses on text-based multiple-choice formats, limiting its ability to assess practical skills in applied disciplines like agriculture or medicine that require multimodal reasoning. Future iterations will address these gaps through broader international collaboration and task diversification.

## C  Ethical Statement

**Human annotation**: Over 80 expert annotators (PhD holders/candidates) and crowd-sourcing annotators (students from top universities) participated in the annotation process. All annotators have consented to voluntarily devote. No vulnerable populations were involved.

**Data provenance and licensing**: Questions were sourced from verified academic materials (e.g., textbooks, peer-reviewed problem sets) with explicit permission or under fair-use provisions. The benchmark will be released under ODC-BY License to grant proper commercial use while encouraging academic adaptation.

**Privacy and bias mitigation**: No personally identifiable information is included. While regional specificity was preserved for legal/contextual questions (e.g., country-specific laws), we explicitly avoided culturally sensitive content. The disciplinary coverage spans 13 major domains, though imbalances persist (see Table 2), which we transparently document.

## D  Comprehensive Experiment Settings and Results

### D.1  Full List of Baselines

The reasoning models include Deepseek-R1 and Deepseek-R1-Zero [Guo et al., 2025], o1 and o1-mini [OpenAI, 2024b], Seed-Thinking-v1.5 [Seed et al., 2025], Gemini-2.5-Pro [Team et al., 2025], Claude3.7-sonnet-thinking [claude, 2025], QwQ-32B [Team, 2025b], QwQ-32B-Preview [Team, 2024b], Qwen3 [Team, 2025a] series and o3-mini [OpenAI, 2025] series models. The chat models include closed-source models such as Doubao-1.5-pro, Qwen-max, Claude-3.5-sonnet [Anthropic, 2024], Claude-3.7-sonnet [claude, 2025], Gemini [Team, 2024a], GPT-4o [OpenAI, 2024a], Grok-3-Beta [grok, 2025], Yi-Lightning [Wake et al., 2024], gpt4.1 [opneai, 2025] series, and open-source models like MiniMax-Text-01 [Li et al., 2025], Qwen2.5 [Yang et al., 2024a] series, Llama-3.1 [Dubey et al., 2024] series, Llama-4 [Meta-Llama, 2025] series, Qwen3 [Team, 2025a] series, Mistral [Jiang et al., 2023] and Mixtral [Jiang et al., 2024] series, Gemma-2 [Team et al., 2024]series, Yi-1.5 [Young et al., 2024] series, Phi4 [Abdin et al., 2024], and Granite-3.1 [Granite Team, 2024]. Additionally, there are fully open-source models like MAP-Neo [Zhang et al., 2024a], K2 [Liu et al., 2025] and OLMo-2 [OLMo et al., 2024]. According to our test results, these models still show a significant gap when compared to industry standards level. The base models include Qwen2.5 [Yang et al., 2024a] series, Deepseek-V3 [Liu et al., 2024], Yi-1.5 [Young et al., 2024] series, Llama-3.1 [Dubey et al., 2024] series, Gemma-2 [Team et al., 2024] series, and Mistral [Jiang et al., 2023] and Mixtral [Jiang et al., 2024] series.

Reasoning models and chat models are evaluated using a zero-shot approach, while base models are assessed using a five-shot evaluation. Specifically, the five-shot evaluation for base models follow a similar methodology to MMLU-Pro. The specific prompts employed for both the zero-shot and five-shot evaluations are detailed in Appendix K. For most evaluations, we follow the default setting where the temperature is set to 0. Reasoning models are allowed a maximum of 32K new tokens, while all other models are capped at 4K tokens. In addition, several models are evaluated using their recommended inference configurations: Qwen3 series: zero-shot evaluations follow the official "thinking" mode and "nothink" mode settings, with temperatures set to 0.6 and 0.7 respectively. Claude-3.7-Sonnet (thinking mode): Evaluation adheres to the official constraints, with temperature fixed at 1.0 and a token budget of 32K and allowed a maximum of 40K new tokens. Seed-Thinking-v1.5 and Gemini 2.5 Pro: Evaluated with a temperature of 0.6.

## D.2 More Comprehensive Analysis of Baseline Performances

Table 5: **Detailed Performance Overview on *SuperGPQA*.** The highest score in each column is indicated with a $\boxed{box}$ ; the second-best score is in **bold**, and the third-best score is underlined.

| Model | Overall (sample) | Overall (subfield) | Overall (field) | Overall (discipline) | Easy (sample) | Middle (sample) | Hard (sample) |
|---|---|---|---|---|---|---|---|
| *Reasoning Models* | | | | | | | |
| Gemini-2.5-Pro-Exp-0325 | 63.56 | 63.74 | 62.48 | 61.81 | 65.15 | **64.10** | 60.91 |
| DeepSeek-R1 | **61.82** | **62.61** | **61.23** | 59.95 | 63.59 | 63.63 | 56.87 |
| Seed-Thinking-v1.5 | 61.34 | 61.86 | 60.63 | 57.85 | 59.14 | 64.50 | **58.48** |
| o1-2024-12-17 | 60.24 | 61.25 | 59.94 | 59.44 | 64.40 | 61.44 | 53.67 |
| DeepSeek-R1-Zero | 60.24 | 61.62 | 60.95 | 60.99 | 65.06 | 62.61 | 50.99 |
| claude-3-7-sonnet-20250219-thinking | 58.31 | 58.73 | 57.32 | 56.94 | 60.91 | 59.27 | 53.87 |
| Qwen3-235B-A22B-thinking | 56.79 | 58.42 | 56.95 | 53.96 | 57.13 | 60.95 | 49.48 |
| o3-mini-2025-01-31-high | 55.22 | 54.94 | 52.11 | 48.32 | 53.05 | 56.09 | 56.16 |
| QwQ-32B | 54.43 | 54.89 | 53.17 | 50.97 | 54.12 | 57.65 | 49.42 |
| o3-mini-2025-01-31-medium | 52.69 | 52.66 | 49.95 | 46.07 | 51.30 | 53.79 | 52.37 |
| Qwen3-30B-A3B-thinking | 51.84 | 52.02 | 49.84 | 46.08 | 50.50 | 55.16 | 47.79 |
| Qwen3-32B-thinking | 49.72 | 51.72 | 50.74 | 48.93 | 51.75 | 54.28 | 39.89 |
| Qwen3-14B-thinking | 49.64 | 50.48 | 49.20 | 45.13 | 49.62 | 52.80 | 44.40 |
| o3-mini-2025-01-31-low | 48.03 | 48.51 | 45.89 | 42.63 | 48.80 | 50.21 | 43.53 |
| o1-mini-2024-09-12 | 45.22 | 45.46 | 42.53 | 39.33 | 46.77 | 47.34 | 40.00 |
| Qwen3-8B-thinking | 43.98 | 45.05 | 43.66 | 40.37 | 46.33 | 47.11 | 36.18 |
| QwQ-32B-Preview | 43.59 | 44.40 | 43.19 | 41.63 | 46.46 | 47.40 | 34.07 |
| Qwen3-4B-thinking | 42.38 | 42.25 | 40.13 | 36.36 | 42.12 | 44.56 | 39.04 |
| Qwen3-1.7B-thinking | 31.15 | 31.45 | 30.12 | 28.04 | 34.96 | 32.86 | 24.14 |
| Qwen3-0.6B-thinking | 20.63 | 21.49 | 21.46 | 20.95 | 25.08 | 21.08 | 15.01 |
| *Chat Models* | | | | | | | |
| Doubao-1.5-pro-32k-250115 | 55.09 | 56.55 | 55.62 | 54.39 | 57.70 | 60.15 | **43.80** |
| Llama-4-Maverick-17B-128E-Instruct | **54.47** | **55.72** | **54.95** | 53.45 | 58.92 | **58.08** | 43.59 |
| gpt-4.1-2025-04-14 | 53.68 | 55.37 | 54.23 | 54.98 | 60.34 | 55.92 | 42.67 |
| grok-3-beta | 53.61 | 54.40 | 53.30 | 52.46 | 59.81 | 54.57 | 45.21 |
| claude-3-7-sonnet-20250219 | 52.21 | 54.35 | 53.91 | 55.09 | 60.64 | 55.36 | 37.70 |
| Doubao-1.5-pro-32k-241225 | 50.93 | 52.41 | 51.76 | 51.24 | 53.54 | 56.56 | 38.70 |
| Qwen-Max-2025-01-25 | 50.08 | 52.75 | 52.47 | 51.65 | 58.16 | 54.95 | 33.09 |
| Qwen3-235B-A22B-nothink | 49.25 | 50.48 | 49.56 | 48.24 | 52.59 | 53.48 | 38.52 |
| gpt-4.1-mini-2025-04-14 | 48.57 | 49.34 | 47.56 | 45.92 | 51.51 | 50.45 | 42.20 |
| claude-3-5-sonnet-20241022 | 48.16 | 51.38 | 51.23 | 53.15 | 59.04 | 51.91 | 29.99 |
| gemini-2.0-flash | 47.73 | 48.70 | 47.80 | 46.10 | 53.06 | 49.56 | 38.84 |
| DeepSeek-V3 | 47.40 | 49.10 | 48.31 | 47.35 | 55.63 | 50.11 | 33.86 |
| MiniMax-Text-01 | 45.11 | 47.46 | 46.97 | 47.06 | 54.51 | 48.60 | 28.98 |
| Llama-4-Scout-17B-16E-Instruct | 44.82 | 46.55 | 45.74 | 43.65 | 50.47 | 49.06 | 31.56 |
| Qwen3-32B-nothink | 44.78 | 46.53 | 45.96 | 44.21 | 48.62 | 49.58 | 32.57 |
| gpt-4o-2024-11-20 | 44.40 | 47.62 | 47.50 | 48.84 | 56.84 | 48.75 | 23.50 |
| Llama-3.1-405B-Instruct | 43.14 | 46.43 | 45.83 | 47.35 | 56.06 | 46.31 | 23.70 |
| Qwen3-30B-A3B-nothink | 41.81 | 43.38 | 42.57 | 40.61 | 47.30 | 45.77 | 29.21 |
| gpt-4o-2024-08-06 | 41.64 | 44.79 | 44.91 | 46.29 | 55.22 | 45.11 | 20.98 |
| Qwen2.5-72B-Instruct | 40.75 | 43.66 | 43.32 | 42.10 | 48.84 | 45.42 | 24.10 |
| Mistral-Large-Instruct-2411 | 40.65 | 43.38 | 43.13 | 43.37 | 52.92 | 43.28 | 22.81 |
| Qwen-Max-2024-09-19 | 39.96 | 42.93 | 42.16 | 41.62 | 50.23 | 43.63 | 22.60 |
| gpt-4o-2024-05-13 | 39.76 | 43.19 | 43.13 | 45.23 | 53.37 | 42.38 | 20.45 |
| Qwen3-14B-nothink | 39.58 | 41.07 | 39.97 | 39.45 | 45.48 | 43.69 | 26.26 |
| Qwen2.5-32B-Instruct | 38.76 | 41.18 | 40.40 | 39.43 | 47.42 | 43.05 | 22.13 |
| Llama-3.3-70B-Instruct | 37.69 | 40.56 | 40.15 | 41.12 | 49.68 | 40.68 | 19.55 |
| phi-4 | 37.65 | 39.59 | 38.61 | 37.66 | 45.43 | 40.91 | 23.69 |
| Qwen3-8B-nothink | 36.54 | 37.98 | 37.11 | 35.36 | 42.63 | 39.20 | 25.43 |
| Qwen2.5-14B-Instruct | 35.15 | 37.72 | 37.41 | 36.07 | 44.82 | 37.90 | 19.97 |
| Llama-3.1-70B-Instruct | 34.86 | 38.94 | 39.18 | 40.57 | 48.22 | 37.85 | 15.22 |
| gpt-4.1-nano-2025-04-14 | 33.71 | 35.00 | 34.14 | 32.63 | 40.34 | 35.59 | 23.30 |
| Yi-Lighting | 33.42 | 36.57 | 36.45 | 36.92 | 43.38 | 35.32 | 19.35 |
| Qwen3-4B-nothink | 31.61 | 32.90 | 31.65 | 30.90 | 38.20 | 33.88 | 20.60 |
| Mixtral-8x22B-Instruct-v0.1 | 29.23 | 32.14 | 32.28 | 32.82 | 42.52 | 29.73 | 13.82 |
| Qwen2.5-7B-Instruct | 28.78 | 30.78 | 30.37 | 30.63 | 37.77 | 30.98 | 15.23 |
| gemma-2-27b-it | 27.43 | 30.50 | 30.42 | 31.30 | 40.90 | 27.45 | 12.64 |
| Yi-1.5-34B-Chat | 26.03 | 28.81 | 28.84 | 29.08 | 36.99 | 26.74 | 12.81 |
| Mistral-Small-Instruct-2409 | 25.89 | 28.46 | 28.69 | 28.59 | 37.93 | 25.89 | 12.70 |
| gemma-2-9b-it | 24.04 | 26.89 | 27.06 | 27.74 | 37.81 | 23.05 | 10.60 |
| Qwen2.5-3B-Instruct | 23.31 | 25.45 | 25.86 | 25.57 | 33.10 | 23.50 | 12.24 |
| Qwen3-1.7B-nothink | 23.17 | 24.49 | 24.33 | 24.18 | 29.38 | 23.39 | 16.00 |
| Yi-1.5-9B-Chat | 23.17 | 25.32 | 25.65 | 26.07 | 32.75 | 24.05 | 11.19 |
| K2-Chat | 22.47 | 24.59 | 24.61 | 24.61 | 30.58 | 21.95 | 14.44 |
| Mixtral-8x7B-Instruct-v0.1 | 22.10 | 24.76 | 24.73 | 26.36 | 34.19 | 20.52 | 11.49 |
| granite-3.1-8b-instruct | 20.83 | 22.85 | 22.92 | 22.26 | 29.48 | 19.79 | 13.09 |
| Llama-3.1-8B-Instruct | 20.50 | 24.07 | 24.52 | 26.12 | 32.82 | 20.37 | 7.22 |
| Yi-1.5-6B-Chat | 19.24 | 21.32 | 21.39 | 22.21 | 27.90 | 18.83 | 10.44 |
| Qwen2.5-1.5B-Instruct | 18.82 | 20.91 | 20.75 | 22.11 | 27.41 | 18.19 | 10.45 |
| OLMo-2-1124-13B-Instruct | 18.66 | 20.46 | 20.60 | 21.80 | 27.10 | 17.85 | 10.74 |
| gemma-2-2b-it | 18.61 | 19.91 | 19.97 | 20.50 | 26.40 | 16.95 | 12.85 |
| Qwen3-0.6B-nothink | 18.03 | 19.10 | 19.36 | 19.55 | 22.04 | 17.55 | 14.43 |
| granite-3.1-2b-instruct | 17.92 | 19.02 | 19.11 | 19.58 | 23.94 | 17.58 | 11.87 |
| Mistral-7B-Instruct-v0.3 | 17.82 | 19.64 | 19.65 | 20.09 | 26.37 | 16.64 | 10.41 |
| MAP-Neo-7B-Instruct-v0.1 | 17.05 | 18.52 | 18.42 | 18.70 | 23.26 | 16.62 | 10.95 |
| OLMo-2-1124-7B-Instruct | 16.81 | 18.08 | 18.57 | 18.85 | 22.80 | 15.82 | 11.90 |

Table 5: Continue. **Detailed Performance Overview on *SuperGPQA*.** The highest score in each column is indicated with a $\boxed{box}$; the second-best score is in **bold**, and the third-best score is underlined.

| Model | Overall (sample) | Overall (subfield) | Overall (field) | Overall (discipline) | Easy (sample) | Middle (sample) | Hard (sample) |
|---|---|---|---|---|---|---|---|
| *Base Models* | | | | | | | |
| Llama-4-Maverick-17B-128E | 37.83 | 40.95 | 40.74 | 40.34 | 46.98 | 41.28 | 22.06 |
| Qwen2.5-72B | **34.33** | **38.08** | **38.70** | **39.54** | **46.20** | **38.12** | 15.01 |
| Qwen2.5-32B | 33.16 | 36.52 | 37.33 | 38.29 | 45.12 | 36.58 | 14.34 |
| Qwen3-14B-Base | 33.02 | 35.92 | 36.18 | 35.71 | 42.75 | 36.42 | 16.70 |
| DeepSeek-V3-Base | 32.14 | 34.79 | 34.58 | 34.71 | 41.28 | 34.50 | **18.20** |
| Llama-4-Scout-17B-16E | 31.51 | 34.41 | 34.49 | 33.92 | 41.31 | 34.00 | 16.60 |
| Qwen3-30B-A3B-Base | 30.36 | 33.27 | 33.60 | 33.54 | 39.04 | 33.31 | 15.94 |
| Qwen2.5-14B | 30.19 | 33.33 | 34.14 | 34.54 | 42.27 | 31.44 | 14.85 |
| Qwen3-8B-Base | 29.02 | 31.60 | 32.05 | 31.09 | 38.44 | 31.05 | 15.30 |
| Yi-1.5-34B | 27.62 | 30.78 | 31.03 | 32.55 | 39.68 | 27.95 | 13.86 |
| Llama-3.1-70B | 27.22 | 30.52 | 31.28 | 32.55 | 40.78 | 26.95 | 12.78 |
| Qwen3-4B-Base | 26.76 | 29.02 | 29.37 | 28.96 | 36.05 | 28.11 | 14.31 |
| Qwen2.5-7B | 25.36 | 28.19 | 28.73 | 29.60 | 36.58 | 25.94 | 12.10 |
| Llama-3.1-405B | 25.23 | 28.09 | 28.33 | 30.15 | 37.58 | 25.12 | 11.86 |
| gemma-2-27b | 24.49 | 27.35 | 27.96 | 28.58 | 36.26 | 24.07 | 12.27 |
| Yi-1.5-9B | 23.10 | 25.40 | 25.69 | 26.17 | 32.52 | 22.98 | 12.96 |
| gemma-2-9b | 22.56 | 25.19 | 25.47 | 26.26 | 33.88 | 21.34 | 12.20 |
| Mixtral-8x22B-v0.1 | 22.41 | 24.71 | 25.04 | 25.02 | 32.78 | 21.67 | 12.26 |
| Mixtral-8x7B-v0.1 | 21.76 | 24.92 | 25.45 | 27.36 | 33.66 | 20.32 | 11.13 |
| K2 | 20.92 | 23.62 | 23.88 | 24.65 | 30.97 | 20.01 | 11.40 |
| Yi-1.5-6B | 20.20 | 22.10 | 22.56 | 23.44 | 28.37 | 19.64 | 12.20 |
| Qwen2.5-3B | 20.14 | 22.81 | 23.30 | 24.42 | 30.42 | 19.81 | 9.40 |
| Llama-3.1-8B | 19.93 | 22.42 | 22.77 | 23.87 | 30.33 | 18.94 | 10.16 |
| Qwen3-1.7B-Base | 19.83 | 21.62 | 22.16 | 21.86 | 27.94 | 20.02 | 10.61 |
| Mistral-7B-v0.3 | 19.48 | 21.50 | 21.81 | 22.27 | 27.62 | 18.65 | 11.96 |
| Qwen2.5-1.5B | 17.17 | 19.31 | 19.80 | 21.35 | 24.52 | 16.79 | 9.74 |
| granite-3.1-2b-base | 16.18 | 17.90 | 17.91 | 18.09 | 23.13 | 14.91 | 10.67 |
| OLMo-2-1124-13B | 16.07 | 18.75 | 19.82 | 21.37 | 27.24 | 14.41 | 6.57 |
| MAP-Neo-7B | 15.76 | 17.48 | 18.26 | 19.54 | 22.86 | 14.64 | 9.83 |
| Qwen3-0.6B-Base | 15.74 | 17.48 | 18.05 | 18.33 | 22.57 | 15.19 | 9.18 |
| granite-3.1-8b-base | 15.69 | 16.98 | 16.79 | 16.65 | 20.40 | 15.65 | 10.60 |
| OLMo-2-1124-7B | 15.15 | 17.62 | 18.30 | 19.60 | 24.43 | 13.83 | 7.15 |
| gemma-2-2b | 13.57 | 14.98 | 15.43 | 16.21 | 19.12 | 13.47 | 7.63 |
| Qwen2.5-0.5B | 10.74 | 11.88 | 12.09 | 13.25 | 14.41 | 10.78 | 6.65 |

Table 6: **Detailed Performance Overview on *SuperGPQA*- Pivot Table 2.** The table presents LLMs' performance on different disciplines. The highest score in each column is indicated with a $\boxed{box}$; the second-best score is in **bold**, and the third-best score is underlined.

| Model | Agr. | Econ. | Edu. | Eng. | Hist. | Law | Lit & Arts | Mgt. | Med. | Mil Sci. | Phil. | Sci. | Soc. |
|---|---|---|---|---|---|---|---|---|---|---|---|---|---|
| *Reasoning Models* | | | | | | | | | | | | | |
| Gemini-2.5-Pro-Exp-0325 | 55.05 | 63.12 | 51.86 | 64.47 | 66.62 | 65.24 | 61.52 | 57.49 | 58.22 | 63.90 | 66.86 | 65.56 | 63.64 |
| DeepSeek-R1 | 54.43 | 66.09 | 54.75 | 63.10 | 55.19 | 65.24 | 52.45 | 57.09 | 59.93 | 57.07 | 63.11 | 63.69 | 67.13 |
| DeepSeek-R1-Zero | 53.81 | 66.44 | 60.54 | 60.28 | 58.61 | 66.77 | 56.86 | 59.68 | 60.65 | 58.54 | 63.69 | 59.93 | 67.13 |
| Seed-Thinking-v1.5 | 52.99 | 67.70 | 56.40 | 63.53 | 42.28 | 62.65 | 44.63 | 56.49 | 61.05 | 55.61 | 65.13 | 64.08 | 59.44 |
| o1-2024-12-17 | 50.93 | 61.17 | 53.31 | 59.17 | 60.98 | 63.87 | 55.79 | 57.29 | 62.25 | 60.49 | 61.38 | 61.76 | 64.34 |
| Qwen3-32B-thinking | 49.07 | 57.73 | 50.00 | 52.10 | 31.75 | 49.24 | 36.04 | 49.10 | 51.94 | 49.76 | 54.18 | 49.88 | 55.24 |
| claude-3-7-sonnet-20250219-thinking | 47.22 | 60.25 | 55.17 | 58.38 | 57.12 | 61.43 | 53.28 | 51.50 | 53.65 | 56.59 | 60.23 | 61.06 | 64.34 |
| Qwen3-235B-A22B-thinking | 47.22 | 61.97 | 53.93 | 60.83 | 38.87 | 59.15 | 44.45 | 50.50 | 59.64 | 53.17 | 60.81 | 56.35 | 54.55 |
| QwQ-32B | 46.19 | 59.79 | 53.31 | 55.27 | 35.31 | 54.12 | 37.65 | 48.50 | 53.03 | 52.68 | 54.18 | 58.69 | 53.85 |
| Qwen3-14B-thinking | 43.30 | 53.72 | 47.52 | 53.03 | 28.64 | 44.05 | 31.44 | 45.51 | 48.68 | 46.34 | 49.28 | 52.55 | 42.66 |
| Qwen3-30B-A3B-thinking | 42.06 | 56.13 | 46.90 | 54.33 | 28.93 | 46.34 | 33.77 | 44.51 | 49.95 | 43.41 | 51.59 | 56.39 | 44.76 |
| o3-mini-2025-01-31-high | 41.03 | 54.07 | 46.28 | 56.41 | 36.80 | 45.73 | 39.44 | 51.50 | 51.72 | 46.34 | 51.01 | 61.72 | 46.15 |
| o3-mini-2025-01-31-medium | 40.62 | 51.32 | 41.74 | 53.83 | 35.01 | 43.60 | 37.11 | 48.50 | 50.34 | 45.85 | 48.13 | 58.79 | 44.06 |
| QwQ-32B-Preview | 38.14 | 47.77 | 45.25 | 43.37 | 29.53 | 45.88 | 32.82 | 41.32 | 43.88 | 40.00 | 43.52 | 46.33 | 43.36 |
| o3-mini-2025-01-31-low | 37.32 | 45.25 | 38.43 | 48.20 | 32.94 | 39.79 | 36.22 | 43.71 | 48.09 | 41.46 | 44.09 | 53.24 | 45.45 |
| Qwen3-8B-thinking | 36.49 | 48.45 | 39.67 | 46.78 | 25.96 | 37.96 | 28.70 | 42.12 | 42.83 | 44.88 | 43.80 | 46.61 | 40.56 |
| o1-mini-2024-09-12 | 34.02 | 45.02 | 36.16 | 45.41 | 26.26 | 35.98 | 32.22 | 40.52 | 44.32 | 39.51 | 39.48 | 51.09 | 41.26 |
| Qwen3-4B-thinking | 30.93 | 45.13 | 34.50 | 44.78 | 22.40 | 35.67 | 26.19 | 37.52 | 35.28 | 33.66 | 42.65 | 48.24 | 35.66 |
| Qwen3-1.7B-thinking | 25.36 | 32.53 | 29.34 | 31.98 | 18.25 | 26.07 | 19.63 | 28.34 | 26.57 | 31.71 | 34.29 | 35.32 | 25.17 |
| Qwen3-0.6B-thinking | 18.14 | 23.02 | 23.14 | 20.86 | 15.43 | 20.12 | 16.47 | 22.75 | 18.37 | 25.37 | 23.92 | 21.62 | 23.08 |
| *Chat Models* | | | | | | | | | | | | | |
| Doubao-1.5-pro-32k-250115 | 50.93 | 65.06 | 55.58 | 55.60 | 42.88 | 58.84 | 42.30 | 53.29 | 59.13 | 54.15 | 61.96 | 55.54 | 51.75 |
| Llama-4-Maverick-17B-128E-Instruct | 48.87 | 59.91 | 50.41 | 56.06 | 51.93 | 57.93 | 42.96 | 55.49 | 56.23 | 50.73 | 54.47 | 54.62 | 55.24 |
| claude-3-7-sonnet-20250219 | 47.84 | 56.82 | 53.51 | 51.67 | 57.86 | 58.84 | 51.55 | 52.50 | 53.28 | 58.54 | 57.93 | 50.82 | 65.03 |
| Doubao-1.5-pro-32k-241225 | 47.42 | 60.14 | 54.75 | 50.76 | 37.54 | 54.12 | 38.31 | 52.69 | 54.99 | 53.66 | 57.06 | 51.57 | 53.15 |
| gpt-4.1-2025-04-14 | 47.22 | 55.78 | 54.55 | 51.93 | 55.79 | 58.08 | 51.37 | 53.09 | 57.17 | 56.10 | 58.21 | 61.54 | |
| claude-3-5-sonnet-20241022 | 47.01 | 56.59 | 53.72 | 47.57 | 53.56 | 60.21 | 50.42 | 51.30 | 49.26 | 59.51 | 52.45 | 45.02 | 64.34 |
| Qwen3-235B-A22B-nothink | 46.60 | 52.35 | 49.38 | 49.82 | 35.01 | 50.61 | 39.68 | 47.11 | 52.20 | 50.73 | 53.03 | 50.35 | |
| grok-3-beta | 46.19 | 51.78 | 48.14 | 53.85 | 55.34 | 55.18 | 49.76 | 51.30 | 51.51 | 56.59 | 51.01 | 55.35 | 55.94 |
| Qwen-Max-2025-01-25 | 44.33 | 57.50 | 56.40 | 50.81 | 44.81 | 54.12 | 44.93 | 54.69 | 56.37 | 49.27 | 56.20 | 47.51 | 54.55 |
| Llama-3.1-405B-Instruct | 43.09 | 49.71 | 45.45 | 41.89 | 50.00 | 55.34 | 43.20 | 47.70 | 49.04 | 52.20 | 48.41 | 39.80 | 49.65 |
| gpt-4o-2024-11-20 | 42.27 | 46.74 | 50.00 | 42.83 | 52.52 | 53.81 | 46.72 | 47.31 | 52.52 | 52.20 | 52.74 | 40.67 | 54.55 |
| gpt-4.1-mini-2025-04-14 | 42.06 | 51.66 | 43.80 | 48.43 | 39.47 | 45.27 | 38.01 | 46.71 | 46.34 | 48.29 | 44.96 | 52.46 | 53.85 |
| gpt-4o-2024-05-13 | 41.44 | 40.78 | 44.21 | 37.47 | 50.74 | 52.90 | 45.11 | 41.32 | 48.82 | 50.73 | 45.24 | 35.41 | 53.85 |
| DeepSeek-V3 | 41.24 | 49.48 | 43.80 | 47.21 | 47.18 | 51.07 | 45.23 | 48.50 | 46.10 | 48.29 | 43.23 | 48.30 | 55.54 |
| Qwen2.5-72B-Instruct | 41.24 | 46.62 | 44.42 | 41.12 | 30.71 | 45.88 | 34.90 | 42.32 | 45.74 | 45.37 | 43.52 | 39.33 | 46.15 |
| Qwen3-32B-nothink | 40.21 | 51.09 | 48.55 | 45.82 | 29.82 | 44.51 | 34.01 | 45.71 | 47.04 | 47.80 | 47.84 | 45.43 | 46.85 |
| MiniMax-Text-01 | 39.79 | 49.60 | 47.52 | 44.88 | 45.25 | 54.27 | 43.02 | 48.90 | 47.08 | 48.29 | 45.53 | 43.81 | 53.85 |
| gpt-4o-2024-08-06 | 39.59 | 40.78 | 44.42 | 38.84 | 50.89 | 54.12 | 46.30 | 45.51 | 50.49 | 50.73 | 47.84 | 38.41 | 53.85 |
| Mistral-Large-Instruct-2411 | 38.76 | 42.27 | 42.15 | 39.66 | 44.96 | 47.87 | 41.11 | 43.31 | 43.01 | 48.78 | 42.07 | 39.24 | 50.35 |
| gemini-2.0-flash | 38.56 | 45.93 | 42.36 | 48.37 | 45.25 | 49.24 | 43.68 | 41.32 | 43.77 | 51.22 | 45.53 | 50.20 | 53.85 |
| Qwen3-30B-A3B-nothink | 37.94 | 41.70 | 46.07 | 42.42 | 26.41 | 41.31 | 32.46 | 41.52 | 43.41 | 47.32 | 45.53 | 43.37 | 38.46 |
| Qwen3-14B-nothink | 37.94 | 43.64 | 43.18 | 39.45 | 26.41 | 38.41 | 31.26 | 40.92 | 41.67 | 45.85 | 43.52 | 40.70 | 39.86 |
| Llama-4-Scout-17B-16E-Instruct | 37.73 | 49.60 | 45.04 | 45.78 | 32.05 | 45.73 | 36.04 | 44.51 | 42.22 | 48.78 | 43.23 | 45.68 | 46.15 |
| Llama-3.1-70B-Instruct | 37.11 | 41.12 | 41.94 | 33.88 | 33.09 | 45.88 | 35.08 | 46.11 | 44.21 | 47.80 | 42.94 | 30.04 | 48.25 |
| Llama-3.3-70B-Instruct | 36.70 | 40.55 | 40.29 | 36.44 | 41.25 | 44.66 | 38.54 | 45.51 | 43.77 | 46.34 | 42.94 | 34.96 | 42.66 |
| Qwen2.5-32B-Instruct | 36.49 | 43.07 | 45.45 | 38.93 | 30.71 | 41.92 | 32.76 | 39.52 | 42.21 | 44.88 | 39.19 | 38.24 | 39.16 |
| Qwen-Max-2024-09-19 | 36.49 | 45.02 | 45.66 | 39.84 | 35.16 | 44.05 | 39.08 | 45.11 | 43.41 | 47.32 | 43.23 | 38.24 | 38.46 |
| Qwen2.5-14B-Instruct | 36.08 | 37.69 | 39.26 | 35.87 | 26.41 | 37.04 | 31.44 | 41.52 | 36.94 | 38.05 | 38.04 | 34.20 | 36.36 |
| Yi-Lighting | 33.81 | 39.18 | 35.95 | 32.53 | 36.05 | 37.35 | 36.34 | 38.12 | 36.95 | 42.44 | 37.75 | 30.85 | 42.66 |
| phi-4 | 32.78 | 40.21 | 39.26 | 37.27 | 30.27 | 41.46 | 31.62 | 39.12 | 37.79 | 39.02 | 40.63 | 38.87 | 41.26 |
| gpt-4.1-nano-2025-04-14 | 31.55 | 34.25 | 36.16 | 32.30 | 26.41 | 31.71 | 27.33 | 35.33 | 35.97 | 33.17 | 31.99 | 35.88 | 32.17 |
| gemma-2-27b-it | 31.13 | 29.32 | 30.99 | 26.72 | 24.78 | 35.06 | 29.42 | 34.93 | 31.00 | 39.02 | 35.45 | 24.81 | 34.27 |
| Qwen3-8B-nothink | 31.13 | 38.49 | 37.40 | 37.23 | 23.15 | 33.99 | 27.68 | 35.53 | 36.91 | 41.95 | 38.04 | 38.37 | 39.86 |
| Qwen2.5-7B-Instruct | 29.28 | 34.59 | 35.33 | 28.38 | 22.85 | 34.22 | 24.28 | 33.33 | 32.60 | 32.68 | 34.58 | 27.54 | 30.07 |
| Mistral-Small-Instruct-2409 | 28.87 | 26.23 | 26.86 | 25.25 | 25.96 | 30.95 | 27.57 | 30.54 | 28.53 | 34.15 | 29.68 | 24.17 | 32.87 |
| Yi-1.5-34B-Chat | 28.87 | 31.27 | 30.99 | 25.67 | 23.00 | 34.45 | 27.15 | 31.54 | 29.73 | 33.17 | 30.26 | 23.27 | 28.67 |
| Qwen3-4B-nothink | 28.66 | 33.68 | 30.99 | 31.64 | 21.22 | 31.25 | 24.22 | 33.53 | 31.14 | 36.10 | 35.16 | 33.39 | 30.77 |
| Mixtral-8x22B-Instruct-v0.1 | 28.25 | 33.45 | 33.68 | 29.02 | 30.86 | 34.60 | 30.55 | 35.13 | 30.53 | 44.39 | 32.85 | 26.95 | 36.36 |
| Mixtral-8x7B-Instruct-v0.1 | 27.22 | 24.51 | 28.31 | 21.16 | 25.37 | 30.79 | 26.85 | 26.15 | 26.13 | 30.73 | 25.94 | 18.70 | 30.77 |
| gemma-2-9b-it | 27.01 | 27.15 | 32.64 | 23.24 | 22.26 | 28.51 | 27.45 | 32.14 | 27.44 | 34.15 | 29.39 | 21.26 | 27.97 |
| Qwen2.5-3B-Instruct | 25.36 | 26.35 | 30.99 | 22.83 | 18.55 | 25.61 | 22.32 | 30.94 | 26.82 | 27.32 | 27.09 | 21.64 | 26.57 |
| granite-3.1-8b-instruct | 24.74 | 20.16 | 23.76 | 21.60 | 16.32 | 24.70 | 20.53 | 25.15 | 20.58 | 27.80 | 21.33 | 19.70 | 23.08 |
| Llama-3.1-8B-Instruct | 24.74 | 24.40 | 29.55 | 19.07 | 20.03 | 26.68 | 20.94 | 34.13 | 30.20 | 34.63 | 27.67 | 16.08 | 31.47 |
| gemma-2-2b-it | 24.33 | 21.31 | 19.42 | 18.30 | 17.06 | 21.80 | 19.21 | 21.56 | 19.27 | 26.83 | 19.60 | 17.53 | 20.28 |
| Qwen3-1.7B-nothink | 24.12 | 24.51 | 25.00 | 23.29 | 17.06 | 22.26 | 19.39 | 27.74 | 22.25 | 31.71 | 31.12 | 23.52 | 22.38 |
| Yi-1.5-9B-Chat | 24.12 | 29.90 | 30.99 | 23.04 | 16.91 | 28.20 | 20.70 | 31.94 | 25.74 | 28.29 | 27.09 | 21.23 | 30.77 |
| K2-Chat | 22.68 | 23.37 | 27.27 | 23.10 | 16.02 | 30.49 | 23.39 | 28.74 | 23.99 | 28.29 | 27.09 | 20.32 | 25.17 |
| Qwen2.5-1.5B-Instruct | 22.27 | 22.57 | 27.17 | 17.64 | 16.47 | 26.37 | 17.84 | 24.95 | 22.58 | 25.37 | 27.67 | 16.85 | 19.58 |
| OLMo-2-1124-13B-Instruct | 20.82 | 22.68 | 25.21 | 18.18 | 16.91 | 22.10 | 19.57 | 21.96 | 21.81 | 25.85 | 27.38 | 16.39 | 24.48 |
| OLMo-2-1124-7B-Instruct | 20.82 | 14.66 | 20.25 | 16.52 | 16.32 | 18.45 | 18.08 | 22.95 | 17.82 | 23.41 | 21.90 | 15.64 | 18.18 |
| Qwen3-0.6B-nothink | 20.41 | 20.27 | 21.07 | 18.93 | 13.65 | 20.43 | 16.71 | 16.97 | 16.73 | 25.85 | 22.77 | 17.22 | 23.08 |
| Yi-1.5-6B-Chat | 20.41 | 23.83 | 25.62 | 18.58 | 13.95 | 25.61 | 18.79 | 24.15 | 21.89 | 29.27 | 25.36 | 17.60 | 23.78 |
| Mistral-7B-Instruct-v0.3 | 20.00 | 17.75 | 18.18 | 17.28 | 15.88 | 21.19 | 21.12 | 23.55 | 20.04 | 26.83 | 22.19 | 16.18 | 20.98 |
| granite-3.1-2b-instruct | 18.14 | 19.93 | 19.83 | 17.64 | 16.77 | 23.02 | 19.03 | 23.15 | 17.46 | 22.44 | 19.02 | 17.70 | 20.98 |
| MAP-Neo-7B-Instruct-v0.1 | 17.94 | 17.64 | 21.69 | 16.79 | 13.50 | 21.19 | 16.35 | 20.96 | 18.91 | 25.85 | 21.61 | 15.99 | 14.69 |
| Qwen2.5-0.5B-Instruct | 16.91 | 16.49 | 17.98 | 9.77 | 10.09 | 13.87 | 12.89 | 15.97 | 14.12 | 13.17 | 12.68 | 8.55 | 12.59 |

Table 6: Continue. **Detailed Performance Overview on *SuperGPQA*- Pivot Table 2.** The table presents LLMs' performance on different disciplines. The highest score in each column is indicated with a $\boxed{box}$; the second-best score is in **bold**, and the third-best score is underlined.

| Model | Agr. | Econ. | Edu. | Eng. | Hist. | Law | Lit & Arts | Mgt. | Med. | Mil Sci. | Phil. | Sci. | Soc. |
|---|---|---|---|---|---|---|---|---|---|---|---|---|---|
| *Base Models* | | | | | | | | | | | | | |
| Qwen2.5-32B | $\boxed{38.76}$ | 40.89 | **45.87** | 32.70 | 28.49 | 39.94 | 30.91 | 40.72 | 38.84 | 47.80 | 43.52 | 29.45 | $\boxed{39.86}$ |
| Qwen2.5-72B | 36.29 | $\boxed{44.33}$ | 45.25 | 33.39 | 31.31 | 44.21 | **35.50** | 43.51 | $\boxed{43.12}$ | 46.34 | 42.94 | 29.38 | 38.46 |
| Llama-4-Maverick-17B-128E | 35.67 | **43.18** | $\boxed{46.69}$ | **38.29** | 29.67 | **42.53** | 35.80 | **44.91** | 42.11 | **44.88** | 45.53 | **35.24** | **39.86** |
| Qwen3-14B-Base | 35.05 | 39.06 | 39.46 | $\boxed{33.48}$ | 24.33 | 35.06 | 28.40 | 40.52 | 37.53 | 42.44 | 38.90 | **30.82** | 39.16 |
| Qwen2.5-14B | 34.02 | 34.82 | 40.08 | 30.26 | 28.49 | 36.89 | 28.94 | 40.32 | 34.37 | 40.00 | 37.75 | 26.68 | 36.36 |
| Qwen3-8B-Base | 32.99 | 31.62 | 33.26 | 29.70 | 20.92 | 30.03 | 25.18 | 34.13 | 32.12 | 37.07 | 35.45 | 27.38 | 34.27 |
| Qwen3-30B-A3B-Base | 31.55 | 38.26 | 42.15 | 30.28 | 24.33 | 33.38 | 27.63 | 36.33 | 37.17 | 35.61 | 37.18 | 27.14 | 34.97 |
| Llama-4-Scout-17B-16E | 30.31 | 37.69 | 39.26 | 32.05 | 25.96 | 36.59 | 30.73 | 36.53 | 35.21 | 40.00 | 38.62 | 28.68 | 29.37 |
| Llama-3.1-405B | 29.28 | 27.95 | 31.20 | 23.61 | $\boxed{34.12}$ | 34.30 | 30.79 | 34.13 | 29.47 | 39.51 | 31.70 | 21.47 | 24.48 |
| Llama-3.1-70B | 28.87 | 29.67 | 36.36 | 26.19 | **32.94** | 33.38 | 32.64 | 36.73 | 30.96 | 40.98 | 36.89 | 23.30 | 34.27 |
| gemma-2-27b | 28.87 | 27.26 | 30.79 | 24.09 | 22.85 | 28.96 | 27.51 | 32.73 | 27.80 | 37.56 | 31.70 | 21.38 | 30.07 |
| DeepSeek-V3-Base | 28.66 | 35.17 | 39.05 | 31.87 | 30.12 | 37.20 | 33.65 | 37.33 | 34.70 | 37.07 | 38.62 | 30.08 | 37.76 |
| Qwen2.5-7B | 28.25 | 31.96 | 38.02 | 24.89 | 22.55 | 30.34 | 25.06 | 28.54 | 30.56 | 31.71 | 36.60 | 22.04 | 34.27 |
| Yi-1.5-34B | 28.25 | 35.85 | 37.60 | 27.32 | 24.63 | 34.45 | 30.25 | 36.13 | 31.18 | 40.49 | 35.45 | 23.80 | 37.76 |
| Mixtral-8x7B-v0.1 | 28.04 | 24.97 | 29.34 | 20.67 | 23.89 | 30.79 | 27.27 | 30.74 | 25.05 | 37.56 | 30.84 | 17.87 | 28.67 |
| Qwen3-4B-Base | 27.84 | 30.81 | 35.12 | 27.36 | 21.81 | 28.05 | 21.72 | 32.34 | 30.05 | 32.68 | 30.84 | 24.99 | 32.87 |
| gemma-2-9b | 27.22 | 26.23 | 30.99 | 21.79 | 21.81 | 26.83 | 23.81 | 28.94 | 26.06 | 32.68 | 27.09 | 20.01 | 27.97 |
| Mistral-7B-v0.3 | 25.98 | 20.96 | 25.62 | 19.02 | 14.84 | 24.24 | 20.11 | 23.55 | 22.61 | 30.24 | 25.94 | 17.47 | 18.88 |
| Yi-1.5-9B | 25.77 | 26.12 | 34.71 | 23.59 | 18.69 | 25.15 | 23.21 | 30.54 | 23.85 | 31.22 | 28.53 | 20.86 | 27.97 |
| Mixtral-8x22B-v0.1 | 23.92 | 23.94 | 27.27 | 22.02 | 20.33 | 27.59 | 24.64 | 30.14 | 22.50 | 28.78 | 24.50 | 20.96 | 28.67 |
| Llama-3.1-8B | 23.30 | 22.22 | 25.00 | 19.56 | 20.92 | 26.98 | 21.48 | 25.95 | 24.54 | 33.17 | 26.80 | 16.62 | 23.78 |
| Qwen3-1.7B-Base | 23.09 | 22.57 | 25.41 | 20.84 | 15.43 | 19.36 | 16.65 | 22.75 | 21.60 | 29.27 | 26.80 | 18.08 | 22.38 |
| K2 | 22.68 | 20.62 | 28.93 | 20.50 | 17.80 | 28.35 | 24.40 | 27.74 | 22.94 | 33.17 | 29.11 | 18.39 | 25.87 |
| Qwen2.5-3B | 22.47 | 23.83 | 28.31 | 19.74 | 18.40 | 26.37 | 21.60 | 28.14 | 25.41 | 30.73 | 29.39 | 16.54 | 26.57 |
| Yi-1.5-6B | 22.06 | 25.89 | 27.07 | 19.93 | 14.99 | 25.46 | 21.36 | 30.54 | 21.92 | 30.73 | 26.80 | 17.98 | 23.78 |
| MAP-Neo-7B | 20.21 | 18.10 | 23.35 | 15.43 | 15.28 | 21.34 | 18.79 | 21.76 | 17.10 | 24.88 | 22.19 | 13.16 | 22.38 |
| Qwen3-0.6B-Base | 20.00 | 18.67 | 24.38 | 15.78 | 11.42 | 17.68 | 16.65 | 20.56 | 16.19 | 24.88 | 21.90 | 14.05 | 16.08 |
| OLMo-2-1124-13B | 19.59 | 20.27 | 26.24 | 15.21 | 19.73 | 22.41 | 20.88 | 26.15 | 19.96 | 25.85 | 26.51 | 11.93 | 23.08 |
| Qwen2.5-1.5B | 19.18 | 20.73 | 28.10 | 16.50 | 15.43 | 21.95 | 17.72 | 23.35 | 20.87 | 24.88 | 25.65 | 14.48 | 28.67 |
| granite-3.1-2b-base | 18.76 | 18.33 | 23.55 | 16.37 | 13.80 | 21.15 | 16.41 | 18.96 | 17.39 | 21.46 | 22.48 | 14.48 | 13.99 |
| gemma-2-2b | 18.35 | 16.15 | 19.21 | 12.82 | 11.72 | 19.21 | 15.75 | 18.36 | 16.52 | 19.02 | 19.60 | 11.41 | 12.59 |
| OLMo-2-1124-7B | 18.35 | 17.30 | 25.00 | 14.88 | 19.14 | 19.36 | 17.36 | 20.76 | 18.73 | 22.44 | 22.48 | 11.72 | 27.27 |
| granite-3.1-8b-base | 15.46 | 17.87 | 15.08 | 16.26 | 11.72 | 18.29 | 13.84 | 14.57 | 14.81 | 20.49 | 20.17 | 15.44 | 22.38 |
| Qwen2.5-0.5B | 15.05 | 12.37 | 16.32 | 10.30 | 11.42 | 12.96 | 14.92 | 11.78 | 12.23 | 16.59 | 15.56 | 8.74 | 13.99 |

In addition to the models discussed in the subsection 4.2, Table 5 and Table 6 provide a more comprehensive analysis of baseline performances.

**Newer versions Lead to Better Results.** We observe the incremental growing of qwen-max series (2024-09-19→2025-01-25: 39.96 →50.08) and GPT-4o series (2024-05-13 → 2024-08-06 → 2024-11-20: 39.76 → 41.64 → 44.40). Such incremental performances following the chronological order suggest that the developers of proprietary models highly value the incorporation of long-tailed knowledge. Moreover, we conjecture that the LLMs from Chinese firms (e.g., Qwen and Doubao) generally show superior performance as their data collection pipelines are more aligned with *SuperGPQA*, i.e., a considerable ratio of the references are translated from Chinese textbooks.

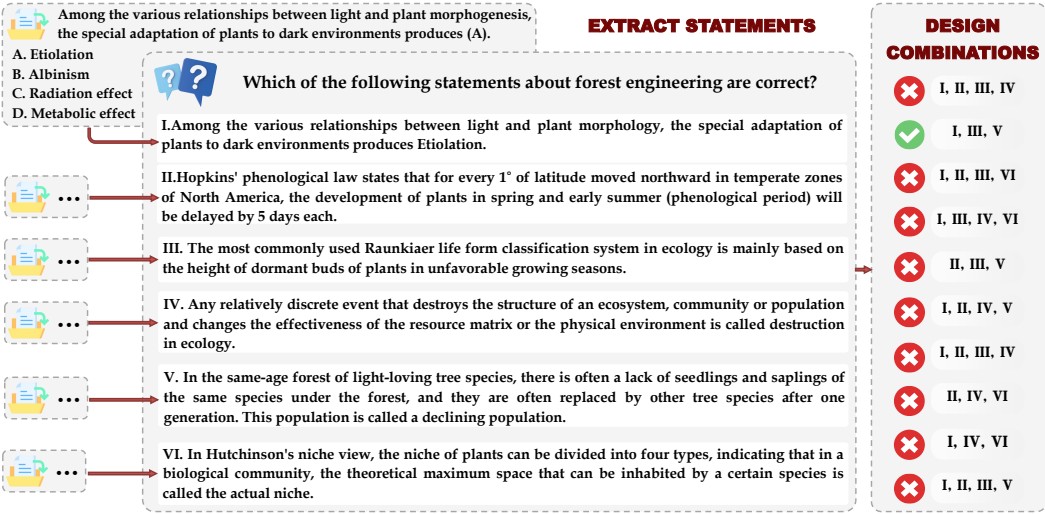

Figure 7: Rewriting Samples of Questions Requiring the Selection of Correct or Incorrect Options.

**Analysis of Disciplinary Discrimination Power**    In Table 7, to assess disciplinary discrimination power, we evaluate model performance distributions through three key metrics: Mean Accuracy reflecting overall discipline difficulty, Standard Deviation (SD) measuring performance dispersion, and Coefficient of Variation (CV) enabling cross-disciplinary comparison through mean-normalized variation. The discrimination analysis complements this by quantifying performance extremes via High-Low Group Difference ($\Delta$), calculated as the mean accuracy gap between top 3 and bottom 3 models in each discipline. In Table 7, we identify two distinct disciplinary discrimination patterns. First, High-discrimination disciplines including History (SD=8.45, CV=0.175, $\Delta$=19.19) and Law (SD=7.17, CV=0.126, $\Delta$=16.62) exhibit strong performance differentiation, suggesting significant model capability variance in handling their complex requirements. This likely stems from context-dependent reasoning demanding real-world knowledge synthesis, cultural nuance interpretation, and ethical judgment variations in open-ended scenarios. Second, Low-discrimination disciplines such as Military Science (SD=4.99, CV=0.093, $\Delta$=11.55), Engineering (SD=5.75, CV=0.107, $\Delta$=13.13), and Management (SD=5.21, CV=0.099, $\Delta$=10.98) show performance convergence among top models. Our results emphasize the critical need for comprehensive cross-domain evaluations to fully capture models' heterogeneous capabilities beyond technical domains.

Table 7: Discrimination Analysis with Key Evaluation Metrics (Mean Acc.: Mean Accuracy, SD: Standard Deviation, CV: Coefficient of Variation, $\Delta$: High-Low Group Difference).

| Discipline | Descriptive Statistics | | | Discrimination Indices Analysis | | |
|---|---|---|---|---|---|---|
| | Mean Acc. | SD | CV | High | Low | $\Delta$ |
| Engineering | 53.93 | 5.75 | 0.107 | 60.85 | 47.72 | 13.13 |
| Philosophy | 55.56 | 7.33 | 0.132 | 62.92 | 46.59 | 16.33 |
| Medicine | 54.42 | 6.44 | 0.118 | 60.94 | 46.38 | 14.57 |
| Economics | 58.25 | 6.96 | 0.120 | 65.86 | 49.83 | 16.04 |
| Science | 54.52 | 6.86 | 0.126 | 62.39 | 46.94 | 15.45 |
| Law | 56.92 | 7.17 | 0.126 | 65.29 | 48.68 | 16.62 |
| History | 48.28 | 8.45 | 0.175 | 58.26 | 39.07 | 19.19 |
| Education | 52.15 | 5.94 | 0.114 | 57.51 | 44.15 | 13.36 |
| Military Science | 53.85 | 4.99 | 0.093 | 59.51 | 47.97 | 11.55 |
| Management | 52.73 | 5.21 | 0.099 | 58.02 | 47.04 | 10.98 |
| Literature & Arts | 46.94 | 6.58 | 0.140 | 55.03 | 40.02 | 15.02 |
| Agronomy | 46.97 | 5.58 | 0.119 | 53.06 | 40.27 | 12.78 |
| Sociology | 57.83 | 7.33 | 0.127 | 66.20 | 50.35 | 15.85 |

# E    Related Work

## E.1    Large Language Models

The landscape of natural language processing has been transformed by recent breakthroughs in Large Language Models (LLMs) [Zhang et al., 2024a, Young et al., 2024]. The introduction of GPT-3 marked a significant milestone, showcasing its ability to interpret tasks and examples from textual inputs with minimal prior training. Recently, the latest generation of LLMs (e.g., GPT-4 [OpenAI, 2023], Claude-3.5 [3], Gemini [Team, 2023], and Llama-3 [AI@Meta, 2024]), have exhibited remarkable progress in sophisticated reasoning across diverse fields. To comprehensively evaluate and challenge the expanding capabilities of these advanced AI systems, we present *SuperGPQA*. This novel benchmark is specifically crafted to probe the knowledge boundaries of existing LLMs.

## E.2    LLM Benchmarks

Recently, the development of the Large Language Model (LLM) has been transformed by the introduction of various benchmarks [Cobbe et al., 2021, Wang et al., 2024a, Bai et al., 2024]. Notable examples include GLUE [Wang et al., 2019b] and its successor SuperGLUE [Wang et al., 2019a], which have been instrumental in propelling advancements in language comprehension tasks. These foundational benchmarks paved the way for more specialized assessments, such as

---
[3] https://www.anthropic.com/news/claude-3-5-sonnet

MMLU [Hendrycks et al., 2020], HotpotQA [Yang et al., 2018], BigBench [Srivastava et al., 2022], HellaSwag [Zellers et al., 2019], CommonsenseQA [Talmor et al., 2019], KOR-Bench [Ma et al., 2024], SimpleQA [Wei et al., 2024a] and Chinese SimpleQA [He et al., 2024a]. These newer benchmarks have expanded the evaluation scope to encompass content generation, knowledge understanding, and complex reasoning abilities. While numerous benchmarks have been developed to evaluate LLMs' capabilities and alignment with human values, these have often focused narrowly on performance within singular tasks or domains. To enable a more comprehensive LLM assessment, we propose *SuperGPQA* to scale the LLM evaluation to 285 graduate-level disciplines and provide a comprehensive and fine-grained analysis of foundation models.

# F Difficulty-Stratified Samples

> ### ✿ Easy Sample
>
> **Question:**
>
> Which of the following statements about dance studies are correct?
>
> 1. According to Danto's definition, context is an art world with modern aspects.
> 2. "La Bayadère" is a ballet created during the French July Revolution.
> 3. The ballet "Sylvia" is a dance drama created during the Paris Commune period in 1871.
> 4. Korean court dance, when calculated according to temporal principles, does not belong to secondary civilization.
>
> **Options:**
>
> A) 1, 3
>
> B) 1, 4
>
> C) 1,2,4
>
> D) 2,3
>
> E) 1,2,3
>
> F) 3
>
> G) 4
>
> H) 3,4
>
> I) 1,2,3,4
>
> J) 2,4
>
> **Answer:** 3, 4
> **Answer letter:** H
> **Discipline:** Literature and Arts
> **Field:** Art Studies
> **Subfield:** Dance Studies
> **Difficulty:** easy

> ### ⊕ Middle Sample
>
> **Question:**
>
> A deck of playing cards has 52 cards, and each shuffle changes the order of the cards, which is a permutation. If each shuffle strictly follows the operation below: first divide the cards into two equal parts, then interlace the cards from each part alternately, so that the first and last cards remain in their original positions while all other cards are rearranged. Express this permutation as a product of disjoint cycles, write out the cyclic structure of this permutation, and determine the minimum number of shuffles needed to restore the deck to its original order.
>
> **Options:**
>
> A) $(1^2, 2, 8^6), 8$
>
> B) $(1^2, 5, 5^6), 6$
>
> C) $(1^3, 1, 8^5), 15$

## 🏆 Hard Sample

**Question:**

A certain transmitter has a transmission power of 10 W, a carrier frequency of 900 MHz, a transmission antenna gain $G_\text{T} = 2$, and a receiving antenna gain $G_\text{R} = 3$. Calculate the output power of the receiver and the path loss at a distance of 10 km from the transmitter in free space.

**Options:**

A) $P_\text{R} = R_\text{T} G_\text{T} G_\text{R} \left(\frac{\lambda}{3\pi d}\right)^2 = 2.55 \times 10^{-12}$ W
   $L = \frac{P_\text{T}}{P_\text{R}} = 5.19 \times 10^{11}$ ( BII 142. 43 dB)

B) $P_\text{R} = R_\text{T} G_\text{T} G_\text{R} \left(\frac{\lambda}{4\pi d}\right)^2 = 5.07 \times 10^{-11}$ W
   $L = \frac{P_\text{T}}{P_\text{R}} = 1.97 \times 10^{10}$ ( BII 101. 95 dB)

C) $P_\text{R} = R_\text{T} G_\text{T} G_\text{R} \left(\frac{\lambda}{4\pi d}\right)^2 = 1.74 \times 10^{-10}$ W
   $L = \frac{P_\text{T}}{P_\text{R}} = 4.81 \times 10^{9}$ ( BII 98. 48 dB)

D) $P_\text{R} = R_\text{T} G_\text{T} G_\text{R} \left(\frac{\lambda}{5\pi d}\right)^2 = 8.40 \times 10^{-10}$ W
   $L = \frac{P_\text{T}}{P_\text{R}} = 3.98 \times 10^{9}$ ( BII 96. 07 dB)

E) $P_\text{R} = R_\text{T} G_\text{T} G_\text{R} \left(\frac{\lambda}{4\pi d}\right)^2 = 2.93 \times 10^{-10}$ W
   $L = \frac{P_\text{T}}{P_\text{R}} = 6.29 \times 10^{9}$ ( BII 99. 90 dB)

F) $P_\text{R} = R_\text{T} G_\text{T} G_\text{R} \left(\frac{\lambda}{3\pi d}\right)^2 = 6.83 \times 10^{-10}$ W
   $L = \frac{P_\text{T}}{P_\text{R}} = 1.46 \times 10^{10}$ ( BII 89. 35 dB)

G) $P_\text{R} = R_\text{T} G_\text{T} G_\text{R} \left(\frac{\lambda}{4\pi d}\right)^3 = 9.11 \times 10^{-11}$ W
   $L = \frac{P_\text{T}}{P_\text{R}} = 3.64 \times 10^{10}$ ( BII 104. 78 dB)

H) $P_\text{R} = R_\text{T} G_\text{T} G_\text{R} \left(\frac{\lambda}{4\pi d}\right)^1 = 3.12 \times 10^{-09}$ W
   $L = \frac{P_\text{T}}{P_\text{R}} = 9.24 \times 10^{9}$ ( BII 99. 63 dB)

I) $P_\text{R} = R_\text{T} G_\text{T} G_\text{R} \left(\frac{\lambda}{4\pi d}\right)^2 = 4.22 \times 10^{-10}$ W
   $L = \frac{P_\text{T}}{P_\text{R}} = 2.37 \times 10^{10}$ ( BII 103. 75 dB)

J) $P_\text{R} = R_\text{T} G_\text{T} G_\text{R} \left(\frac{\lambda}{6\pi d}\right)^2 = 7.95 \times 10^{-11}$ W
   $L = \frac{P_\text{T}}{P_\text{R}} = 2.50 \times 10^{10}$ ( BII 102. 20 dB)

**Answer:**

$$P_\text{R} = R_\text{T} G_\text{T} G_\text{R} \left(\frac{\lambda}{4\pi d}\right)^2 = 4.22 \times 10^{-10} \text{ W}$$

$$L = \frac{P_\text{T}}{P_\text{R}} = 2.37 \times 10^{10} \text{ ( BII 103. 75 dB)}$$

**Answer letter:** I

**Discipline:** Engineering
**Field:** Information and Communication Engineering
**Subfield:** Communication Principles
**Difficulty:** hard

# G  Annotation Tutorial

## G.1  Material Requirements

The specific requirements for material selection are as follows:

- Ensure the selected materials are accurate, with correct answers.
- Ensure the materials cover a variety of knowledge points and are free from regional bias.
- The selected materials should be correctly stated and involve a certain level of academic reasoning, with a difficulty appropriate for graduate-level study.
- Avoid using materials that rely on images as conditions.
- Verify that the materials comply with copyright requirements.

## G.2  Annotation Methods

During the annotation process, the following four methods are primarily used: **1) Original Transcription Method**, **2) Non-Choice Conversion Method**, and **3) Statement Combination Method**.

### G.2.1  Original Transcription Method

> 🔧 **Original Transcription**
>
> **Description:**
> - Directly transcribe the original multiple-choice questions, ensuring the question maintains its original meaning and correctness. Additional distractors are included to increase the difficulty of the question or enhance its discriminative power.
>
> **Operations:**
> 1. **Material Review:**
>    - Ensure that the materials meet the aforementioned requirements, including accuracy, diversity, neutrality (free from regional bias), non-image-based content, and compliance with copyright regulations.
> 2. **Question Transcription:**
>    - Use OCR tools to recognize the text in the materials or directly paste the original content. Transcribe the question content word-for-word, ensuring no key information is omitted, with particular attention to the accurate transcription of formulas.
> 3. **Option Transcription:**
>    - Use OCR tools to recognize the text in the materials or directly paste the original content. Transcribe the options word-for-word, ensuring no key information is omitted, with particular attention to the accurate transcription of formulas.
> 4. **Answer Identification:**
>    - Clearly mark the correct answer within the list of options, ensuring its accuracy and uniqueness.
> 5. **Distractor Addition:**
>    - Add distractors while maintaining the quality of the options (avoiding meaningless or excess correct answers) until the annotator deems the level of confusion sufficient. The number of options should be between 4 and 10, with more options being preferred to increase the difficulty and discriminatory power of the question.
> 6. **Field Completion:**

- Complete the category fields ("discipline","field","subfield") and select the appropriate difficulty level to ensure the integrity and accuracy of the question information.

### G.2.2  Non-Choice Conversion Method

> 🔧 **Non-Choice Conversion**
>
> **Description:**
> - Convert non-multiple-choice questions (such as calculation questions, fill-in-the-blank questions, etc.) into multiple-choice format, ensuring that the conditions of the question are complete and the final result is accurate while generating reasonable distractors based on the analysis steps or calculation process.
>
> **Operations:**
>
> 1. **Material Review:**
>    - Ensure that the materials meet the aforementioned requirements, including accuracy, diversity, neutrality (free from regional bias), non-image-based content, and compliance with copyright regulations.
>    - Ensure that the conditions of the question are complete and do not cause ambiguity. Add supplementary explanations if necessary.
>    - Ensure that the answer does not contain any essential explanatory process that is unsuitable for adaptation or transcription.
>
> 2. **Question Transcription:**
>    - Use OCR tools to recognize the text in the material or directly paste the original text. Transcribe the content of the original question word by word, ensuring that all numbers, formulas, or condition information in the question are accurate and complete to avoid ambiguity or incorrect answers.
>    - Add supplementary information if the question requires prior conditions from other questions.
>
> 3. **Answer Transcription:**
>    - Identify and select the correct result for the appropriate question based on the answer analysis process. item Use OCR tools to recognize results in the material and confirm that all numerical and formula information in the answer is accurate.
>
> 4. **Distractor Addition:**
>    - On the basis of the correct answer, consider setting common calculation errors (such as rounding errors, sign mistakes, digit errors, etc.) as distractors.
>    - For numerical results, the setting of distractors should consider reasonable ranges of calculation errors to ensure differentiation between options.
>
> 5. **Answer Identification:**
>    - Clearly mark the correct answer within the list of options, ensuring its accuracy and uniqueness.
>
> 6. **Field Completion:**
>    - Complete the category fields ("discipline","field","subfield") and select the appropriate difficulty level to ensure the integrity and accuracy of the question information.

### G.2.3  Statement Combination Method

> 🔧 **Statement Combination**
>
> **Description:**
> - By integrating stated expressions related to multiple concepts or knowledge points, a multi-level, multi-perspective multiple-choice question is constructed to increase the comprehensiveness and depth of examination of the topic.

**Operations:**

1. **Statement Extraction:**

   - Extract core concepts, definitions, relationships between concepts, application cases, and common misconceptions from textbooks and related learning resources, ensuring that the statements are representative and comprehensive.

   - Extract important statements from multiple-choice questions, covering multiple knowledge points to avoid being limited to a single definition or formula, thus enhancing the overall comprehensiveness of the questions.

   - Ensure that the extracted statements are accurate and concise, avoiding redundancy or vague expressions. The content should cover both correct and incorrect situations to provide a foundation for subsequent adaptation.

2. **Statement Adaptation:**

   - Adapt the extracted statements to include both correct and incorrect versions.

   - Incorrect statements should be somewhat misleading, avoiding obvious or easily dismissible errors.

   - Number the adapted statements (e.g., I, II, III, etc.). Arrange the statements in a reasonable order, avoiding bias towards any direction (e.g., always placing correct statements first).

3. **Question Design:**

   - Depending on the need, questions may limit the scope or conditions being tested.

   - "Which of the following statements about [knowledge point] is correct?"

   - "Which of the following statements about [knowledge point] is incorrect?"

4. **Combination Design:**

   - Combine the numbered statements to create options, such as I and II; II and III; III, VI, and VII.

   - Ensure that there is exactly one correct answer among the options.

5. **Answer Identification:**

   - Clearly mark the correct answer within the list of options, ensuring its accuracy and uniqueness.

6. **Field Completion:**

   - Complete the category fields ("discipline","field","subfield") and select the appropriate difficulty level to ensure the integrity and accuracy of the question information.

### G.2.4 Confusion-options Generation

During the annotation process, we use large models to assist in generating distractors. We select Claude-3.5, GPT-4, Doubao, and Qwen2.5-72B-Instruct, and follow the prompt below to generate confusion options. In the annotation process, a model is randomly selected to generate a confusion option, which is then double-checked by the annotator. Finally, the confirmed confusion option is used as the final option.

---

**🔧 Prompt**

You are an expert in creating multiple-choice questions. Your task is to generate plausible but incorrect distractors for a given question that only has one correct answer. You are skilled at introducing subtle yet distinct mathematical errors to the existing answer options, making the distractor look reasonable but still wrong.

**Input Description:**

- You will be given:
  - A question stem.
  - A set of answer options (no numbering).
  - The correct answer (no numbering).

**Guidelines:**

---

1. **Generate Distractor:**
   - The distractor must be incorrect (not the correct answer).
   - The distractor should introduce a subtle mathematical error while maintaining the formula structure.
   - It must be **distinct** from all existing options, including the correct answer.
   - Avoid any **repetition** or overlap with the existing answer options in terms of value or meaning.
   - **Plausibility**: The distractor should look reasonable and appear to be a potential answer, but still be wrong.
   - Ensure the question still has exactly **one correct answer** after adding the distractor.
2. **Uniqueness Check:**
   - **Thoroughly check** all existing options (including the correct one) to ensure the distractor is unique.
   - If the distractor matches any existing option, regenerate it.
   - The distractor must **not** create ambiguity; the correct answer must remain the sole valid choice.
3. **Formatting:**
   - Ensure the distractor matches the format of the existing options (e.g., fractions, exponents, etc.).
   - Avoid formatting inconsistencies such as misplaced symbols or spaces.

**Output Format:**

```
<distractor> your generated distractor here </distractor>
```

Do not include explanations or extra information. Do not include numbering.

**Input:**

```
    Question: "{}",
    Options: {},
    Correct Answer: "{}"
```

**Output:**

# H  Data Filtering and Manual Review Process Details

In the data processing workflow, we employ a three-stage filtering and review mechanism to ensure data quality:

1. **Rule-Based Pre-Check subsection H.1**: Data undergoes initial screening based on pre-defined code rules, quickly identifying and eliminating clearly invalid data points, thereby enhancing efficiency and reducing subsequent workload.
2. **LLM-Based Quality Inspection subsection H.2**: Large language models are utilized to perform in-depth quality assessments of the data, identifying potential errors and inconsistencies, thereby improving the accuracy and completeness of the data.
3. **Manual Quality Review subsection H.3**: Experienced data analysts conduct a final review to ensure the high quality and usability of the data, correcting potential misjudgments and incorporating domain knowledge for a comprehensive evaluation.

## H.1  Rule-Based Pre-Check

Table 8 presents the rules for data filtering and pre-checking, including text normalization, validation of the consistency of questions and options, integrity checks of answers, and requirements for the completeness of metadata such as difficulty, discipline, and others.

| Category | Rules |
|---|---|
| **Text** | Replace full-width punctuation with half-width punctuation. |
| | Remove unnecessary whitespace characters (e.g., spaces, newline characters, tab spaces). |
| | Replace common escape characters. |
| | Add missing escape characters. |
| **Question** | Not Empty or Placeholder. |
| | Ensure the question field is in string format to avoid errors. |
| | Text length > 5. |
| | Text Standardization. |
| | Perplexity $\leq 100$, calculated using the *Qwen2.5-0.5B-Instruct* model. |
| | No Semantic Duplicates: For a given question $q_i$, the cosine similarity with each existing question $q_j$ : $$\text{CosineSimilarity}(q_i, q_j) = \frac{\mathbf{e_i} \cdot \mathbf{e_j}}{\|\mathbf{e_i}\|\|\mathbf{e_j}\|} < 0.90$$ where $\mathbf{e_i}$ and $\mathbf{e_j}$ are embeddings generated by the model SentenceTransformer("all-MiniLM-L6-v2"), and *Faiss* is used for efficient similarity search. |
| **Options** | $4 \leq$ Option count $\leq 10$. |
| | No empty or meaningless options (e.g., no empty strings, no only whitespace characters, no common placeholders like 'none', 'null'). |
| | No duplicate options. |
| | Remove built-in option prefixes, using regular expressions(`r'^([A-Ja-j])[\.\,\s]'`,`r'^\(([A-Ja-j])\)'`, `r'^([A-Ja-j])\)'` ). |
| | Text Standardization for each option. |
| **Answer** | Not Empty or Placeholder. |
| | Within options list. |
| | Text Standardization. |
| **Difficulty** | Ensure difficulty exists. |
| **Discipline** | Ensure discipline, field, and subfield exist. |

Table 8: Detailed Criteria for Rule-Based Pre-Check.

## H.2 LLM-Based Quality Inspection

This section outlines the quality inspection process of LLMs in data processing, including: **1) Validity Check**, **2) Negative and Extreme Inquiry Detection**, **3) Multimodal Exclusion**, **4) Field Relevance Evaluation**, **5) Solvability Assessment**.

### H.2.1 Validity Check

> ⊘ **Purpose**
>
> - Validate the question's correctness, ensuring it follows the standards, can be answered with the given text, and the options are complete and reasonable.

> ⓘ **Prompt**
>
> You are a master of format checking, skilled at verifying whether multiple-choice questions in JSON format conform to the specifications.
>
> You will receive an input in JSON format containing a question, options, and the answer, for example:
>
> ```
> {
> question: "Question content",
> options: ['Option content', 'Option content',
>           'Option content', 'Option content'],
> answer: "Answer content"
> }
> ```
>
> Please determine whether the text in the JSON represents a complete and solvable multiple-choice question. It must meet the following requirements:

1. The question must explicitly pose a **specific problem** or **ask a clear question** that can be answered or calculated. If the question does not clearly ask something (e.g., if it is vague, incomplete, or does not directly ask for a specific answer), it should be considered **invalid**. A valid question should directly require an answer, such as asking for a numerical value or selection (e.g., "What is the value of X?", "Which of the following is correct?", etc.). Questions that lack a clear problem or don't ask for a specific answer must be marked invalid.

2. The question, options, and answer must be fully defined. If any part is missing or unclear (e.g., if the answer does not match any of the listed options), the question should be deemed invalid. Note that the number of options is not a factor in determining completeness—having only one option is acceptable as long as the content is complete and coherent.

3. The question, options, and answer must be directly relevant to each other. The options should not reference each other, and there should be no circular dependencies or inter-referencing between options (e.g., "Option A is true because Option B is false"). Each option must stand independently, with no cross-references to other options.

4. Negative phrasing such as "The following options are incorrect" or "None of the above" is not allowed in the question. The question should use positive phrasing (e.g., "Which of the following is correct?") to avoid confusion. If negative phrasing is detected in the question, it should be deemed invalid.

5. The question must be in a valid multiple-choice format. If the question is not suitable for a multiple-choice format (e.g., it is a free-form answer question like a problem or essay), it should be deemed invalid. Ensure that the question is designed specifically for multiple-choice answering, rather than being a question that requires an open-ended response.

Please return the result in the following JSON format:

```
{
"is_valid": true/false
}
```

Ensure that the output is valid JSON format and do not return any unrelated information.
Input:

```
{
question: "{}",
options: {},
answer: "{}"
}
```

Output:

## H.2.2 Negative and Extreme Inquiry Detection

### ⊘ Purpose

- Identify negative questions that may appear in the options, such as 'Which of the following options is incorrect?' or 'Which of the following is not correct?'.

- Also, identify questions that use "most" to indicate a higher degree of likelihood or preference, such as 'Which of the following is most likely to occur?' or 'Which option is most appropriate?'.

- These types of questions are prone to having distractors that meet the conditions.

### ❶ Prompt 1

Please evaluate whether the following text is a valid and well-formed multiple-choice question, adhering to the following criteria:

1. The question must not be a negation (e.g., "Which of the following is NOT...?" or "What is not...?").

2. The question should avoid vague or ambiguous phrasing, such as "Which of the following is the best/worst...?" or other uncertain expressions.

3. The answer choices should not be all affirmations or all negations, such as "All of the above" or "None of the above," as these are considered inappropriate.

Please return the result in the following JSON format:

```
{
"is_valid": true/false
}
```

Ensure that the output is valid JSON and do not return any unrelated information.

Example input 1:

```
{
question: Which of the following procedure is not done in CHC?,
options: ["Aboion", "Blood transfusion", "Caesaran section",
          "Urine microscopy and culture sensitivity"],
answer: Urine microscopy and culture sensitivity
}
```

Example output 1:

```
{
"is_valid": false
}
```

Example input 2:

```
{
question: Most common viral cause of acquired aqueductal stenosis
is?,
options: ["Rubella", "Mumps", "Toxoplasma", "Enterovirus"],
answer: Mumps
}
```

Example output 2:

```
{
"is_valid": false
}
```

Example input 3:

```
{
question: Acute cerebral edema occurs at high altitude due to,
options: ["apillary blood pressure",
          "Cerebral arteriolar dilation",
          "Hypoxic damage to capillary
           walls leading to capillary
           leak",
          "All of the above"],
answer: All of the above
}
```

Example output 3:

```
{
"is_valid": false
}
```

Example input 4:

```
{
question: Rideal Walker test is used to determine the efficiency
of the,
options: ["Disinfectant",
          "Moist heat sterilisation",
          "Antibiotics",
          "Dry heat sterilization"],
answer: Disinfectant
}
```

Example output 4:

```
    {
    "is_valid": true
    }
Input:
    {
    "question": {},
    "options": {},
    "answer": {}
    }
Output:
```

## ⓘ Prompt 2

**Input Description:**

You will be given:

- A question statement (Question).

**Guidelines:**

1. Analyze the final part of the question (the actual question being asked) to check for specific phrases:

   - "incorrect" in a context like "which of the following is incorrect"
   - "most" in a context like "which is most likely"

2. If either of these phrases appears in the question's final asking part, output `true`, followed by a brief explanation of why the phrase is present.

3. If neither phrase is found in the question's final part, output `false`, followed by a brief explanation.

**Output Format:**

Provide a response with:

- `true` or `false` as the first part of the output
- A brief explanation that justifies your answer

**Example Input:**

**Question:**
"A new technology is introduced in a manufacturing plant. Which of the following is most likely to occur?"

**Example Output:**

```
{
    "is_valid": true/false,
}
```

**Explanation:** Brief explanation of why the phrase is present.

**Question:**

```
{}
```

**Output:**

```
{
    "is_valid": true/false,
}
```

### H.2.3 Multimodal Exclusion

**⊙ Purpose**

- Check if the question relies on multimodal image information, and remove questions involving image information.

**ⓘ Prompt**

You will be given a question or problem statement. Please determine if it strictly requires visual/image input to be solved.

Only output 'true' if the question absolutely cannot be solved without an image (e.g. "What color is the car in this image?", "Describe the graph shown").

For all other cases where the question could potentially be answered with just text, output 'false' (e.g. math problems, logic puzzles, verbal descriptions).

Output format:

- If visual input is strictly required: 'true' + brief reason
- Otherwise: 'false' + brief reason

Question:

{}

### H.2.4 Field Relevance Evaluation

**⊙ Purpose**

- Assess whether the classification labels are appropriate.
- Evaluate the classification level by level:
  - First, check if the "discipline" is relevant to the question.
  - If the "discipline" is appropriate, assess whether the "field" is relevant.
  - If the "field" is suitable, evaluate whether the "subfield" is relevant.
  - If the "subfield" is highly relevant to the question, the entire classification ("discipline," "field," and "subfield") is considered appropriate, even if there are minor mismatches in the "discipline" or "field."
  - If no levels meet the above criteria, or if any level other than "subfield" is deemed inappropriate and the "subfield" is not highly relevant, the classification is considered not relevant.

**ⓘ Prompt**

You are an expert in hierarchical classification. Based on the given question and the provided three-level classification structure (`discipline`, `field`, `subfield`), evaluate whether the classification assigned to the question is appropriate.

**Classification Structure:**

```
{
    "discipline": "{}",
    "field": "{}",
    "subfield": "{}"
}
```

**Question:**

{}

**Instructions:**

1. Evaluate the classification level by level:

- First, check if the `discipline` is relevant to the question.
- If `discipline` is appropriate, evaluate if the `field` is relevant.
- If `field` is appropriate, evaluate if the `subfield` is relevant.

2. Special rule:
   - If the `subfield` is found to be **highly relevant** to the question, the entire classification (`discipline`, `field`, and `subfield`) is considered appropriate, regardless of minor mismatches in `discipline` or `field`.

3. If no levels meet the above criteria, or if any level other than `subfield` is deemed inappropriate without `subfield` being highly relevant, the classification is considered not relevant.

4. Output **strictly and exclusively** in the following JSON format and don't output any explanation, just output JSON:

```
{
    "is_relevant": true/false
}
```

### H.2.5 Completeness Assessment

> **⊘ Purpose**
>
> - Determine if the question is solvable, considering the following two dimensions:
>   - Confidence: The level of confidence in the answer, categorized as "High," "Medium," or "Low."
>   - Missing Information: If an accurate answer cannot be provided, assess whether it is due to missing information (e.g., diagrams, formulas, known conditions) that makes the problem unsolvable, or if it is due to other reasons (e.g., high difficulty or uncertainty).

> **ⓘ Prompt**
>
> Please review and solve the following problems:
>
> 1. Attempt to solve the problem and assess your confidence level in the answer.
> 2. If you cannot provide an accurate answer, evaluate whether it is due to missing information (e.g., diagrams, formulas, known conditions) that makes the problem unsolvable, or if it's due to other reasons (e.g., high difficulty or uncertainty in solving).
> 3. Output format:
>
> ```
> {
>     "final_answer_letter": "A",
>     "confidence": "High",
>     "missing_info": false
> }
> ```
>
> **Rules:**
> - If the problem is missing a "diagram" (such as a geometry question requiring visual input), mark it as `missing_info: true` and explain that the absence of the diagram or visual representation makes the question unsolvable.
> - If the problem contains sufficient information but is difficult (e.g., requiring complex reasoning or having multiple possible solutions), mark `missing_info: false` and assign a confidence level based on the model's reasoning process:
>   - `"High"`: If the problem is simple and straightforward, with a clear solution.
>   - `"Medium"`: If the problem involves multiple steps or requires further reasoning.
>   - `"Low"`: If the problem is complex, involves extensive information, or the model is uncertain about the solution.
> - If the problem is missing key necessary information (such as numbers, formulas, or critical conditions), mark it as `missing_info: true`.

**Output Format:**

```
# The final answer option letter (e.g., A, B, C...)
final_answer_letter: "A"
# Confidence level (High, Medium, Low)
confidence: "High"
# Whether key information is missing, boolean (true or false)
missing_info: false
```

**Examples:**

- **Question:** "Calculate the difference between A and B."

    - `missing_info:  true`
    - `explanation: "Missing the values of A and B, so it cannot be calculated."`
    - `answer: "A"`
    - `confidence: "Low"`

- **Question:** "Given a geometric shape, calculate its area."

    - `missing_info:  true`
    - `explanation: "The diagram is missing, so it cannot be solved."`
    - `answer: "B"`
    - `confidence: "Low"`

- **Question:** "Given the radius of a circle as 5, calculate the area of the circle."

    - `missing_info:  false`
    - `explanation: "The problem is complete and solvable, high confidence."`
    - `answer: "A"`
    - `confidence: "High"`

**Question:**

`{}`

**Output:**

```
final_answer_letter: "A"
confidence: "High"
missing_info: false
```

## H.3  Manual Quality Review

Table 9 outlines the manual review process, focusing on ensuring consistency, clarity, accuracy, global perspective, and appropriate difficulty, while validating the correctness and uniqueness of answers and proper categorization.

| Category | Focus | Description |
|---|---|---|
| **Question** | Source Consistency | Align question source with annotations. |
| | Condition Completeness | Base answers solely on provided text. |
| | | Avoid needing additional conditions like charts or legal clauses. |
| | Question Clarity | Be clear and specific. |
| | | Have only one correct answer. |
| | | Avoid vague or open-ended questions. |
| | Expression Accuracy | Write in English. |
| | | Use precise terminology. |
| | | Accurately translate specialized terms. |
| | Formula and Numerical Accuracy | Write formulas in LaTeX. |
| | | Align key formulas and numerical details. |
| | Global Perspective | Frame from a global perspective. |
| | | Avoid regional bias. |
| **Options** | Expression Accuracy | Accurately translate proper nouns and terminology. |
| | Formula and Numerical Accuracy | Ensure formula and numerical accuracy. |
| | | Correctly format in LaTeX. |
| | Distractor Relevance | Make incorrect options meaningful but not correct. |
| | | Distinguish from correct answers. |
| | | Match correct options in length and format. |
| **Answer** | Answer Correctness | Ensure the answer is in the options. |
| | | Accurately answers the question. |
| | Answer Uniqueness | Ensure the answer is unique. |
| | | Avoid multiple valid answers due to vague wording or incorrect options. |
| **Difficulty** | Difficulty Matching | Match difficulty to graduate-level. |
| | Difficulty Reasonableness | Ensure difficulty is reasonable based on complexity and presented options. |
| **Discipline** | Category Accuracy | Categorize correctly with accurate primary and secondary subjects. |
| | Category Validity | Correctly fill category fields. |
| | | No missing or unnecessary fields. |

Table 9: Detailed Guidelines for Manual Quality Review.

## H.4 Reasons for Failing Quality Inspection

---

### 🌿 Question Block 1

**Question:** Consider a resistor made from a hollow cylinder of carbon as shown below. The inner radius of the cylinder is $R_i = 0.2$mm and the outer radius is $R_o = 0.3$mm. The length of the resistor is $L = 0.9$mm. The resistivity of the carbon is $\rho = 3.5 \times 10^{-5} \ \Omega \cdot m$. What is the resistance?

**Options:**
- 3.0
- 1.0
- 5.0
- 2.0
- 1.5
- 2.8
- 3.5
- 4.0
- 2.5
- 1.8

**Answer:** 2.5

---

## ⚠ Error Analysis Block 1

**Error Type:** Condition Setting Defect
**Error Analysis:** The solution to the correction problem requires a diagram as a condition; without the diagram, the problem cannot be solved. Therefore, this question lacks condition setting and cannot be answered.

## ✎ Question Block 2

**Question:** Which of the following descriptions about the function of gastrointestinal motility is incorrect?

**Options:**

- Controlling the release of bile from the gallbladder
- Facilitating the absorption of nutrients into the bloodstream
- Assisting in the immune response of the gut
- Regulating the pace of digestion
- Stimulating enzyme production in the pancreas
- Breaking down large food molecules into smaller molecules
- Mixing food with digestive juices
- Propelling food through the digestive tract
- Coordinating muscle contractions in the digestive system
- Maintaining the pH balance in the stomach

**Answer:** Breaking down large food molecules into smaller molecules

## ⚠ Error Analysis Block 2

**Error Type:** Incorrect option construction
**Error Analysis:** Options such as stimulating enzyme production in the pancreas, maintaining the pH balance in the stomach, and several other choices are correct answers to the question. Questions involving terms like "incorrect" or "most" require careful construction of distractors, but the reviewer fails to ensure the uniqueness of the correct answer.

## ✎ Question Block 3

**Question:** Ross claims that we learn of our prima facie duties:

**Options:**

- from the moral judgments we make in various situations.
- by seeing the prima facie rightness of particular acts, and then apprehending general principles.
- from the explicit moral instruction we receive as children.
- through societal norms and cultural values.
- from religious teachings or scriptures.
- by observing the consequences of our actions.
- by intuitively understanding moral obligations.
- by proving them philosophically.
- by apprehending general principles, and then inferring the prima facie rightness of particular acts.
- through legal regulations and laws.

**Answer:** by seeing the prima facie rightness of particular acts, and then apprehending general principles

## ⚠ Error Analysis Block 3

**Error Type:** Missing Contextual Information
**Error Analysis:** The problem does not provide sufficient contextual information, such as Ross's background or theories, to allow for proper comprehension and answer selection.

## ✎ Question Block 4

**Question:** About the I-type system tracking the step signal, what's the steady-state error between them?

**Options:**

- 10
- infinity
- 2
- -0.5
- 1
- 0.5
- undefined
- 0
- -1
- 100

**Answer:** 0

## ⚠ Error Analysis Block 4

**Error Type:** Distractor Quality Issue
**Error Analysis:** Some options, such as "infinity" and "undefined," may not be relevant or may be poorly constructed. These distractors may not effectively contribute to creating meaningful confusion.

## ✎ Question Block 5

**Question:** A cube of iron was heated to 70°C and transferred to a beaker containing 100g of water at 20°C. The final temperature of the water and the iron was 23°C. What is the heat capacity?

**Options:**

- $0.310 \ K^{-1} \ g^{-1}$
- $0.740 \ K^{-1} \ g^{-1}$
- $0.582 \ K^{-1} \ g^{-1}$
- $1.200 \ K^{-1} \ g^{-1}$
- $0.453 \ K^{-1} \ g^{-1}$
- $0.235 \ K^{-1} \ g^{-1}$
- $1.004 \ K^{-1} \ g^{-1}$
- $0.505 \ K^{-1} \ g^{-1}$
- $0.875 \ K^{-1} \ g^{-1}$
- $0.690 \ K^{-1} \ g^{-1}$

**Answer:** $0.453 \ K^{-1} \ g^{-1}$

### ⚠ Error Analysis Block 5

**Error Type:** Condition Setting Defect
**Error Analysis:** The problem lacks critical information about the mass of the iron cube, which makes it unsolvable.

### 🖉 Question Block 6

**Question:** The Central Committee of the Communist Party of China and the Central Government are striving to enhance our country's military security functions. In order to achieve this, they should further promote the governance of the military according to law and strict discipline, focusing on the crucial ( ).

**Options:**

- Checks and Balances on the Exercise of Power
- Procedures and Control of Power
- Maintenance and Supervision of Power
- Regulation and Oversight of Authority
- Coordination and Monitoring of Authority
- Systems and Regulation of Governance
- Security and Management of Authority
- Power Operation Management
- Power Operation Guarantee
- Power Operation Coordination

**Answer:** Checks and Balances on the Exercise of Power

### ⚠ Error Analysis Block 6

**Error Type:** Region-Specific Context Missing
**Error Analysis:** The question has a clear regional context (China) that should be accounted for in item selection or modification. Additionally, the item could be better adapted or refined to enhance clarity and relevance.

### 🖉 Question Block 7

**Question:** Use the example below to answer the question that follows. Identify the type of dissonance found above the asterisk in the given voice-leading example.

**Options:**

- Neighbor tone
- Appoggiatura
- Suspension
- Passing tone

**Answer:** Passing tone

### ⚠ Error Analysis Block 7

**Error Type:** Missing Contextual Information
**Error Analysis:** The question lacks the necessary example (voice-leading notation) and cannot be answered.

## Question Block 8

**Question:** Suppose an electric field $\mathbf{E}(x, y, z)$ has the form

$$E_x = ax, \quad E_y = 0, \quad E_z = 0$$

where $a$ is a constant. What is the charge density? How do you account for the fact that the field points in a particular direction, when the charge density is uniform?

**Options:**

- $\epsilon_0 x$
- $\epsilon_0 y$
- $xa\epsilon$
- $\epsilon_0 a$
- $x\epsilon_0$
- $\epsilon_z a$
- $\epsilon_0 z$
- $a\epsilon_y$
- $\epsilon a_y$
- $\epsilon_x a$

**Answer:** $\epsilon_0 a$

## ⚠ Error Analysis Block 8

**Error Type:** Incomplete Question and Answer Construction
**Error Analysis:** The question contains two sub-questions, but the answer and options only address one, causing ambiguity.

| Discipline | Field : Subfield |
|---|---|
| **Agronomy(485)** | **Animal Husbandry(103)** : Animal Nut. & Feed Sci.; Animal Rear. & Breed.
**Aquaculture(56)** : Aquacult.
**Crop Science(145)** : Crop Sci.
**Forestry(131)** : Forest Cult. & Gen. Breed.; Landsc. Plants & Orn. Hort.
**Veterinary Medicine(50)** : Vet. Med. |
| **Economics(873)** | **Applied Economics(723)** : Econ. Stats.; Fin.; Indus. Econ.; Int. Trade; Labor Econ.; Nat. & Def. Econ.; Pub. Fin.; Quant. Econ.
**Theoretical Economics(150)** : Econ. Hist.; Pol. Econ.; West. Econ. |
| **Education(484)** | **Education(247)** : Edu. Tech. & Prin.; Presch. Edu.; Spec. Edu.; Theory of Curric. & Instr.
**Physical Education(150)** : Phys. Edu. & Train.; Sports Hum. & Socio.; Sports Sci. & Med.
**Psychology(87)** : Psychol. |
| **Engineering(7892)** | **Aeronautical and Astronautical Science and Technology(119)** : Aeronaut. & Astronaut. Sci. & Tech.
**Agricultural Engineering(104)** : Agric. Environ. & Soil-Water Eng.; Agric. Mech. Eng.
**Architecture(162)** : Arch. Design & Theory; Arch. Hist.; Urban Plan. & Design
**Chemical Engineering and Technology(410)** : Chem. Transport Eng.; Elem. of Chem. React. Eng.; Fluid Flow & Heat Transfer in Chem. Eng.; Mass Trans. & Sep. Process in Chem. Eng.
**Civil Engineering(358)** : Bridge & Tunnel Eng.; Geotech. Eng.; Struct. Eng.; Urban Infra. Eng.
**Computer Science and Technology(763)** : Adv. Prog. Lang.; Comp. Arch.; Comp. Net.; Comp. Soft. & Theory; Data Struc.; Databases; Formal Lang.; Oper. Sys.; Pattern Recog.; Princip. of Comp. Org.
**Control Science and Engineering(190)** : Control Theory & Eng.; Guid. Nav. & Control; Oper. Res. & Cyber.
**Electrical Engineering(556)** : Elect. Theory & New Tech.; High Volt. & Insul. Tech.; Power Elec. & Elec. Drives; Power Sys. & Autom.
**Electronic Science and Technology(246)** : Circuits & Sys.; Electromag. Field & Microwave Tech.; Microelect. & Solid-State Elec.
**Environmental Science and Engineering(189)** : Environ. Eng.; Environ. Sci.; Environ. & Res. Protect.
**Food Science and Engineering(109)** : Food Biochem.; Food Proc. & Stor. Eng.
**Forestry Engineering(100)** : Forest Eng.; Wood Sci. & Tech.
**Geological Resources and Geological Engineering(50)** : Geol. Res. & Geol. Eng.
**Hydraulic Engineering(218)** : Hydraul. & Hydrol.; Water Cons. & Hydropower Eng.
**Information and Communication Engineering(504)** : Antenna & Radio Comm.; Comm. Prin.; Comm. & Info. Sys.; Optical Fiber Comm.; Signal & Info. Proc.
**Instrument Science and Technology(50)** : Instr. Sci. & Tech.
**Materials Science and Engineering(289)** : Mater. Phys. & Chem.; Mater. Proc. Eng.
**Mechanical Engineering(176)** : Manuf. Autom.; Mechatron. Eng.
**Mechanics(908)** : Fund. of Dyn. & Control; Rigid Body Mech.; Solid Mech.; Theor. Fluid Mech.; Theor. Mech.
**Metallurgical Engineering(255)** : Iron & Steel Metall.; Non-fer. Metall.; Phys. Chem. of Metall. Proc.; Princip. of Metall.
**Mining Engineering(100)** : Mineral Proc. Eng.; Mining & Safety Eng.
**Naval Architecture and Ocean Engineering(138)** : Marine Eng.; Ship Mech. & Design Prin.
**Nuclear Science and Technology(107)** : Nuc. Energy & React. Tech.; Radiation Prot. & Nuclear Tech. Appl.
**Optical Engineering(376)** : Applied Opt.; Laser Tech.; Optoelect. Tech.; Theor. Opt.
**Petroleum and Natural Gas Engineering(112)** : Oil & Gas Field Dev. & Stor. & Trans. Eng.; Poromech. & Res. Phys.
**Power Engineering and Engineering Thermophysics(684)** : Eng. Fluid Mech.; Eng. Thermophys.; Fluid Mach. & Eng.; Heat Trans.; Internal Comb. Eng.; Power Mach. & Eng.; Refrig. & Cryogen. Eng.; Thermal Energy Eng.
**Surveying and Mapping Science and Technology(168)** : Carto. & Geo. Info. Eng.; Dig. Survey. & Remote Sens. Appl.; Geodesy & Survey. Eng.
**Textile Science and Engineering(100)** : Text. Chem. & Dyeing Eng.; Text. Mater. Sci.
**Transportation Engineering(251)** : Road & Rail. Eng.; Traffic Info. Eng. & Control; Transp. Plan. & Manag.; Vehicle Oper. Eng.
**Weapon Science and Technology(100)** : Mil. Chem. & Pyro.; Weapon Syst. Sci. & Eng. |
| **History(674)** | **History(674)** : Archaeol. & Museol.; Hist. Geo.; World Hist. |

Table 10: The Disciplinary Categories of *SuperGPQA* (1/2).

| Discipline | Field : Subfield |
|---|---|
| Law(656) | **Law(591)** : Civil & Comm. Law; Const. & Admin. Law; Contract Law; Crim. Law; Int. Law; Law & Soc. Gov.; Legal Theory & Hist.; Mil. Law; Proced. Law
**Political Science(65)** : Pol. Sci. |
| Literature and Arts(1676) | **Art Studies(603)** : Broad. & TV Art; Dance Stud.; Design Arts; Drama & Opera Stud.; Film Stud.; Fine Arts
**Journalism and Communication(207)** : Comm. & Broad.; Hist. & Theory of Jour. & Media Mngmt.; Jour. & News Prac.
**Language and Literature(440)** : Class. Chinese Lit.; Fr. Lang. & Lit.; Ling. & Appl. Ling.; Lit. Hist.; Lit. Theory; Mod. & Cont. Chinese Lit.; Phil. & Bib.; Russ. Lang. & Lit.
**Musicology(426)** : Comp.; Harm.; Instr. & Perf.; Music Hist., Ed. & Tech.; Music Forms & Anal.; Pitch & Scales |
| Management(501) | **Business Administration(142)** : Bus. & Acct. Mngmt.; Tour. Mngmt. & Tech. Econ. Mngmt.
**Library, Information and Archival Management(150)** : Info. Mngmt. Sci.; Info. Mngmt. & Comm.; Lib. & Arch. Sci.
**Management Science and Engineering(58)** : Mngmt. Sci. & Eng.
**Public Administration(151)** : Ed. Econ., Mngmt. & Soc. Sec.; Land Res. Mngmt. & Admin. Mngmt.; Soc. Med. & Health Mngmt. |
| Medicine(2755) | **Basic Medicine(567)** : For. Med.; Hum. Anat. & Hist.-Emb.; Immun.; Path. Biol.; Pathol. & Pathophys.; Rad. Med.
**Clinical Medicine(1218)** : Anesth.; Clin. Lab. Diagn.; Derm. & Ven.; Emerg. Med.; Geriat. Med.; Imag. & Nucl. Med.; Intern. Med.; Neurol.; Nurs. & Rehabil. Med.; Obst. & Gyneco.; Oncol.; Ophth.; Oto. & Rhinol.; Pediatr.; Psych. & Ment. Health; Surg.
**Pharmacy(278)** : Medic. Chem.; Microbiol. & Biochem. Pharm.; Pharm. Anal.; Pharmaceut.; Pharmacol.
**Public Health and Preventive Medicine(292)** : Epidemiol. & Health Stats.; Health Tox. & Envir. Health; Matern., Child & Adol. Health; Nutr. & Food Hyg.
**Stomatology(132)** : Basic Stom.; Clin. Stom.
**Traditional Chinese Medicine(268)** : Trad. Chin. Health Pres.; Trad. Chin. Med. Theory; Trad. Chin. Pharm. |
| Military Science(205) | **Military Science(205)** : Mil. Command & Info. Systems; Mil. Logistics & Equip.; Mil. Mngmt.; Mil. Thought & Hist. |
| Philosophy(347) | **Philosophy(347)** : Ethics; Logic; Phil. Aesth.; Phil. of Sci. & Tech.; Relig. Stud. |
| Science(9838) | **Astronomy(405)** : Astron. Obs. & Tech.; Astrophys.; Cosmology; Solar Sys. Sci.; Stell. & Interst. Evol.
**Atmospheric Science(203)** : Atm. Phys. & Envir.; Dyn. Meteorol.; Meteorol.
**Biology(1120)** : Biochem. & Mol. Biol.; Biophys.; Botany; Cell Biol.; Ecol.; Genet.; Microbiol.; Physiol.; Zool.
**Chemistry(1769)** : Analyt. Chem.; Electrochem.; Inorg. Chem.; Org. Chem.; Phys. Chem.; Polym. Chem. & Phys.; Radiochem.
**Geography(133)** : Hum. Geogr.; Phys. Geogr.
**Geology(341)** : Geochem.; Miner., Petrol. & Econ. Geol.; Paleontol. & Stratig.; Prin. of Seism. Expl.; Struct. Geol.
**Geophysics(100)** : Solid Earth Geophys.; Space Phys.
**Mathematics(2622)** : Adv. Algebra; Combinat. Math.; Comput. Math.; Crypt.; Discr. Math.; Func. of Complex Vars.; Func. of Real Vars.; Fund. Math.; Fuzzy Math.; Geo. & Topol.; Graph Theory; Group Theory; Math. Anal.; Num. Theory; Num. Anal.; Ord. Diff. Eq.; Poly. & Ser. Exp.; Prob. & Stats.; Spec. Num. Theory; Stoch. Proc.
**Oceanography(200)** : Hydrogeol.; Marine Biol.; Marine Chem.; Underwater Acou.
**Physical Oceanography(50)** : Phys. Oceanogr.
**Physics(2845)** : Acou.; Atom. & Mol. Phys.; Electrodyn.; Fluid Phys.; Part. & Nucl. Phys.; Polym. Phys.; Quant. Mech.; Relativity; Semicond. Phys.; Solid State Phys.; Stat. Mech.; Subatom. & Atom. Phys.; Thermodyn.; Thermo. & Stat. Phys.
**Systems Science(50)** : Sys. Sci. |
| Sociology(143) | **Sociology(143)** : Demo. & Anthrop.; Soc. & Folklore Studies |

Table 11: The Disciplinary Categories of *SuperGPQA* (2/2).

# I Quantitative Overview of Disciplines at Three Levels

| Discipline | Field | Subfield |
|---|---|---|
| Agronomy(485) | Animal Husbandry(103) | Animal Nutrition and Feed Science(52) |
| | | Animal Rearing and Breeding(51) |
| | Aquaculture(56) | Aquaculture(56) |
| | Crop Science(145) | Crop Science(145) |
| | Forestry(131) | Forest Cultivation and Genetic Breeding(81) |
| | | Landscape Plants and Ornamental Horticulture(50) |
| | Veterinary Medicine(50) | Veterinary Medicine(50) |
| Economics(873) | Applied Economics(723) | Economic Statistics(50) |
| | | Finance(176) |
| | | Industrial Economics(84) |
| | | International Trade(69) |
| | | Labor Economics(74) |
| | | National and Defense Economics(50) |
| | | Public Finance(120) |
| | | Quantitative Economics(100) |
| | Theoretical Economics(150) | Economic History(50) |
| | | Political Economy(50) |
| | | Western Economics(50) |
| Education(484) | Education(247) | Educational Technology and Principles(96) |
| | | Preschool Education(50) |
| | | Special Education(50) |
| | | Theory of Curriculum and Instruction(51) |
| | Physical Education(150) | Physical Education and Training(50) |
| | | Sports Humanities and Sociology(50) |
| | | Sports Science and Medicine(50) |
| | Psychology(87) | Psychology(87) |
| Engineering(7892) | Aeronautical and Astronautical Science and Technology(119) | Aeronautical and Astronautical Science and Technology(119) |
| | Agricultural Engineering(104) | Agricultural Environment and Soil-Water Engineering(54) |
| | | Agricultural Mechanization Engineering(50) |
| | Architecture(162) | Architectural Design and Theory(50) |
| | | Architectural History(62) |
| | | Urban Planning and Design(50) |
| | Chemical Engineering and Technology(410) | Chemical Transport Engineering(50) |
| | | Elements of Chemical Reaction Engineering(107) |
| | | Fluid Flow and Heat Transfer in Chemical Engineering(107) |
| | | Mass Transport and Separation Process in Chemical Engineering(146) |
| | Civil Engineering(358) | Bridge and Tunnel Engineering(51) |
| | | Geotechnical Engineering(179) |
| | | Structural Engineering(57) |
| | | Urban Infrastructure Engineering(71) |
| | Computer Science and Technology(763) | Advanced Programming Languages(50) |
| | | Computer Architecture(50) |
| | | Computer Networks(59) |
| | | Computer Software and Theory(79) |
| | | Data Structures(99) |
| | | Databases(157) |
| | | Formal Languages(52) |
| | | Operating Systems(85) |
| | | Pattern Recognition(50) |
| | | Principles of Computer Organization(82) |
| | Control Science and Engineering(190) | Control Theory and Control Engineering(89) |
| | | Guidance, Navigation and Control(51) |
| | | Operations Research and Cybernetics(50) |
| | Electrical Engineering(556) | Electrical Theory and New Technologies(190) |
| | | High Voltage and Insulation Technology(88) |
| | | Power Electronics and Electrical Drives(182) |
| | | Power Systems and Automation(96) |
| | Electronic Science and Technology(246) | Circuits and Systems(108) |
| | | Electromagnetic Field and Microwave Technology(88) |
| | | Microelectronics and Solid-State Electronics(50) |
| | Environmental Science and Engineering(189) | Environmental Engineering(89) |
| | | Environmental Science(50) |
| | | Environmental and Resource Protection(50) |
| | Food Science and Engineering(109) | Food Biochemistry(50) |
| | | Food Processing and Storage Engineering(59) |
| | Forestry Engineering(100) | Forest Engineering(50) |
| | | Wood Science and Technology(50) |
| Engineering(7892) | Geological Resources and Geological Engineering(50) | Geological Resources and Geological Engineering(50) |
| | Hydraulic Engineering(218) | Hydraulics and Hydrology(136) |
| | | Water conservancy and Hydropower Engineering(82) |
| | Information and Communication Engineering(504) | Antenna and Radio Communication(114) |
| | | Communication Principles(116) |
| | | Communication and Information Systems(69) |
| | | Optical Fiber Communication(50) |
| | | Signal and Information Processing(155) |
| | Instrument Science and Technology(50) | Instrument Science and Technology(50) |
| | Materials Science and Engineering(289) | Materials Physics and Chemistry(199) |
| | | Materials Processing Engineering(90) |
| | Mechanical Engineering(176) | Manufacturing Automation(126) |
| | | Mechatronic Engineering(50) |
| | Mechanics(908) | Fundamentals of Dynamics and Control(347) |
| | | Rigid Body Mechanics(173) |
| | | Solid Mechanics(93) |
| | | Theoretical Fluid Mechanics(140) |
| | | Theoretical Mechanics(155) |
| | Metallurgical Engineering(255) | Iron and Steel Metallurgy(105) |
| | | Non-ferrous Metallurgy(50) |
| | | Physical Chemistry of Metallurgical Process(50) |
| | | Principles of Metallurgy(50) |
| | Mining Engineering(100) | Mineral Processing Engineering(50) |
| | | Mining and Safety Engineering(50) |
| | Naval Architecture and Ocean Engineering(138) | Marine Engineering(50) |
| | | Ship Mechanics and Design Principles(88) |
| | Nuclear Science and Technology(107) | Nuclear Energy and Reactor Technology(57) |
| | | Radiation Protection and Nuclear Technology Applications(50) |
| | Optical Engineering(376) | Applied Optics(124) |
| | | Laser Technology(50) |
| | | Optoelectronic Technology(67) |
| | | Theoretical Optics(135) |
| | Petroleum and Natural Gas Engineering(112) | Oil and Gas Field Development and Storage & Transportation Engineering(62) |
| | | Poromechanics and Reservoir Physics(50) |
| | Power Engineering and Engineering Thermophysics(684) | Engineering Fluid Mechanics(84) |
| | | Engineering Thermophysics(146) |
| | | Fluid Machinery and Engineering(64) |
| | | Heat Transfer(129) |
| | | Internal Combustion Engineering(50) |
| | | Power Machinery and Engineering(52) |
| | | Refrigeration and Cryogenic Engineering(50) |
| | | Thermal Energy Engineering(109) |
| | Surveying and Mapping Science and Technology(168) | Cartography and Geographic Information Engineering(50) |
| | | Digital Surveying and Remote Sensing Applications(68) |
| | | Geodesy and Surveying Engineering(50) |
| | Textile Science and Engineering(100) | Textile Chemistry and Dyeing Engineering(50) |
| | | Textile Materials Science(50) |
| | Transportation Engineering(251) | Road and Railway Engineering(50) |
| | | Traffic Information Engineering and Control(65) |
| | | Transportation Planning and Management(86) |
| | | Vehicle Operation Engineering(50) |
| | Weapon Science and Technology(100) | Military Chemistry and Pyrotechnics(50) |
| | | Weapon Systems Science and Engineering(50) |
| History(674) | History(674) | Archaeology and Museology(104) |
| | | Historical Geography(142) |
| | | World History(428) |
| Law(656) | Law(591) | Civil and Commercial Law(118) |
| | | Constitutional and Administrative Law(50) |
| | | Contract Law(50) |
| | | Criminal Law(116) |
| | | International Law(57) |
| | | Law and Social Governance(50) |
| | | Legal Theory and Legal History(50) |
| | | Military Law(50) |
| | | Procedural Law(50) |
| | Political Science(65) | Political Science(65) |
| - | - | - |

Table 12: Comprehensive Overview of Three Levels of Disciplines and Their Sample Sizes.

Left half:

| Discipline | Field | Subfield |
|---|---|---|
| Literature and Arts(1676) | Art Studies(603) | Broadcasting and Television Art(50) |
| | | Dance Studies(50) |
| | | Design Arts(62) |
| | | Drama and Opera Studies(82) |
| | | Film Studies(158) |
| | | Fine Arts(201) |
| | Journalism and Communication(207) | Communication and Broadcasting(58) |
| | | History and Theory of Journalism and Media Management(59) |
| | | Journalism and News Practice(90) |
| | Language and Literature(440) | Classical Chinese Literature(50) |
| | | French Language and Literature(50) |
| | | Linguistics and Applied Linguistics(50) |
| | | Literary History(69) |
| | | Literary Theory(64) |
| | | Modern and Contemporary Chinese Literature(50) |
| | | Philology and Bibliography(50) |
| | | Russian Language and Literature(57) |
| | Musicology(426) | Composition(50) |
| | | Harmony(60) |
| | | Instrumentation and Performance(65) |
| | | Music History, Education, and Technology(145) |
| | | Musical Forms and Analysis(50) |
| | | Pitch and Scales(56) |
| Management(501) | Business Administration(142) | Business and Accounting Management(92) |
| | | Tourism Management and Technological Economics Management(50) |
| | Library, Information and Archival Management(150) | Information Management Science(50) |
| | | Information Management and Communication(50) |
| | | Library and Archival Science(50) |
| | Management Science and Engineering(58) | Management Science and Engineering(58) |
| | Public Administration(151) | Education Economics, Management and Social Security(50) |
| | | Land Resource Management and Administrative Management(51) |
| | | Social Medicine and Health Management(50) |
| Medicine(2755) | Basic Medicine(567) | Forensic Medicine(50) |
| | | Human Anatomy and Histology-Embryology(189) |
| | | Immunology(69) |
| | | Pathogen Biology(94) |
| | | Pathology and Pathophysiology(115) |
| | | Radiation Medicine(50) |
| | Clinical Medicine(1218) | Anesthesiology(50) |
| | | Clinical Laboratory Diagnostics(50) |
| | | Dermatology and Venereology(66) |
| | | Emergency Medicine(66) |
| | | Geriatric Medicine(50) |
| | | Imaging and Nuclear Medicine(67) |
| | | Internal Medicine(202) |
| | | Neurology(189) |
| | | Nursing and Rehabilitation Medicine(50) |
| | | Obstetrics and Gynecology(102) |
| | | Oncology(58) |
| | | Ophthalmology(50) |
| | | Otorhinolaryngology(50) |
| | | Pediatrics(50) |
| | | Psychiatry and Mental Health(50) |
| | | Surgery(68) |
| | Pharmacy(278) | Medicinal Chemistry(50) |
| | | Microbiology and Biochemical Pharmacy(50) |
| | | Pharmaceutical Analysis(50) |
| | | Pharmaceutics(73) |
| | | Pharmacology(55) |
| | Public Health and Preventive Medicine(292) | Epidemiology and Health Statistics(83) |
| | | Health Toxicology and Environmental Health(80) |
| | | Maternal, Child and Adolescent Health(79) |
| | | Nutrition and Food Hygiene(50) |
| | Stomatology(132) | Basic Stomatology(53) |
| | | Clinical Stomatology(79) |
| | Traditional Chinese Medicine(268) | Traditional Chinese Health Preservation(59) |
| | | Traditional Chinese Medicine Theory(90) |
| | | Traditional Chinese Pharmacy(119) |
| Military Science(205) | Military Science(205) | Military Command and Information Systems(50) |
| | | Military Logistics and Equipment(54) |
| | | Military Management(50) |
| | | Military Thought and History(51) |
| Philosophy(347) | Philosophy(347) | Ethics(68) |
| | | Logic(70) |
| | | Philosophical Aesthetics(109) |
| | | Philosophy of Science and Technology(50) |

Right half:

| Discipline | Field | Subfield |
|---|---|---|
| Philosophy(347) | Philosophy(347) | Religious Studies(50) |
| Science(9838) | Astronomy(405) | Astronomical Observation and Technology(100) |
| | | Astrophysics(69) |
| | | Cosmology(78) |
| | | Solar System Science(88) |
| | | Stellar and Interstellar Evolution(70) |
| | Atmospheric Science(203) | Atmospheric Physics and Atmospheric Environment(69) |
| | | Dynamic Meteorology(50) |
| | | Meteorology(84) |
| | Biology(1120) | Biochemistry and Molecular Biology(99) |
| | | Biophysics(50) |
| | | Botany(166) |
| | | Cell Biology(87) |
| | | Ecology(144) |
| | | Genetics(184) |
| | | Microbiology(161) |
| | | Physiology(120) |
| | | Zoology(109) |
| | Chemistry(1769) | Analytical Chemistry(396) |
| | | Electrochemistry(222) |
| | | Inorganic Chemistry(138) |
| | | Organic Chemistry(171) |
| | | Physical Chemistry(706) |
| | | Polymer Chemistry and Physics(76) |
| | | Radiochemistry(60) |
| | Geography(133) | Human Geography(50) |
| | | Physical Geography(83) |
| | Geology(341) | Geochemistry(50) |
| | | Mineralogy, Petrology, and Economic Geology(121) |
| | | Paleontology and Stratigraphy(50) |
| | | Principles of Seismic Exploration(50) |
| | | Structural Geology(70) |
| | Geophysics(100) | Solid Earth Geophysics(50) |
| | | Space physics(50) |
| | Mathematics(2622) | Advanced Algebra(202) |
| | | Combinatorial Mathematics(191) |
| | | Computational Mathematics(50) |
| | | Cryptography(50) |
| | | Discrete Mathematics(89) |
| | | Functions of Complex Variables(122) |
| | | Functions of Real Variables(67) |
| | | Fundamental Mathematics(197) |
| | | Fuzzy Mathematics(50) |
| | | Geometry and Topology(265) |
| | | Graph Theory(51) |
| | | Group Theory(50) |
| | | Mathematical Analysis(288) |
| | | Number Theory(220) |
| | | Numerical Analysis(67) |
| | | Ordinary Differential Equations(160) |
| | | Polynomials and Series Expansions(179) |
| | | Probability and Statistics(224) |
| | | Special Number Theory(50) |
| | | Stochastic Processes(50) |
| | Oceanography(200) | Hydrogeology(50) |
| | | Marine Biology(50) |
| | | Marine Chemistry(50) |
| | | Underwater Acoustics(50) |
| | Physical Oceanography(50) | Physical Oceanography(50) |
| | Physics(2845) | Acoustics(245) |
| | | Atomic and Molecular Physics(210) |
| | | Electrodynamics(583) |
| | | Fluid Physics(134) |
| | | Particle and Nuclear Physics(225) |
| | | Polymer Physics(109) |
| | | Quantum Mechanics(206) |
| | | Relativity(79) |
| | | Semiconductor Physics(61) |
| | | Solid State Physics(147) |
| | | Statistical Mechanics(50) |
| | | Subatomic and Atomic Physics(51) |
| | | Thermodynamics(215) |
| | | Thermodynamics and Statistical Physics(530) |
| | Systems Science(50) | Systems Science(50) |
| Sociology(143) | Sociology(143) | Demography and Anthropology(50) |
| | | Social and Folklore Studies(93) |

Table 12: Continued: Comprehensive Overview of Three Levels of Disciplines and Their Sample Sizes.

## J  Detailed Dataset Statistics

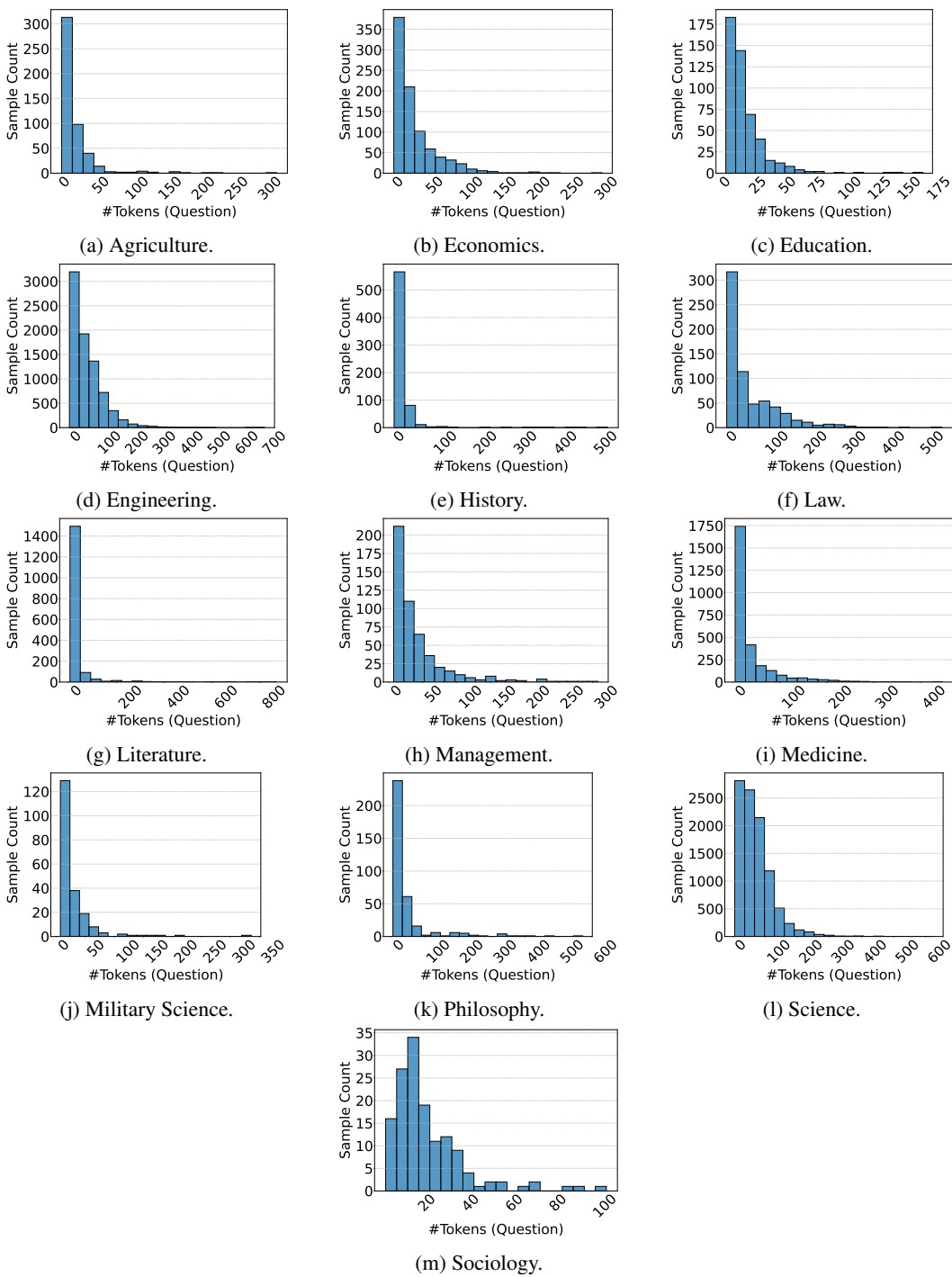

Figure 8: Question Length Distribution Across Disciplines.

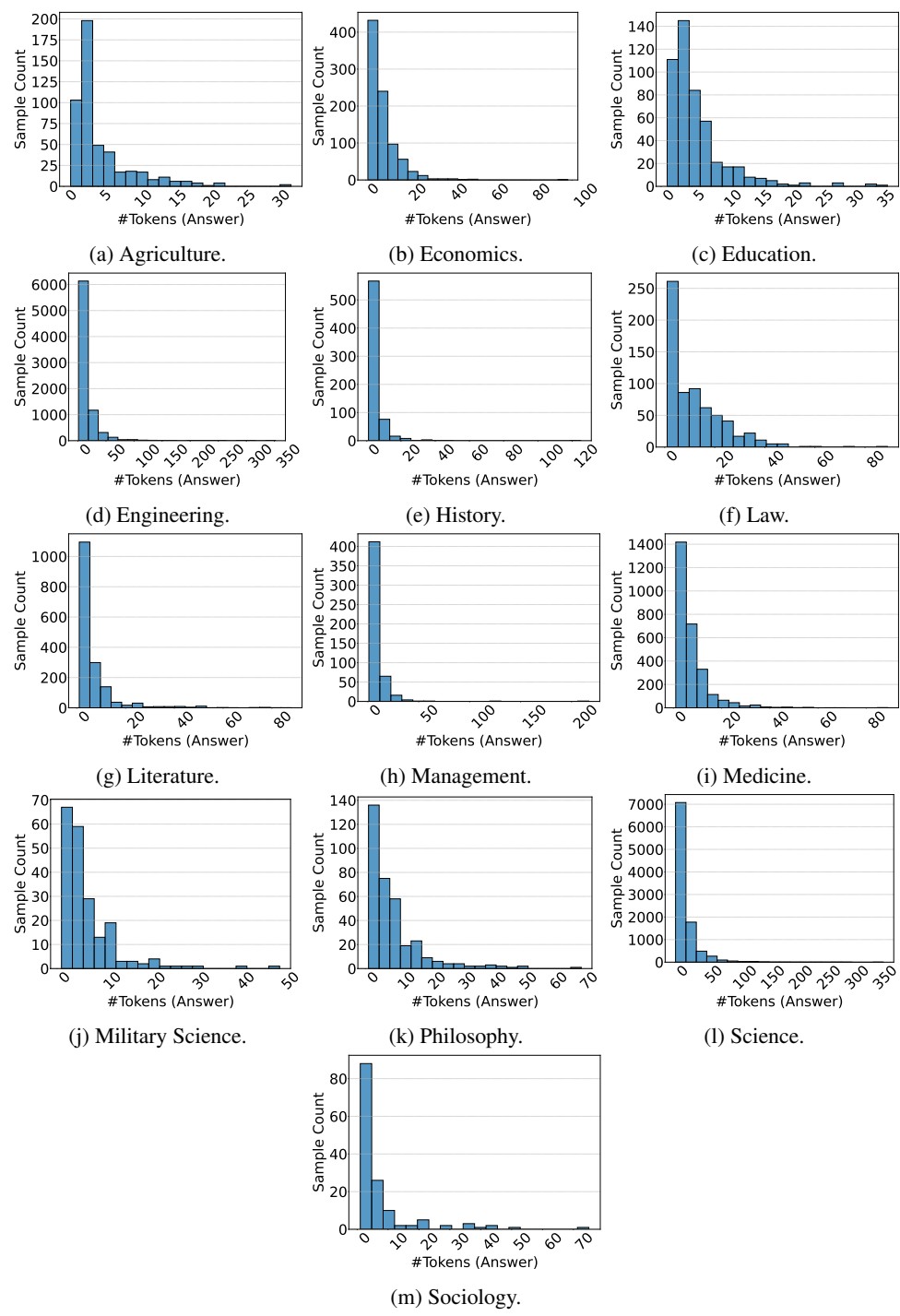

Figure 9: Answer Length Distribution Across Disciplines.

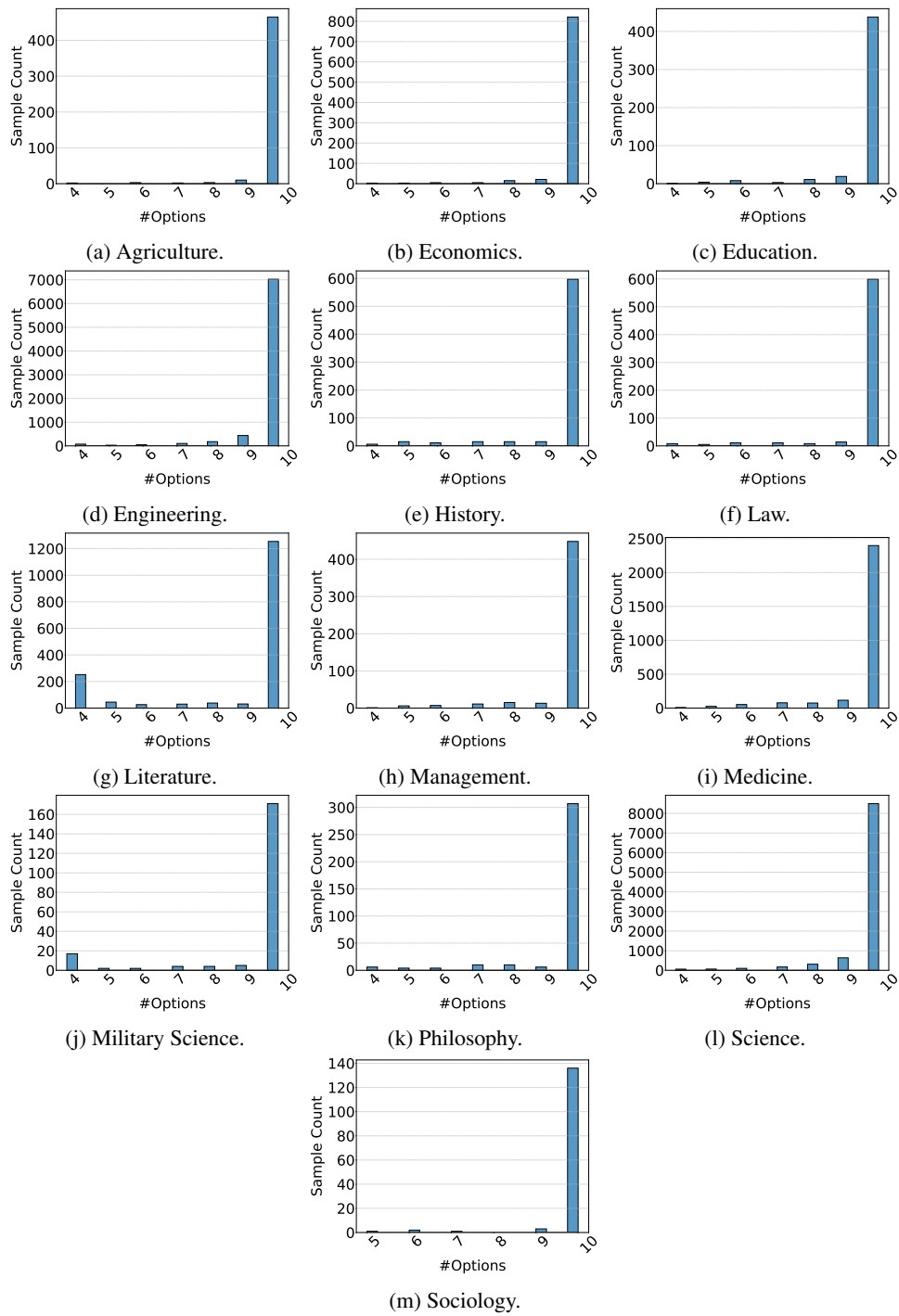

Figure 10: Option Count Distribution Across Disciplines.

# K   Evaluation Prompt

## K.1   Zero-shot Prompt

> ❶ **Zero-shot**
>
> Answer the following multiple-choice question. There is only one correct answer. The last line of your response should be in the format 'Answer: $LETTER (without quotes), where LETTER is one of A, B, C, D, E, F, G, H, I, or J.
>
> {Question}

> ❶ **Question format example:**
>
> Question: A microwave oven is connected to an outlet, 120 V, and draws a current of 2 amps. At what rate is energy being used by the microwave oven?
> A) 240 W
> B) 120 W
> C) 10 W
> D) 480 W
> E) 360 W
> F) 200 W
> G) 30 W
> H) 150 W
> I) 60 W
> J) 300 W

## K.2   Five-shot Prompt

> ❶ **Five-shot**
>
> Answer the following multiple-choice question. There is only one correct answer. The last line of your response should be in the format 'Answer: $LETTER (without quotes), where LETTER is one of A, B, C, D, E, F, G, H, I, or J.
>
> Question: A refracting telescope consists of two converging lenses separated by 100 cm. The eye-piece lens has a focal length of 20 cm. The angular magnification of the telescope is ( ).
> A) 10
> B) 40
> C) 6
> D) 25
> E) 15
> F) 50
> G) 30
> H) 4
> I) 5
> J) 20
> Answer: Let's think step by step. In a refracting telescope, if both lenses are converging, the focus of both lenses must be between the two lenses, and thus the focal lengths of the two lenses must add up to their separation. Since the focal length of one lens is 20 cm, the focal length of the other must be 80 cm. The magnification is the ratio of these two focal lengths, or 4.
> Answer: H.
>
> Question: Say the pupil of your eye has a diameter of 5 mm and you have a telescope with an aperture of 50 cm. How much more light can the telescope gather than your eye?
> A) 1000 times more
> B) 50 times more
> C) 5000 times more
> D) 500 times more
> E) 10000 times more

F) 20000 times more
G) 2000 times more
H) 100 times more
I) 10 times more
J) N/A

Answer: Let's think step by step. The amount of light a telescope can gather compared to the human eye is proportional to the area of its apertures. The area of a circle is given by the formula $A = \pi \left(\frac{D}{2}\right)^2$, where $D$ is the diameter. Therefore, the relative light-gathering power is calculated as:

$$\frac{\left(\frac{50 \text{ cm}}{2}\right)^2}{\left(\frac{5 \text{ mm}}{2}\right)^2} = \frac{\left(\frac{50 \text{ cm}}{0.1 \text{ cm}}\right)^2}{\left(\frac{5 \text{ mm}}{0.1 \text{ cm}}\right)^2} = \frac{500^2}{5^2} = 10000.$$

Answer: E.

Question: Where do most short-period comets come from and how do we know?
A) The Kuiper belt; short period comets tend to be in the plane of the solar system like the Kuiper belt.
B) The asteroid belt; short period comets tend to come from random directions indicating a spherical distribution of comets called the asteroid belt.
C) The asteroid belt; short period comets tend to be in the plane of the solar system just like the asteroid belt.
D) The Oort cloud; short period comets have orbital periods similar to asteroids like Vesta and are found in the plane of the solar system just like the Oort cloud.
E) The Oort Cloud; short period comets tend to come from random directions indicating a spherical distribution of comets called the Oort Cloud.
F) The Oort cloud; short period comets tend to be in the plane of the solar system just like the Oort cloud.
G) The asteroid belt; short period comets have orbital periods similar to asteroids like Vesta and are found in the plane of the solar system just like the asteroid belt.

Answer: Let's think step by step. Most short-period comets originate from the Kuiper belt. This is deduced from the observation that these comets tend to follow orbits that lie in the plane of the solar system, similar to the distribution of objects in the Kuiper belt itself. Thus, the alignment of these cometary orbits with the ecliptic plane points to their Kuiper belt origin.
Answer: A.

Question: Colors in a soap bubble result from light ( ).
A) dispersion
B) deflection
C) refraction
D) reflection
E) interference
F) converted to a different frequency
G) polarization
H) absorption
I) diffraction
J) transmission

Answer: Let's think step by step.The colorful patterns observed in a soap bubble are caused by the phenomenon of light interference. This occurs when light waves bounce between the two surfaces of the soap film, combining constructively or destructively based on their phase differences and the varying thickness of the film. These interactions result in vibrant color patterns due to variations in the intensity of different wavelengths of light.
Answer: E.

Question: A microwave oven is connected to an outlet, 120 V, and draws a current of 2 amps. At what rate is energy being used by the microwave oven?
A) 240 W
B) 120 W
C) 10 W
D) 480 W
E) 360 W
F) 200 W
G) 30 W
H) 150 W
I) 60 W

## L  Further Experiment Analysis

### L.1  Impact of Subfield Information

We conduct zero-shot evaluations under two conditions to investigate the impact of subfield information. 1) zero-shot-with-subfield, where the prompt includes a description of the problem's subfield, and (2)zero-shot-without-subfield, where no subfield information is provided.The prompts employed for the zero-shot-with-subfield evaluation are presented below.

**ⓘ Zero-shot-with-subfield**

Answer the following multiple choice question about

{subfield}

There is only one correct answer. The last line of your response should be in the format 'Answer: $LETTER' (without quotes), where LETTER is one of A, B, C, D, E, F, G, H, I, or J.

**ⓘ Question format example:**

The common-mode rejection ratio of the first stage amplification circuit in a three-op-amp differential circuit is determined by ( ).
A) the absolute value of the difference in the common-mode rejection ratio of A1 and A2 themselves
B) all of the above
C) the average of A1 and A2's common-mode rejection ratios
D) the sum of A1 and A2's common-mode rejection ratios
E) the product of A1 and A2's common-mode rejection ratios
F) the square root of the product of A1 and A2's common-mode rejection ratios
G) the size of A2's common-mode rejection ratio
H) the size of A1's common-mode rejection ratio
I) The difference in the common-mode rejection ratio of A1 and A2 themselves
J) input resistance

### L.2  Robustness of the Evaluation

We employ 4 types of initial prompts and 6 types of question formats, resulting in a combination of 24 different prompt styles to verify the robustness of our bench. The 4 initial prompts and 6 types of question formats are listed below.

**ⓘ Initial Prompt 1**

Answer the following multiple choice question. There is only one correct answer. The last line of your response should be in the format 'Answer: $LETTER' (without quotes), where LETTER is one of A, B, C, D, E, F, G, H, I, or J.

**ⓘ Initial Prompt 2**

You are a helpful assistant. Answer the given multiple-choice question. Only one option is correct. The last line of your response should be in the format 'The correct answer is: $LETTER', where LETTER is one of A, B, C, D, E, F, G, H, I, or J.

**ⓘ Initial Prompt 3**

Select the correct answer for the following multiple-choice question. There is only one valid choice. The last line of your response should be in the format 'Answer: $LETTER' (without quotes), where LETTER is one of A, B, C, D, E, F, G, H, I, or J.

**ⓘ Initial Prompt 4**

Review the following multiple-choice question and choose the one correct answer. Ensure that your response concludes with a line exactly formatted as 'The correct answer is: $LETTER', where LETTER represents one of A, B, C, D, E, F, G, H, I, or J.

E) the product of A1 and A2's common-mode rejection ratios

F) the square root of the product of A1 and A2's common-mode rejection ratios

G) the size of A2's common-mode rejection ratio

H) the size of A1's common-mode rejection ratio

I) The difference in the common-mode rejection ratio of A1 and A2 themselves

J) input resistance Please begin answering.

---

## ⓘ Question Format Example 4

Q: The common-mode rejection ratio of the first stage amplification circuit in a three-op-amp differential circuit is determined by ( ). (A) the absolute value of the difference in the common-mode rejection ratio of A1 and A2 themselves (B) all of the above (C) the average of A1 and A2's common-mode rejection ratios (D) the sum of A1 and A2's common-mode rejection ratios (E) the product of A1 and A2's common-mode rejection ratios (F) the square root of the product of A1 and A2's common-mode rejection ratios (G) the size of A2's common-mode rejection ratio (H) the size of A1's common-mode rejection ratio (I) The difference in the common-mode rejection ratio of A1 and A2 themselves (J) input resistance

---

## ⓘ Question Format Example 5

**Question**: The common-mode rejection ratio of the first stage amplification circuit in a three-op-amp differential circuit is determined by ( ).
**Options**:
(A) the absolute value of the difference in the common-mode rejection ratio of A1 and A2 themselves
(B) all of the above
(C) the average of A1 and A2's common-mode rejection ratios
(D) the sum of A1 and A2's common-mode rejection ratios
(E) the product of A1 and A2's common-mode rejection ratios
(F) the square root of the product of A1 and A2's common-mode rejection ratios
(G) the size of A2's common-mode rejection ratio
(H) the size of A1's common-mode rejection ratio
(I) The difference in the common-mode rejection ratio of A1 and A2 themselves
(J) input resistance

---

## ⓘ Question Format Example 6

Question: The common-mode rejection ratio of the first stage amplification circuit in a three-op-amp differential circuit is determined by ( ).
Options:
A: the absolute value of the difference in the common-mode rejection ratio of A1 and A2 themselves
B: all of the above
C: the average of A1 and A2's common-mode rejection ratios
D: the sum of A1 and A2's common-mode rejection ratios
E: the product of A1 and A2's common-mode rejection ratios
F: the square root of the product of A1 and A2's common-mode rejection ratios
G: the size of A2's common-mode rejection ratio
H: the size of A1's common-mode rejection ratio
I: The difference in the common-mode rejection ratio of A1 and A2 themselves
J: input resistance

# M  Benchmarks for Data Expansion

Table 13: Benchmark List for Data Supplementation in SuperGPQA Annotation.

| Benchmark Name | Title | Year |
|---|---|---|
| LawBench | LawBench: Benchmarking Legal Knowledge of Large Language Models Fei et al. [2023] | 2023 |
| MedMCQA | MedMCQA: A Large-scale Multi-Subject Multi-Choice Dataset for Medical domain Question Answering Pal et al. [2022] | 2022 |
| MedQA | What Disease does this Patient Have? A Large-scale Open Domain Question Answering Dataset from Medical Exams Jin et al. [2020] | 2020 |
| MMLU-Pro | MMLU-Pro: A More Robust and Challenging Multi-Task Language Understanding Benchmark Wang et al. [2024b] | 2024 |
| MMLU-CF | MMLU-CF: A Contamination-free Multi-task Language Understanding Benchmark Zhao et al. [2024] | 2024 |
| ShoppingMMLU | Shopping MMLU: A Massive Multi-Task Online Shopping Benchmark for Large Language Models Jin et al. [2024] | 2024 |
| UTMath | UTMath: Math Evaluation with Unit Test via Reasoning-to-Coding Thoughts Yang et al. [2024b] | 2024 |
| MusicTheoryBench | ChatMusician: Understanding and Generating Music Intrinsically with LLM Yuan et al. [2024] | 2024 |
| Omni-Math | Omni-MATH: A Universal Olympiad Level Mathematic Benchmark For Large Language Models Gao et al. [2024] | 2024 |
| U-MATH | U-MATH: A University-Level Benchmark for Evaluating Mathematical Skills in LLMs Chernyshev et al. [2024] | 2024 |
| Putnam-AXIOM | Putnam-AXIOM: A Functional and Static Benchmark for Measuring Higher Level Mathematical Reasoning Fronsdal et al. [2024] | 2024 |
| Short-form Factuality | Measuring short-form factuality in large language models Wei et al. [2024b] | 2024 |
| Chinese SimpleQA | Chinese SimpleQA: A Chinese Factuality Evaluation for Large Language Models He et al. [2024b] | 2024 |
| AIME-AOPS | AIME Problems and Solutions of Problem Solving [2024] | 2024 |
| AIMO Validation AIME | AIMO Validation AIME: Internal Validation Set for AIMO Progress Prize AI-MO [2024] | 2024 |

# N  Ranking of the Top Five Models in Each of the 285 Subfields

Table 14: Detailed Listing of the Top Five Models in Each of the 285 Subfields.

| Model | Score | Model | Score | Model | Score |
|---|---|---|---|---|---|
| Animal Nutrition and Feed Science | | Applied Optics | | Pharmaceutics | |
| DeepSeek-R1 | 50.00 | DeepSeek-R1 | 58.06 | DeepSeek-R1-Zero | 75.34 |
| DeepSeek-R1-Zero | 50.00 | o3-mini-2025-01-31-high | 57.26 | qwen-max-2025-01-25 | 69.86 |
| claude-3-5-sonnet-20241022 | 48.08 | o1-2024-12-17 | 56.45 | Doubao-1.5-pro-32k-250115 | 68.49 |
| o3-mini-2025-01-31-medium | 48.08 | o3-mini-2025-01-31-medium | 53.23 | Doubao-1.5-pro-32k-241225 | 67.12 |
| Qwen2.5-72B-Instruct | 48.08 | o3-mini-2025-01-31-low | 51.61 | DeepSeek-R1 | 64.38 |
| Animal Rearing and Breeding | | Laser Technology | | Pharmacology | |
| Doubao-1.5-pro-32k-250115 | 49.02 | DeepSeek-R1 | 82.00 | DeepSeek-R1-Zero | 56.36 |
| Doubao-1.5-pro-32k-241225 | 47.06 | o3-mini-2025-01-31-high | 76.00 | DeepSeek-R1 | 54.55 |
| DeepSeek-R1 | 47.06 | o1-2024-12-17 | 70.00 | o1-2024-12-17 | 52.73 |
| MiniMax-Text-01 | 45.10 | DeepSeek-R1-Zero | 70.00 | claude-3-5-sonnet-20241022 | 52.73 |
| DeepSeek-R1-Zero | 45.10 | o3-mini-2025-01-31-medium | 68.00 | qwen-max-2025-01-25 | 52.73 |
| Aquaculture | | Optoelectronic Technology | | Epidemiology and Health Statistics | |
| DeepSeek-R1 | 53.57 | o3-mini-2025-01-31-medium | 53.73 | DeepSeek-R1 | 56.63 |
| DeepSeek-R1-Zero | 53.57 | DeepSeek-V3 | 50.75 | Doubao-1.5-pro-32k-241225 | 55.42 |
| o1-2024-12-17 | 46.43 | qwen-max-2025-01-25 | 49.25 | DeepSeek-R1-Zero | 54.22 |
| claude-3-5-sonnet-20241022 | 42.86 | o3-mini-2025-01-31-low | 47.76 | o1-2024-12-17 | 51.81 |
| Doubao-1.5-pro-32k-250115 | 42.86 | o3-mini-2025-01-31-high | 46.27 | qwen-max-2025-01-25 | 51.81 |
| Crop Science | | Theoretical Optics | | Health Toxicology and Environmental Health | |
| Doubao-1.5-pro-32k-250115 | 53.10 | DeepSeek-R1 | 63.70 | o1-2024-12-17 | 73.75 |
| DeepSeek-R1-Zero | 53.10 | o3-mini-2025-01-31-high | 60.00 | Llama-3.1-405B-Instruct | 68.75 |
| DeepSeek-R1 | 50.34 | DeepSeek-R1-Zero | 57.78 | o3-mini-2025-01-31-high | 67.50 |
| o1-2024-12-17 | 49.66 | o3-mini-2025-01-31-medium | 57.04 | gpt-4o-2024-11-20 | 67.50 |
| qwen-max-2025-01-25 | 46.90 | o1-2024-12-17 | 54.81 | claude-3-5-sonnet-20241022 | 65.00 |
| Forest Cultivation and Genetic Breeding | | Oil and Gas Field Development and Storage & Transportation Engineering | | Maternal, Child and Adolescent Health | |
| DeepSeek-R1 | 56.79 | DeepSeek-R1 | 59.68 | o1-2024-12-17 | 73.42 |
| o1-2024-12-17 | 54.32 | o1-2024-12-17 | 56.45 | claude-3-5-sonnet-20241022 | 72.15 |
| Doubao-1.5-pro-32k-250115 | 51.85 | DeepSeek-R1-Zero | 56.45 | DeepSeek-R1-Zero | 69.62 |
| DeepSeek-R1-Zero | 51.85 | qwen-max-2025-01-25 | 51.61 | Doubao-1.5-pro-32k-250115 | 68.35 |
| Doubao-1.5-pro-32k-241225 | 50.62 | o3-mini-2025-01-31-medium | 50.00 | gpt-4o-2024-11-20 | 68.35 |
| Landscape Plants and Ornamental Horticulture | | Poromechanics and Reservoir Physics | | Nutrition and Food Hygiene | |
| DeepSeek-R1-Zero | 68.00 | o1-2024-12-17 | 68.00 | DeepSeek-R1-Zero | 82.00 |
| DeepSeek-R1 | 66.00 | DeepSeek-R1 | 68.00 | claude-3-5-sonnet-20241022 | 72.00 |
| claude-3-5-sonnet-20241022 | 64.00 | DeepSeek-R1-Zero | 68.00 | o1-2024-12-17 | 70.00 |
| gpt-4o-2024-05-13 | 64.00 | Doubao-1.5-pro-32k-250115 | 66.00 | o3-mini-2025-01-31-medium | 70.00 |
| Mistral-Large-Instruct-2411 | 62.00 | Doubao-1.5-pro-32k-241225 | 64.00 | Doubao-1.5-pro-32k-241225 | 70.00 |
| Veterinary Medicine | | Engineering Fluid Mechanics | | Basic Stomatology | |
| o1-2024-12-17 | 64.00 | o3-mini-2025-01-31-medium | 71.43 | qwen-max-2025-01-25 | 54.72 |
| DeepSeek-R1 | 64.00 | DeepSeek-R1 | 67.86 | o1-2024-12-17 | 52.83 |
| Doubao-1.5-pro-32k-241225 | 58.00 | o3-mini-2025-01-31-low | 66.67 | Doubao-1.5-pro-32k-250115 | 52.83 |
| Doubao-1.5-pro-32k-250115 | 58.00 | o3-mini-2025-01-31-high | 64.29 | claude-3-5-sonnet-20241022 | 49.06 |
| DeepSeek-R1-Zero | 58.00 | o1-2024-12-17 | 63.10 | gpt-4o-2024-08-06 | 47.17 |
| Economic Statistics | | Engineering Thermophysics | | Clinical Stomatology | |
| o1-2024-12-17 | 80.00 | DeepSeek-R1 | 69.86 | o1-2024-12-17 | 55.70 |
| DeepSeek-R1 | 74.00 | DeepSeek-R1-Zero | 65.07 | DeepSeek-R1 | 55.70 |
| o3-mini-2025-01-31-high | 72.00 | o3-mini-2025-01-31-high | 60.96 | Doubao-1.5-pro-32k-250115 | 53.16 |
| o3-mini-2025-01-31-medium | 72.00 | o1-2024-12-17 | 60.27 | qwen-max-2025-01-25 | 51.90 |
| DeepSeek-R1-Zero | 70.00 | o3-mini-2025-01-31-medium | 55.48 | DeepSeek-R1-Zero | 48.10 |
| Finance | | Fluid Machinery and Engineering | | Traditional Chinese Health Preservation | |
| DeepSeek-R1 | 68.18 | DeepSeek-R1 | 68.75 | o1-2024-12-17 | 81.36 |
| DeepSeek-R1-Zero | 68.18 | o1-2024-12-17 | 64.06 | DeepSeek-R1-Zero | 81.36 |
| Doubao-1.5-pro-32k-250115 | 67.61 | o3-mini-2025-01-31-high | 56.25 | DeepSeek-R1 | 79.66 |
| o1-2024-12-17 | 66.48 | o3-mini-2025-01-31-medium | 56.25 | Doubao-1.5-pro-32k-250115 | 79.66 |
| o3-mini-2025-01-31-high | 62.50 | gemini-2.0-flash | 51.56 | qwen-max-2025-01-25 | 69.49 |
| Industrial Economics | | Heat Transfer | | Traditional Chinese Medicine Theory | |

Continued

| International Trade | | Internal Combustion Engineering | | Traditional Chinese Pharmacy | |
|---|---|---|---|---|---|
| DeepSeek-R1-Zero | 66.67 | DeepSeek-R1 | 73.64 | DeepSeek-R1-Zero | 70.00 |
| DeepSeek-R1 | 65.48 | o1-2024-12-17 | 70.54 | qwen-max-2025-01-25 | 67.78 |
| Doubao-1.5-pro-32k-250115 | 65.48 | o3-mini-2025-01-31-high | 63.57 | DeepSeek-R1 | 66.67 |
| Doubao-1.5-pro-32k-241225 | 64.29 | DeepSeek-R1-Zero | 63.57 | o1-2024-12-17 | 63.33 |
| qwen-max-2025-01-25 | 63.10 | o3-mini-2025-01-31-medium | 61.24 | Doubao-1.5-pro-32k-241225 | 60.00 |
| **International Trade** | | **Internal Combustion Engineering** | | **Traditional Chinese Pharmacy** | |
| DeepSeek-R1 | 71.01 | Doubao-1.5-pro-32k-250115 | 68.00 | Doubao-1.5-pro-32k-250115 | 63.87 |
| DeepSeek-R1-Zero | 71.01 | gemini-2.0-flash | 66.00 | DeepSeek-R1-Zero | 62.18 |
| Doubao-1.5-pro-32k-250115 | 69.57 | o1-2024-12-17 | 64.00 | DeepSeek-R1 | 60.50 |
| Doubao-1.5-pro-32k-241225 | 66.67 | claude-3-5-sonnet-20241022 | 64.00 | o1-2024-12-17 | 57.98 |
| qwen-max-2025-01-25 | 62.32 | DeepSeek-R1 | 64.00 | qwen-max-2025-01-25 | 55.46 |
| **Labor Economics** | | **Power Machinery and Engineering** | | **Military Command and Information Systems** | |
| DeepSeek-R1 | 63.51 | DeepSeek-R1 | 46.15 | DeepSeek-R1-Zero | 58.00 |
| Doubao-1.5-pro-32k-250115 | 59.46 | DeepSeek-R1-Zero | 46.15 | Doubao-1.5-pro-32k-241225 | 56.00 |
| DeepSeek-R1-Zero | 59.46 | o3-mini-2025-01-31-medium | 42.31 | claude-3-5-sonnet-20241022 | 54.00 |
| o1-2024-12-17 | 55.41 | Doubao-1.5-pro-32k-250115 | 42.31 | gemini-2.0-flash | 54.00 |
| claude-3-5-sonnet-20241022 | 54.05 | o3-mini-2025-01-31-high | 38.46 | Qwen2.5-72B-Instruct | 54.00 |
| **National and Defense Economics** | | **Refrigeration and Cryogenic Engineering** | | **Military Logistics and Equipment** | |
| DeepSeek-R1 | 68.00 | o3-mini-2025-01-31-medium | 48.00 | o1-2024-12-17 | 55.56 |
| o1-2024-12-17 | 66.00 | DeepSeek-R1 | 48.00 | claude-3-5-sonnet-20241022 | 53.70 |
| DeepSeek-R1-Zero | 66.00 | Doubao-1.5-pro-32k-250115 | 46.00 | gpt-4o-2024-08-06 | 50.00 |
| Doubao-1.5-pro-32k-241225 | 62.00 | DeepSeek-R1-Zero | 44.00 | gpt-4o-2024-05-13 | 50.00 |
| Doubao-1.5-pro-32k-250115 | 60.00 | DeepSeek-V3 | 42.00 | Llama-3.1-405B-Instruct | 50.00 |
| **Public Finance** | | **Thermal Energy Engineering** | | **Military Management** | |
| Doubao-1.5-pro-32k-250115 | 65.00 | DeepSeek-R1 | 63.30 | claude-3-5-sonnet-20241022 | 70.00 |
| DeepSeek-R1-Zero | 64.17 | o3-mini-2025-01-31-high | 58.72 | qwen-max-2024-09-19 | 70.00 |
| Doubao-1.5-pro-32k-241225 | 60.83 | DeepSeek-R1-Zero | 55.96 | DeepSeek-V3 | 68.00 |
| DeepSeek-R1 | 59.17 | Doubao-1.5-pro-32k-250115 | 55.05 | Doubao-1.5-pro-32k-241225 | 68.00 |
| o1-2024-12-17 | 55.00 | o1-2024-12-17 | 54.13 | gemini-2.0-flash | 66.00 |
| **Quantitative Economics** | | **Cartography and Geographic Information Engineering** | | **Military Thought and History** | |
| DeepSeek-R1-Zero | 67.00 | o3-mini-2025-01-31-low | 66.00 | o1-2024-12-17 | 76.47 |
| DeepSeek-R1 | 66.00 | Doubao-1.5-pro-32k-241225 | 66.00 | DeepSeek-R1 | 70.59 |
| o3-mini-2025-01-31-high | 65.00 | o1-2024-12-17 | 64.00 | DeepSeek-R1-Zero | 70.59 |
| Doubao-1.5-pro-32k-250115 | 65.00 | o3-mini-2025-01-31-high | 62.00 | Doubao-1.5-pro-32k-250115 | 62.75 |
| o1-2024-12-17 | 64.00 | o3-mini-2025-01-31-medium | 62.00 | claude-3-5-sonnet-20241022 | 60.78 |
| **Economic History** | | **Digital Surveying and Remote Sensing Applications** | | **Ethics** | |
| DeepSeek-R1 | 78.00 | DeepSeek-R1 | 61.76 | o1-2024-12-17 | 63.24 |
| DeepSeek-R1-Zero | 78.00 | o1-2024-12-17 | 60.29 | gpt-4o-2024-11-20 | 58.82 |
| claude-3-5-sonnet-20241022 | 72.00 | DeepSeek-R1-Zero | 60.29 | qwen-max-2025-01-25 | 55.88 |
| Doubao-1.5-pro-32k-250115 | 72.00 | claude-3-5-sonnet-20241022 | 55.88 | DeepSeek-R1 | 55.88 |
| Doubao-1.5-pro-32k-241225 | 68.00 | qwen-max-2025-01-25 | 54.41 | Doubao-1.5-pro-32k-241225 | 54.41 |
| **Political Economy** | | **Geodesy and Surveying Engineering** | | **Logic** | |
| DeepSeek-R1 | 60.00 | o1-2024-12-17 | 74.00 | o1-2024-12-17 | 64.29 |
| Doubao-1.5-pro-32k-250115 | 60.00 | DeepSeek-R1 | 74.00 | DeepSeek-R1 | 62.86 |
| Doubao-1.5-pro-32k-241225 | 58.00 | DeepSeek-R1-Zero | 72.00 | Doubao-1.5-pro-32k-250115 | 61.43 |
| DeepSeek-R1-Zero | 56.00 | o3-mini-2025-01-31-high | 70.00 | o3-mini-2025-01-31-high | 58.57 |
| DeepSeek-V3 | 54.00 | Doubao-1.5-pro-32k-250115 | 70.00 | DeepSeek-R1-Zero | 55.71 |
| **Western Economics** | | **Textile Chemistry and Dyeing Engineering** | | **Philosophical Aesthetics** | |
| Mistral-Large-Instruct-2411 | 64.00 | DeepSeek-R1 | 62.00 | DeepSeek-R1-Zero | 61.47 |
| DeepSeek-R1-Zero | 64.00 | claude-3-5-sonnet-20241022 | 56.00 | DeepSeek-R1 | 59.63 |
| qwen-max-2025-01-25 | 58.00 | DeepSeek-R1-Zero | 56.00 | Doubao-1.5-pro-32k-250115 | 52.29 |
| DeepSeek-R1 | 58.00 | o1-2024-12-17 | 52.00 | o1-2024-12-17 | 50.46 |
| Doubao-1.5-pro-32k-250115 | 58.00 | Doubao-1.5-pro-32k-250115 | 52.00 | Doubao-1.5-pro-32k-241225 | 49.54 |
| **Educational Technology and Principles** | | **Textile Materials Science** | | **Philosophy of Science and Technology** | |
| DeepSeek-R1 | 61.46 | DeepSeek-R1-Zero | 80.00 | DeepSeek-R1-Zero | 80.00 |
| Doubao-1.5-pro-32k-241225 | 60.42 | o1-2024-12-17 | 78.00 | Doubao-1.5-pro-32k-241225 | 78.00 |
| DeepSeek-R1-Zero | 60.42 | Doubao-1.5-pro-32k-250115 | 78.00 | Doubao-1.5-pro-32k-250115 | 76.00 |
| qwen-max-2025-01-25 | 55.21 | qwen-max-2025-01-25 | 74.00 | o1-2024-12-17 | 74.00 |

| | | | | | |
|---|---|---|---|---|---|
| Doubao-1.5-pro-32k-250115 | 54.17 | DeepSeek-R1 | 74.00 | gpt-4o-2024-11-20 | 70.00 |

### Preschool Education

| Model | Score |
|---|---|
| qwen-max-2025-01-25 | 68.00 |
| gpt-4o-2024-11-20 | 66.00 |
| DeepSeek-R1-Zero | 60.00 |
| MiniMax-Text-01 | 58.00 |
| DeepSeek-R1 | 58.00 |

### Road and Railway Engineering

| Model | Score |
|---|---|
| DeepSeek-R1-Zero | 56.00 |
| DeepSeek-R1 | 54.00 |
| qwen-max-2025-01-25 | 52.00 |
| Doubao-1.5-pro-32k-241225 | 50.00 |
| o1-2024-12-17 | 46.00 |

### Religious Studies

| Model | Score |
|---|---|
| Doubao-1.5-pro-32k-250115 | 80.00 |
| DeepSeek-R1 | 76.00 |
| DeepSeek-R1-Zero | 76.00 |
| qwen-max-2025-01-25 | 72.00 |
| phi-4 | 72.00 |

### Special Education

| Model | Score |
|---|---|
| DeepSeek-R1-Zero | 64.00 |
| qwen-max-2024-09-19 | 62.00 |
| qwen-max-2025-01-25 | 62.00 |
| Qwen2.5-72B | 62.00 |
| claude-3-5-sonnet-20241022 | 60.00 |

### Traffic Information Engineering and Control

| Model | Score |
|---|---|
| DeepSeek-R1 | 56.92 |
| o1-2024-12-17 | 55.38 |
| DeepSeek-R1-Zero | 52.31 |
| o3-mini-2025-01-31-high | 47.69 |
| Doubao-1.5-pro-32k-241225 | 46.15 |

### Astronomical Observation and Technology

| Model | Score |
|---|---|
| DeepSeek-R1-Zero | 72.00 |
| MiniMax-Text-01 | 66.00 |
| o1-2024-12-17 | 65.00 |
| o3-mini-2025-01-31-high | 65.00 |
| DeepSeek-V3 | 64.00 |

### Theory of Curriculum and Instruction

| Model | Score |
|---|---|
| DeepSeek-R1-Zero | 66.67 |
| DeepSeek-R1 | 62.75 |
| Doubao-1.5-pro-32k-250115 | 62.75 |
| qwen-max-2025-01-25 | 58.82 |
| o1-2024-12-17 | 56.86 |

### Transportation Planning and Management

| Model | Score |
|---|---|
| o1-2024-12-17 | 66.28 |
| DeepSeek-R1-Zero | 66.28 |
| qwen-max-2025-01-25 | 65.12 |
| Doubao-1.5-pro-32k-241225 | 60.47 |
| DeepSeek-R1 | 59.30 |

### Astrophysics

| Model | Score |
|---|---|
| o3-mini-2025-01-31-medium | 60.87 |
| o1-2024-12-17 | 59.42 |
| DeepSeek-R1 | 57.97 |
| o3-mini-2025-01-31-high | 56.52 |
| MiniMax-Text-01 | 55.07 |

### Physical Education and Training

| Model | Score |
|---|---|
| qwen-max-2025-01-25 | 50.00 |
| DeepSeek-R1-Zero | 50.00 |
| o1-2024-12-17 | 48.00 |
| o3-mini-2025-01-31-low | 46.00 |
| DeepSeek-R1 | 44.00 |

### Vehicle Operation Engineering

| Model | Score |
|---|---|
| o1-2024-12-17 | 66.00 |
| o3-mini-2025-01-31-high | 64.00 |
| DeepSeek-R1 | 62.00 |
| DeepSeek-R1-Zero | 62.00 |
| gemini-2.0-flash | 60.00 |

### Cosmology

| Model | Score |
|---|---|
| o3-mini-2025-01-31-high | 71.79 |
| DeepSeek-R1 | 70.51 |
| Doubao-1.5-pro-32k-250115 | 69.23 |
| o1-2024-12-17 | 66.67 |
| qwen-max-2025-01-25 | 64.10 |

### Sports Humanities and Sociology

| Model | Score |
|---|---|
| DeepSeek-R1-Zero | 66.00 |
| gpt-4o-2024-11-20 | 60.00 |
| o1-2024-12-17 | 56.00 |
| claude-3-5-sonnet-20241022 | 56.00 |
| gpt-4o-2024-08-06 | 56.00 |

### Military Chemistry and Pyrotechnics

| Model | Score |
|---|---|
| claude-3-5-sonnet-20241022 | 68.00 |
| gpt-4o-2024-05-13 | 68.00 |
| o1-2024-12-17 | 62.00 |
| gpt-4o-2024-11-20 | 62.00 |
| DeepSeek-R1-Zero | 62.00 |

### Solar System Science

| Model | Score |
|---|---|
| o3-mini-2025-01-31-high | 69.32 |
| DeepSeek-R1 | 67.05 |
| o1-2024-12-17 | 62.50 |
| o3-mini-2025-01-31-medium | 60.23 |
| DeepSeek-R1-Zero | 56.82 |

### Sports Science and Medicine

| Model | Score |
|---|---|
| Doubao-1.5-pro-32k-250115 | 72.00 |
| DeepSeek-R1-Zero | 70.00 |
| Doubao-1.5-pro-32k-241225 | 68.00 |
| claude-3-5-sonnet-20241022 | 66.00 |
| qwen-max-2024-09-19 | 66.00 |

### Weapon Systems Science and Engineering

| Model | Score |
|---|---|
| phi-4 | 58.00 |
| o1-2024-12-17 | 56.00 |
| gemini-2.0-flash | 54.00 |
| o1-mini-2024-09-12 | 52.00 |
| gpt-4o-2024-08-06 | 52.00 |

### Stellar and Interstellar Evolution

| Model | Score |
|---|---|
| o3-mini-2025-01-31-medium | 67.14 |
| o1-2024-12-17 | 65.71 |
| DeepSeek-R1 | 64.29 |
| DeepSeek-R1-Zero | 64.29 |
| o3-mini-2025-01-31-high | 60.00 |

### Psychology

| Model | Score |
|---|---|
| Doubao-1.5-pro-32k-250115 | 56.32 |
| Doubao-1.5-pro-32k-241225 | 54.02 |
| DeepSeek-R1-Zero | 52.87 |
| claude-3-5-sonnet-20241022 | 50.57 |
| o1-2024-12-17 | 48.28 |

### Archaeology and Museology

| Model | Score |
|---|---|
| DeepSeek-R1-Zero | 73.08 |
| DeepSeek-R1 | 72.12 |
| o1-2024-12-17 | 70.19 |
| qwen-max-2025-01-25 | 60.58 |
| claude-3-5-sonnet-20241022 | 59.62 |

### Atmospheric Physics and Atmospheric Environment

| Model | Score |
|---|---|
| claude-3-5-sonnet-20241022 | 55.07 |
| Doubao-1.5-pro-32k-250115 | 52.17 |
| Doubao-1.5-pro-32k-241225 | 50.72 |
| DeepSeek-R1-Zero | 50.72 |
| DeepSeek-R1 | 49.28 |

### Aeronautical and Astronautical Science and Technology

| Model | Score |
|---|---|
| DeepSeek-R1 | 65.55 |
| DeepSeek-R1-Zero | 57.14 |
| o1-2024-12-17 | 56.30 |
| o3-mini-2025-01-31-high | 54.62 |
| o3-mini-2025-01-31-medium | 52.94 |

### Historical Geography

| Model | Score |
|---|---|
| o1-2024-12-17 | 54.93 |
| claude-3-5-sonnet-20241022 | 54.23 |
| Llama-3.1-405B-Instruct | 53.52 |
| DeepSeek-R1-Zero | 49.30 |
| gpt-4o-2024-11-20 | 47.18 |

### Dynamic Meteorology

| Model | Score |
|---|---|
| DeepSeek-R1 | 72.00 |
| o1-2024-12-17 | 66.00 |
| DeepSeek-R1-Zero | 66.00 |
| claude-3-5-sonnet-20241022 | 64.00 |
| Doubao-1.5-pro-32k-241225 | 64.00 |

### Agricultural Environment and Soil-Water Engineering

| Model | Score |
|---|---|
| DeepSeek-R1-Zero | 64.81 |
| o1-2024-12-17 | 57.41 |
| DeepSeek-R1 | 55.56 |
| MiniMax-Text-01 | 53.70 |
| o3-mini-2025-01-31-high | 53.70 |

### World History

| Model | Score |
|---|---|
| o1-2024-12-17 | 60.75 |
| DeepSeek-R1-Zero | 58.18 |
| DeepSeek-R1 | 54.44 |
| gpt-4o-2024-11-20 | 53.27 |
| claude-3-5-sonnet-20241022 | 51.87 |

### Meteorology

| Model | Score |
|---|---|
| DeepSeek-R1 | 67.86 |
| DeepSeek-R1-Zero | 65.48 |
| claude-3-5-sonnet-20241022 | 54.76 |
| Doubao-1.5-pro-32k-250115 | 54.76 |
| Doubao-1.5-pro-32k-241225 | 52.38 |

### Agricultural Mechanization Engineering

| Model | Score |
|---|---|
| o3-mini-2025-01-31-high | 70.00 |

### Civil and Commercial Law

| Model | Score |
|---|---|
| DeepSeek-R1-Zero | 62.71 |

### Biochemistry and Molecular Biology

| Model | Score |
|---|---|
| DeepSeek-R1 | 68.69 |

| Model | Score | Model | Score | Model | Score |
|---|---|---|---|---|---|
| DeepSeek-R1 | 70.00 | MiniMax-Text-01 | 59.32 | Doubao-1.5-pro-32k-250115 | 64.65 |
| o1-2024-12-17 | 66.00 | o1-2024-12-17 | 58.47 | o1-2024-12-17 | 63.64 |
| o3-mini-2025-01-31-medium | 66.00 | DeepSeek-R1 | 58.47 | DeepSeek-R1-Zero | 63.64 |
| DeepSeek-V3 | 64.00 | claude-3-5-sonnet-20241022 | 57.63 | Doubao-1.5-pro-32k-241225 | 62.63 |
| Architectural Design and Theory | | Constitutional and Administrative Law | | Biophysics | |
| qwen-max-2024-09-19 | 60.00 | DeepSeek-R1-Zero | 72.00 | o1-2024-12-17 | 72.00 |
| Doubao-1.5-pro-32k-250115 | 60.00 | DeepSeek-R1 | 70.00 | DeepSeek-R1 | 72.00 |
| Llama-3.1-405B-Instruct | 58.00 | claude-3-5-sonnet-20241022 | 66.00 | DeepSeek-R1-Zero | 70.00 |
| DeepSeek-R1-Zero | 58.00 | qwen-max-2025-01-25 | 62.00 | o1-mini-2024-09-12 | 68.00 |
| o1-2024-12-17 | 56.00 | Doubao-1.5-pro-32k-250115 | 62.00 | DeepSeek-V3 | 68.00 |
| Architectural History | | Contract Law | | Botany | |
| DeepSeek-R1-Zero | 67.74 | DeepSeek-R1-Zero | 78.00 | o1-2024-12-17 | 54.82 |
| DeepSeek-R1 | 66.13 | o1-2024-12-17 | 74.00 | DeepSeek-R1 | 54.22 |
| o1-2024-12-17 | 61.29 | claude-3-5-sonnet-20241022 | 70.00 | DeepSeek-R1-Zero | 53.61 |
| Doubao-1.5-pro-32k-250115 | 61.29 | DeepSeek-R1 | 70.00 | Doubao-1.5-pro-32k-241225 | 49.40 |
| qwen-max-2025-01-25 | 56.45 | Llama-3.3-70B-Instruct | 70.00 | Doubao-1.5-pro-32k-250115 | 46.99 |
| Urban Planning and Design | | Criminal Law | | Cell Biology | |
| DeepSeek-R1 | 68.00 | DeepSeek-R1 | 68.10 | DeepSeek-R1 | 59.77 |
| qwen-max-2025-01-25 | 64.00 | Doubao-1.5-pro-32k-250115 | 67.24 | o1-2024-12-17 | 56.32 |
| DeepSeek-R1-Zero | 64.00 | DeepSeek-R1-Zero | 62.93 | o3-mini-2025-01-31-low | 54.02 |
| Doubao-1.5-pro-32k-250115 | 60.00 | o1-2024-12-17 | 62.07 | Doubao-1.5-pro-32k-250115 | 52.87 |
| MiniMax-Text-01 | 58.00 | qwen-max-2025-01-25 | 58.62 | DeepSeek-R1-Zero | 52.87 |
| Chemical Transport Engineering | | International Law | | Ecology | |
| DeepSeek-R1 | 76.00 | DeepSeek-R1-Zero | 56.14 | o1-2024-12-17 | 56.94 |
| o3-mini-2025-01-31-high | 70.00 | o1-2024-12-17 | 50.88 | DeepSeek-R1-Zero | 55.56 |
| o1-2024-12-17 | 66.00 | DeepSeek-R1 | 50.88 | DeepSeek-R1 | 50.00 |
| o3-mini-2025-01-31-medium | 66.00 | claude-3-5-sonnet-20241022 | 47.37 | Doubao-1.5-pro-32k-250115 | 50.00 |
| gemini-2.0-flash | 64.00 | Doubao-1.5-pro-32k-241225 | 47.37 | o3-mini-2025-01-31-high | 47.22 |
| Elements of Chemical Reaction Engineering | | Law and Social Governance | | Genetics | |
| o1-2024-12-17 | 70.09 | o1-2024-12-17 | 74.00 | DeepSeek-R1 | 66.85 |
| o3-mini-2025-01-31-high | 70.09 | qwen-max-2025-01-25 | 68.00 | o1-2024-12-17 | 65.22 |
| DeepSeek-R1 | 67.29 | DeepSeek-R1-Zero | 68.00 | DeepSeek-R1-Zero | 64.13 |
| DeepSeek-R1-Zero | 66.36 | gemini-2.0-flash | 66.00 | Doubao-1.5-pro-32k-250115 | 63.04 |
| o3-mini-2025-01-31-medium | 64.49 | DeepSeek-R1 | 66.00 | o3-mini-2025-01-31-high | 61.96 |
| Fluid Flow and Heat Transfer in Chemical Engineering | | Legal Theory and Legal History | | Microbiology | |
| DeepSeek-R1 | 62.62 | DeepSeek-R1 | 78.00 | o1-2024-12-17 | 49.69 |
| DeepSeek-R1-Zero | 58.88 | DeepSeek-R1-Zero | 72.00 | Doubao-1.5-pro-32k-250115 | 49.69 |
| Doubao-1.5-pro-32k-250115 | 54.21 | o1-2024-12-17 | 70.00 | DeepSeek-R1 | 49.07 |
| o1-2024-12-17 | 51.40 | claude-3-5-sonnet-20241022 | 70.00 | o3-mini-2025-01-31-medium | 48.45 |
| Doubao-1.5-pro-32k-241225 | 51.40 | Llama-3.1-405B-Instruct | 68.00 | Doubao-1.5-pro-32k-241225 | 47.20 |
| Mass Transport and Separation Process in Chemical Engineering | | Military Law | | Physiology | |
| DeepSeek-R1 | 72.60 | DeepSeek-R1-Zero | 82.00 | o1-2024-12-17 | 61.67 |
| DeepSeek-R1-Zero | 71.23 | claude-3-5-sonnet-20241022 | 76.00 | DeepSeek-R1 | 61.67 |
| o1-2024-12-17 | 65.75 | Doubao-1.5-pro-32k-250115 | 76.00 | Doubao-1.5-pro-32k-250115 | 57.50 |
| o3-mini-2025-01-31-high | 60.96 | gpt-4o-2024-11-20 | 76.00 | o3-mini-2025-01-31-high | 53.33 |
| Doubao-1.5-pro-32k-250115 | 58.90 | Llama-3.1-405B-Instruct | 76.00 | DeepSeek-R1-Zero | 53.33 |
| Bridge and Tunnel Engineering | | Procedural Law | | Zoology | |
| Doubao-1.5-pro-32k-250115 | 58.82 | o1-2024-12-17 | 78.00 | o1-2024-12-17 | 57.80 |
| Doubao-1.5-pro-32k-241225 | 54.90 | claude-3-5-sonnet-20241022 | 74.00 | DeepSeek-R1 | 53.21 |
| o1-2024-12-17 | 52.94 | DeepSeek-R1 | 68.00 | DeepSeek-R1-Zero | 53.21 |
| DeepSeek-R1-Zero | 52.94 | DeepSeek-R1-Zero | 68.00 | Doubao-1.5-pro-32k-241225 | 47.71 |
| DeepSeek-R1 | 50.98 | gpt-4o-2024-11-20 | 62.00 | Doubao-1.5-pro-32k-250115 | 47.71 |
| Geotechnical Engineering | | Political Science | | Analytical Chemistry | |
| DeepSeek-R1 | 60.34 | DeepSeek-R1-Zero | 60.00 | DeepSeek-R1 | 58.33 |
| DeepSeek-R1-Zero | 51.96 | DeepSeek-R1 | 58.46 | DeepSeek-R1-Zero | 55.56 |
| Doubao-1.5-pro-32k-250115 | 49.72 | gpt-4o-2024-11-20 | 55.38 | Doubao-1.5-pro-32k-250115 | 54.80 |
| o1-2024-12-17 | 49.16 | o1-2024-12-17 | 53.85 | o1-2024-12-17 | 54.04 |
| claude-3-5-sonnet-20241022 | 46.93 | DeepSeek-V3 | 53.85 | o3-mini-2025-01-31-high | 52.27 |
| Structural Engineering | | Broadcasting and Television Art | | Electrochemistry | |
| DeepSeek-R1 | 56.14 | o1-2024-12-17 | 66.00 | o3-mini-2025-01-31-high | 50.90 |

Continued

| Model | Score | Model | Score | Model | Score |
|---|---|---|---|---|---|
| Doubao-1.5-pro-32k-250115 | 54.39 | DeepSeek-R1-Zero | 64.00 | o1-2024-12-17 | 48.65 |
| Doubao-1.5-pro-32k-241225 | 52.63 | gpt-4o-2024-08-06 | 62.00 | DeepSeek-R1 | 47.75 |
| DeepSeek-R1-Zero | 52.63 | gpt-4o-2024-05-13 | 62.00 | DeepSeek-R1-Zero | 47.30 |
| o1-2024-12-17 | 43.86 | Mistral-Large-Instruct-2411 | 60.00 | Doubao-1.5-pro-32k-250115 | 46.85 |
| **Urban Infrastructure Engineering** | | **Dance Studies** | | **Inorganic Chemistry** | |
| DeepSeek-R1-Zero | 61.97 | o1-2024-12-17 | 46.00 | o1-2024-12-17 | 56.52 |
| claude-3-5-sonnet-20241022 | 52.11 | gpt-4o-2024-08-06 | 46.00 | DeepSeek-R1 | 55.80 |
| qwen-max-2025-01-25 | 52.11 | claude-3-5-sonnet-20241022 | 42.00 | o3-mini-2025-01-31-high | 51.45 |
| Qwen2.5-72B-Instruct | 50.70 | gpt-4o-2024-05-13 | 42.00 | o3-mini-2025-01-31-low | 49.28 |
| DeepSeek-R1 | 49.30 | DeepSeek-R1-Zero | 40.00 | Doubao-1.5-pro-32k-250115 | 47.83 |
| **Advanced Programming Languages** | | **Design Arts** | | **Organic Chemistry** | |
| o1-2024-12-17 | 82.00 | DeepSeek-R1 | 46.77 | DeepSeek-R1 | 57.31 |
| o3-mini-2025-01-31-high | 78.00 | o1-2024-12-17 | 45.16 | DeepSeek-R1-Zero | 56.73 |
| o3-mini-2025-01-31-medium | 76.00 | claude-3-5-sonnet-20241022 | 45.16 | o3-mini-2025-01-31-high | 54.39 |
| DeepSeek-R1-Zero | 76.00 | DeepSeek-R1-Zero | 45.16 | o1-2024-12-17 | 52.05 |
| o3-mini-2025-01-31-low | 74.00 | Mistral-Large-Instruct-2411 | 40.32 | o3-mini-2025-01-31-medium | 49.71 |
| **Computer Architecture** | | **Drama and Opera Studies** | | **Physical Chemistry** | |
| o1-2024-12-17 | 72.00 | DeepSeek-R1 | 58.54 | DeepSeek-R1-Zero | 47.31 |
| DeepSeek-R1-Zero | 72.00 | DeepSeek-R1-Zero | 58.54 | o3-mini-2025-01-31-high | 46.46 |
| DeepSeek-R1 | 64.00 | o1-2024-12-17 | 57.32 | o1-2024-12-17 | 46.18 |
| QwQ | 62.00 | claude-3-5-sonnet-20241022 | 54.88 | DeepSeek-R1 | 45.04 |
| qwen-max-2025-01-25 | 60.00 | Doubao-1.5-pro-32k-250115 | 53.66 | Doubao-1.5-pro-32k-250115 | 44.48 |
| **Computer Networks** | | **Film Studies** | | **Polymer Chemistry and Physics** | |
| DeepSeek-R1 | 64.41 | DeepSeek-R1-Zero | 65.82 | DeepSeek-R1 | 67.11 |
| o1-2024-12-17 | 62.71 | o1-2024-12-17 | 60.13 | o3-mini-2025-01-31-medium | 55.26 |
| MiniMax-Text-01 | 59.32 | DeepSeek-R1 | 56.33 | DeepSeek-R1-Zero | 55.26 |
| o3-mini-2025-01-31-low | 59.32 | claude-3-5-sonnet-20241022 | 55.06 | o1-2024-12-17 | 53.95 |
| o3-mini-2025-01-31-medium | 57.63 | DeepSeek-V3 | 54.43 | o3-mini-2025-01-31-high | 53.95 |
| **Computer Software and Theory** | | **Fine Arts** | | **Radiochemistry** | |
| DeepSeek-R1 | 56.96 | claude-3-5-sonnet-20241022 | 61.69 | o3-mini-2025-01-31-high | 60.00 |
| o1-2024-12-17 | 55.70 | o1-2024-12-17 | 58.21 | o1-2024-12-17 | 58.33 |
| claude-3-5-sonnet-20241022 | 53.16 | DeepSeek-R1-Zero | 56.22 | DeepSeek-R1 | 58.33 |
| o3-mini-2025-01-31-high | 51.90 | gpt-4o-2024-08-06 | 53.23 | o3-mini-2025-01-31-medium | 55.00 |
| DeepSeek-R1-Zero | 51.90 | gpt-4o-2024-11-20 | 53.23 | DeepSeek-R1-Zero | 55.00 |
| **Data Structures** | | **Communication and Broadcasting** | | **Human Geography** | |
| DeepSeek-R1 | 65.66 | o1-2024-12-17 | 62.07 | gpt-4o-2024-11-20 | 80.00 |
| o3-mini-2025-01-31-high | 62.63 | DeepSeek-R1-Zero | 51.72 | DeepSeek-R1-Zero | 80.00 |
| o1-2024-12-17 | 59.60 | gemini-2.0-flash | 46.55 | o1-2024-12-17 | 72.00 |
| o3-mini-2025-01-31-medium | 57.58 | gpt-4o-2024-11-20 | 46.55 | gpt-4o-2024-08-06 | 72.00 |
| DeepSeek-R1-Zero | 56.57 | DeepSeek-V3 | 43.10 | Doubao-1.5-pro-32k-241225 | 72.00 |
| **Databases** | | **History and Theory of Journalism and Media Management** | | **Physical Geography** | |
| DeepSeek-R1 | 66.88 | DeepSeek-R1-Zero | 54.24 | DeepSeek-R1-Zero | 56.63 |
| qwen-max-2025-01-25 | 64.97 | DeepSeek-R1 | 52.54 | o1-2024-12-17 | 55.42 |
| DeepSeek-R1-Zero | 64.97 | o1-2024-12-17 | 50.85 | DeepSeek-R1 | 55.42 |
| o3-mini-2025-01-31-high | 63.69 | qwen-max-2025-01-25 | 50.85 | claude-3-5-sonnet-20241022 | 53.01 |
| Doubao-1.5-pro-32k-250115 | 62.42 | MiniMax-Text-01 | 45.76 | DeepSeek-V3 | 46.99 |
| **Formal Languages** | | **Journalism and News Practice** | | **Geochemistry** | |
| o3-mini-2025-01-31-high | 69.23 | o1-2024-12-17 | 58.89 | gpt-4o-2024-05-13 | 64.00 |
| o3-mini-2025-01-31-medium | 65.38 | DeepSeek-R1-Zero | 55.56 | o1-2024-12-17 | 62.00 |
| o1-2024-12-17 | 61.54 | gpt-4o-2024-05-13 | 52.22 | gpt-4o-2024-08-06 | 62.00 |
| DeepSeek-R1 | 61.54 | claude-3-5-sonnet-20241022 | 51.11 | DeepSeek-R1 | 62.00 |
| DeepSeek-R1-Zero | 59.62 | gpt-4o-2024-11-20 | 51.11 | Llama-3.3-70B-Instruct | 62.00 |
| **Operating Systems** | | **Classical Chinese Literature** | | **Mineralogy, Petrology, and Economic Geology** | |
| DeepSeek-R1-Zero | 63.53 | DeepSeek-R1-Zero | 68.00 | o1-2024-12-17 | 57.85 |
| qwen-max-2025-01-25 | 60.00 | Doubao-1.5-pro-32k-241225 | 58.00 | DeepSeek-R1-Zero | 56.20 |
| o1-2024-12-17 | 58.82 | Doubao-1.5-pro-32k-250115 | 58.00 | DeepSeek-R1 | 54.55 |
| o3-mini-2025-01-31-high | 58.82 | DeepSeek-R1 | 56.00 | Doubao-1.5-pro-32k-241225 | 52.07 |
| o3-mini-2025-01-31-medium | 58.82 | DeepSeek-V3 | 48.00 | qwen-max-2025-01-25 | 50.41 |
| **Pattern Recognition** | | **French Language and Literature** | | **Paleontology and Stratigraphy** | |
| o1-2024-12-17 | 82.00 | gemini-2.0-flash | 64.00 | o3-mini-2025-01-31-high | 46.00 |
| claude-3-5-sonnet-20241022 | 80.00 | o1-2024-12-17 | 60.00 | o3-mini-2025-01-31-medium | 46.00 |

Continued

| Mistral-Large-Instruct-2411 | 80.00 | Llama-3.1-405B-Instruct | 60.00 | qwen-max-2025-01-25 | 46.00 |
| Doubao-1.5-pro-32k-250115 | 80.00 | gpt-4o-2024-08-06 | 58.00 | Doubao-1.5-pro-32k-250115 | 46.00 |
| Llama-3.1-405B-Instruct | 80.00 | Mistral-Large-Instruct-2411 | 58.00 | QwQ | 46.00 |

| Principles of Computer Organization | | Linguistics and Applied Linguistics | | Principles of Seismic Exploration | |
| --- | --- | --- | --- | --- | --- |
| DeepSeek-R1 | 67.07 | claude-3-5-sonnet-20241022 | 70.00 | DeepSeek-R1-Zero | 68.00 |
| DeepSeek-R1-Zero | 65.85 | gpt-4o-2024-11-20 | 64.00 | o1-2024-12-17 | 62.00 |
| Doubao-1.5-pro-32k-250115 | 63.41 | Llama-3.1-405B-Instruct | 64.00 | Doubao-1.5-pro-32k-250115 | 62.00 |
| claude-3-5-sonnet-20241022 | 60.98 | o1-2024-12-17 | 62.00 | claude-3-5-sonnet-20241022 | 60.00 |
| o3-mini-2025-01-31-medium | 60.98 | gpt-4o-2024-08-06 | 62.00 | qwen-max-2025-01-25 | 58.00 |

| Control Theory and Control Engineering | | Literary History | | Structural Geology | |
| --- | --- | --- | --- | --- | --- |
| DeepSeek-R1 | 83.15 | DeepSeek-R1-Zero | 71.01 | DeepSeek-R1 | 60.00 |
| o3-mini-2025-01-31-high | 82.02 | Doubao-1.5-pro-32k-250115 | 62.32 | o1-2024-12-17 | 54.29 |
| o3-mini-2025-01-31-medium | 77.53 | o1-2024-12-17 | 60.87 | DeepSeek-R1-Zero | 54.29 |
| DeepSeek-R1-Zero | 77.53 | DeepSeek-R1 | 60.87 | Doubao-1.5-pro-32k-241225 | 51.43 |
| o1-2024-12-17 | 76.40 | Doubao-1.5-pro-32k-241225 | 50.72 | Doubao-1.5-pro-32k-250115 | 50.00 |

| Guidance, Navigation and Control | | Literary Theory | | Solid Earth Geophysics | |
| --- | --- | --- | --- | --- | --- |
| DeepSeek-R1 | 74.51 | DeepSeek-R1 | 68.75 | o3-mini-2025-01-31-high | 82.00 |
| DeepSeek-R1-Zero | 72.55 | DeepSeek-R1-Zero | 65.62 | o1-2024-12-17 | 76.00 |
| o1-2024-12-17 | 66.67 | o1-2024-12-17 | 60.94 | qwen-max-2025-01-25 | 74.00 |
| o3-mini-2025-01-31-high | 66.67 | Doubao-1.5-pro-32k-241225 | 60.94 | o3-mini-2025-01-31-medium | 72.00 |
| o1-mini-2024-09-12 | 62.75 | claude-3-5-sonnet-20241022 | 59.38 | o3-mini-2025-01-31-low | 72.00 |

| Operations Research and Cybernetics | | Modern and Contemporary Chinese Literature | | Space physics | |
| --- | --- | --- | --- | --- | --- |
| o3-mini-2025-01-31-medium | 86.00 | DeepSeek-R1-Zero | 58.00 | o1-2024-12-17 | 56.00 |
| o3-mini-2025-01-31-high | 84.00 | Doubao-1.5-pro-32k-241225 | 54.00 | o3-mini-2025-01-31-high | 54.00 |
| DeepSeek-R1 | 82.00 | Doubao-1.5-pro-32k-250115 | 54.00 | Qwen2.5-72B-Instruct | 54.00 |
| o1-2024-12-17 | 78.00 | DeepSeek-R1 | 52.00 | gemini-2.0-flash | 50.00 |
| DeepSeek-R1-Zero | 74.00 | o1-2024-12-17 | 48.00 | DeepSeek-R1 | 50.00 |

| Electrical Theory and New Technologies | | Philology and Bibliography | | Advanced Algebra | |
| --- | --- | --- | --- | --- | --- |
| DeepSeek-R1 | 50.00 | DeepSeek-R1-Zero | 72.00 | DeepSeek-R1-Zero | 84.16 |
| o3-mini-2025-01-31-high | 45.26 | Doubao-1.5-pro-32k-250115 | 64.00 | DeepSeek-R1 | 82.67 |
| DeepSeek-R1-Zero | 45.26 | qwen-max-2025-01-25 | 60.00 | o3-mini-2025-01-31-high | 81.19 |
| o1-2024-12-17 | 44.74 | claude-3-5-sonnet-20241022 | 58.00 | o1-mini-2024-09-12 | 77.72 |
| Doubao-1.5-pro-32k-241225 | 43.68 | MiniMax-Text-01 | 58.00 | o3-mini-2025-01-31-medium | 77.72 |

| High Voltage and Insulation Technology | | Russian Language and Literature | | Combinatorial Mathematics | |
| --- | --- | --- | --- | --- | --- |
| DeepSeek-R1 | 47.73 | claude-3-5-sonnet-20241022 | 50.88 | o3-mini-2025-01-31-high | 78.01 |
| qwen-max-2025-01-25 | 46.59 | DeepSeek-R1 | 50.88 | o3-mini-2025-01-31-medium | 73.30 |
| DeepSeek-R1-Zero | 46.59 | o1-2024-12-17 | 49.12 | DeepSeek-R1 | 72.77 |
| o1-2024-12-17 | 45.45 | DeepSeek-R1-Zero | 49.12 | o1-2024-12-17 | 71.20 |
| DeepSeek-V3 | 45.45 | Llama-3.1-405B-Instruct | 47.37 | o1-mini-2024-09-12 | 64.40 |

| Power Electronics and Electrical Drives | | Composition | | Computational Mathematics | |
| --- | --- | --- | --- | --- | --- |
| DeepSeek-R1 | 58.79 | o3-mini-2025-01-31-high | 56.00 | o3-mini-2025-01-31-high | 72.00 |
| DeepSeek-R1-Zero | 56.04 | o1-2024-12-17 | 54.00 | DeepSeek-R1 | 72.00 |
| o3-mini-2025-01-31-high | 50.00 | DeepSeek-R1-Zero | 50.00 | o3-mini-2025-01-31-medium | 70.00 |
| o3-mini-2025-01-31-medium | 48.90 | DeepSeek-V3 | 48.00 | o3-mini-2025-01-31-low | 70.00 |
| o1-2024-12-17 | 47.80 | o3-mini-2025-01-31-medium | 48.00 | o1-2024-12-17 | 68.00 |

| Power Systems and Automation | | Harmony | | Cryptography | |
| --- | --- | --- | --- | --- | --- |
| DeepSeek-R1-Zero | 55.21 | DeepSeek-R1 | 51.67 | o1-2024-12-17 | 78.00 |
| DeepSeek-R1 | 54.17 | o1-2024-12-17 | 48.33 | o3-mini-2025-01-31-high | 76.00 |
| o1-2024-12-17 | 41.67 | DeepSeek-R1-Zero | 48.33 | DeepSeek-V3 | 76.00 |
| qwen-max-2025-01-25 | 40.62 | MiniMax-Text-01 | 43.33 | o3-mini-2025-01-31-medium | 76.00 |
| o3-mini-2025-01-31-high | 38.54 | o3-mini-2025-01-31-medium | 43.33 | DeepSeek-R1 | 76.00 |

| Circuits and Systems | | Instrumentation and Performance | | Discrete Mathematics | |
| --- | --- | --- | --- | --- | --- |
| DeepSeek-R1 | 67.59 | o1-2024-12-17 | 61.54 | o3-mini-2025-01-31-high | 80.90 |
| o3-mini-2025-01-31-high | 61.11 | DeepSeek-R1 | 55.38 | o3-mini-2025-01-31-medium | 76.40 |
| DeepSeek-R1-Zero | 61.11 | gemini-2.0-flash | 52.31 | o1-2024-12-17 | 74.16 |
| o1-2024-12-17 | 59.26 | DeepSeek-R1-Zero | 50.77 | DeepSeek-R1 | 73.03 |
| o3-mini-2025-01-31-medium | 57.41 | gpt-4o-2024-11-20 | 49.23 | o3-mini-2025-01-31-low | 67.42 |

| Electromagnetic Field and Microwave Technology | | Music History, Education, and Technology | | Functions of Complex Variables | |
| --- | --- | --- | --- | --- | --- |
| DeepSeek-R1 | 81.82 | o1-2024-12-17 | 62.07 | DeepSeek-R1 | 86.07 |
| o3-mini-2025-01-31-high | 78.41 | DeepSeek-R1-Zero | 55.86 | o3-mini-2025-01-31-high | 81.97 |
| DeepSeek-R1-Zero | 77.27 | gpt-4o-2024-11-20 | 53.10 | o3-mini-2025-01-31-medium | 81.97 |

| | | | | | |
|---|---|---|---|---|---|
| o1-2024-12-17 | 76.14 | gpt-4o-2024-08-06 | 51.03 | DeepSeek-R1-Zero | 80.33 |
| o3-mini-2025-01-31-medium | 71.59 | DeepSeek-R1 | 51.03 | o1-mini-2024-09-12 | 77.87 |
| **Microelectronics and Solid-State Electronics** | | **Musical Forms and Analysis** | | **Functions of Real Variables** | |
| DeepSeek-R1 | 76.00 | claude-3-5-sonnet-20241022 | 56.00 | o1-2024-12-17 | 79.10 |
| o1-2024-12-17 | 68.00 | o1-2024-12-17 | 52.00 | o3-mini-2025-01-31-high | 77.61 |
| gemini-2.0-flash | 68.00 | o3-mini-2025-01-31-high | 52.00 | DeepSeek-R1 | 77.61 |
| o3-mini-2025-01-31-high | 66.00 | DeepSeek-R1 | 52.00 | o3-mini-2025-01-31-medium | 76.12 |
| Doubao-1.5-pro-32k-250115 | 66.00 | Qwen2.5-72B-Instruct | 52.00 | o1-mini-2024-09-12 | 74.63 |
| **Environmental Engineering** | | **Pitch and Scales** | | **Fundamental Mathematics** | |
| DeepSeek-R1 | 67.42 | DeepSeek-R1-Zero | 48.21 | DeepSeek-R1 | 83.76 |
| DeepSeek-R1-Zero | 65.17 | o3-mini-2025-01-31-high | 42.86 | o3-mini-2025-01-31-high | 80.71 |
| o1-2024-12-17 | 60.67 | Llama-3.1-405B-Instruct | 42.86 | o3-mini-2025-01-31-medium | 80.71 |
| o3-mini-2025-01-31-high | 56.18 | DeepSeek-R1 | 41.07 | o1-2024-12-17 | 78.68 |
| Doubao-1.5-pro-32k-241225 | 55.06 | o1-2024-12-17 | 39.29 | DeepSeek-R1-Zero | 77.16 |
| **Environmental Science** | | **Business and Accounting Management** | | **Fuzzy Mathematics** | |
| DeepSeek-R1-Zero | 78.00 | DeepSeek-R1-Zero | 58.70 | o3-mini-2025-01-31-high | 82.00 |
| o1-2024-12-17 | 76.00 | o1-2024-12-17 | 53.26 | DeepSeek-R1 | 78.00 |
| DeepSeek-R1 | 74.00 | Doubao-1.5-pro-32k-241225 | 53.26 | DeepSeek-R1-Zero | 78.00 |
| Doubao-1.5-pro-32k-250115 | 74.00 | Doubao-1.5-pro-32k-250115 | 52.17 | o3-mini-2025-01-31-medium | 74.00 |
| qwen-max-2025-01-25 | 72.00 | DeepSeek-R1 | 50.00 | o3-mini-2025-01-31-low | 74.00 |
| **Environmental and Resource Protection** | | **Tourism Management and Technological Economics Management** | | **Geometry and Topology** | |
| claude-3-5-sonnet-20241022 | 64.00 | o1-2024-12-17 | 56.00 | o3-mini-2025-01-31-high | 75.47 |
| o1-2024-12-17 | 62.00 | Doubao-1.5-pro-32k-250115 | 56.00 | DeepSeek-R1 | 71.32 |
| gpt-4o-2024-05-13 | 62.00 | o3-mini-2025-01-31-medium | 54.00 | o1-2024-12-17 | 69.43 |
| Doubao-1.5-pro-32k-250115 | 62.00 | qwen-max-2025-01-25 | 54.00 | o3-mini-2025-01-31-medium | 68.30 |
| DeepSeek-V3 | 60.00 | o3-mini-2025-01-31-high | 52.00 | o1-mini-2024-09-12 | 63.77 |
| **Food Biochemistry** | | **Information Management Science** | | **Graph Theory** | |
| o1-2024-12-17 | 88.00 | qwen-max-2025-01-25 | 34.00 | o3-mini-2025-01-31-high | 68.63 |
| Doubao-1.5-pro-32k-241225 | 86.00 | DeepSeek-R1-Zero | 34.00 | o3-mini-2025-01-31-medium | 66.67 |
| DeepSeek-R1 | 86.00 | Qwen2.5-32B-Instruct | 32.00 | DeepSeek-R1 | 62.75 |
| Doubao-1.5-pro-32k-250115 | 86.00 | Qwen2.5-72B-Instruct | 32.00 | o1-2024-12-17 | 58.82 |
| DeepSeek-R1-Zero | 86.00 | MiniMax-Text-01 | 30.00 | DeepSeek-V3 | 54.90 |
| **Food Processing and Storage Engineering** | | **Information Management and Communication** | | **Group Theory** | |
| Doubao-1.5-pro-32k-250115 | 54.24 | o1-2024-12-17 | 92.00 | DeepSeek-R1 | 68.00 |
| DeepSeek-R1 | 52.54 | claude-3-5-sonnet-20241022 | 84.00 | o3-mini-2025-01-31-high | 60.00 |
| Doubao-1.5-pro-32k-241225 | 50.85 | DeepSeek-V3 | 84.00 | o3-mini-2025-01-31-medium | 58.00 |
| DeepSeek-R1-Zero | 50.85 | qwen-max-2025-01-25 | 84.00 | DeepSeek-R1-Zero | 58.00 |
| o1-2024-12-17 | 47.46 | DeepSeek-R1 | 84.00 | o1-2024-12-17 | 56.00 |
| **Forest Engineering** | | **Library and Archival Science** | | **Mathematical Analysis** | |
| o1-2024-12-17 | 66.00 | o3-mini-2025-01-31-high | 58.00 | o3-mini-2025-01-31-high | 87.50 |
| gpt-4o-2024-08-06 | 62.00 | qwen-max-2025-01-25 | 52.00 | DeepSeek-R1 | 86.11 |
| o3-mini-2025-01-31-medium | 62.00 | DeepSeek-R1-Zero | 52.00 | o1-2024-12-17 | 82.64 |
| qwen-max-2025-01-25 | 62.00 | o1-2024-12-17 | 48.00 | o3-mini-2025-01-31-medium | 82.29 |
| o3-mini-2025-01-31-high | 58.00 | o3-mini-2025-01-31-medium | 48.00 | DeepSeek-R1-Zero | 80.21 |
| **Wood Science and Technology** | | **Management Science and Engineering** | | **Number Theory** | |
| o1-2024-12-17 | 72.00 | DeepSeek-R1-Zero | 70.69 | o3-mini-2025-01-31-high | 90.00 |
| claude-3-5-sonnet-20241022 | 66.00 | DeepSeek-R1 | 68.97 | o3-mini-2025-01-31-medium | 88.18 |
| o3-mini-2025-01-31-high | 66.00 | o1-2024-12-17 | 62.07 | o1-2024-12-17 | 86.82 |
| o3-mini-2025-01-31-medium | 66.00 | o3-mini-2025-01-31-high | 62.07 | DeepSeek-R1 | 83.18 |
| DeepSeek-R1-Zero | 66.00 | Doubao-1.5-pro-32k-250115 | 62.07 | o3-mini-2025-01-31-low | 74.09 |
| **Geological Resources and Geological Engineering** | | **Education Economics, Management and Social Security** | | **Numerical Analysis** | |
| gemini-2.0-flash | 66.00 | DeepSeek-R1 | 66.00 | DeepSeek-R1 | 85.07 |
| DeepSeek-R1-Zero | 66.00 | DeepSeek-R1-Zero | 60.00 | o1-2024-12-17 | 74.63 |
| o1-2024-12-17 | 64.00 | o1-2024-12-17 | 56.00 | o3-mini-2025-01-31-high | 74.63 |
| claude-3-5-sonnet-20241022 | 64.00 | Doubao-1.5-pro-32k-241225 | 56.00 | o3-mini-2025-01-31-medium | 74.63 |
| DeepSeek-R1 | 62.00 | MiniMax-Text-01 | 54.00 | DeepSeek-R1-Zero | 73.13 |
| **Hydraulics and Hydrology** | | **Land Resource Management and Administrative Management** | | **Ordinary Differential Equations** | |
| DeepSeek-R1 | 66.91 | DeepSeek-R1-Zero | 66.67 | o3-mini-2025-01-31-high | 82.50 |

Continued

| Model | Score | Model | Score | Model | Score |
|---|---|---|---|---|---|
| Doubao-1.5-pro-32k-250115 | 62.50 | qwen-max-2025-01-25 | 58.82 | DeepSeek-R1 | 81.25 |
| DeepSeek-R1-Zero | 61.03 | DeepSeek-R1 | 58.82 | o3-mini-2025-01-31-medium | 80.62 |
| o1-2024-12-17 | 59.56 | o1-2024-12-17 | 56.86 | o1-2024-12-17 | 78.75 |
| o3-mini-2025-01-31-high | 58.82 | DeepSeek-V3 | 52.94 | DeepSeek-R1-Zero | 78.12 |
| **Water conservancy and Hydropower Engineering** | | **Social Medicine and Health Management** | | **Polynomials and Series Expansions** | |
| DeepSeek-R1 | 53.66 | qwen-max-2025-01-25 | 70.00 | o3-mini-2025-01-31-high | 82.12 |
| DeepSeek-R1-Zero | 50.00 | o1-2024-12-17 | 68.00 | DeepSeek-R1 | 82.12 |
| o1-mini-2024-09-12 | 47.56 | DeepSeek-R1 | 68.00 | o3-mini-2025-01-31-medium | 79.89 |
| o3-mini-2025-01-31-high | 46.34 | MiniMax-Text-01 | 64.00 | DeepSeek-R1-Zero | 79.33 |
| o1-2024-12-17 | 45.12 | qwen-max-2024-09-19 | 64.00 | o1-2024-12-17 | 76.54 |
| **Antenna and Radio Communication** | | **Forensic Medicine** | | **Probability and Statistics** | |
| DeepSeek-R1 | 73.68 | o1-2024-12-17 | 68.00 | DeepSeek-R1 | 82.14 |
| o1-2024-12-17 | 69.30 | qwen-max-2025-01-25 | 62.00 | o3-mini-2025-01-31-high | 80.36 |
| o3-mini-2025-01-31-high | 69.30 | gemini-2.0-flash | 60.00 | DeepSeek-R1-Zero | 76.79 |
| o3-mini-2025-01-31-medium | 69.30 | DeepSeek-R1 | 60.00 | o1-2024-12-17 | 76.34 |
| DeepSeek-R1-Zero | 68.42 | gpt-4o-2024-11-20 | 60.00 | o1-mini-2024-09-12 | 72.77 |
| **Communication Principles** | | **Human Anatomy and Histology-Embryology** | | **Special Number Theory** | |
| DeepSeek-R1 | 75.86 | o1-2024-12-17 | 61.90 | DeepSeek-R1 | 86.00 |
| DeepSeek-R1-Zero | 68.10 | DeepSeek-R1 | 60.32 | o1-2024-12-17 | 80.00 |
| o3-mini-2025-01-31-high | 66.38 | DeepSeek-R1-Zero | 59.79 | o3-mini-2025-01-31-high | 78.00 |
| o3-mini-2025-01-31-medium | 63.79 | Doubao-1.5-pro-32k-250115 | 59.26 | o1-mini-2024-09-12 | 68.00 |
| o1-2024-12-17 | 61.21 | qwen-max-2025-01-25 | 53.97 | o3-mini-2025-01-31-medium | 68.00 |
| **Communication and Information Systems** | | **Immunology** | | **Stochastic Processes** | |
| DeepSeek-R1 | 71.01 | Doubao-1.5-pro-32k-250115 | 56.52 | o1-2024-12-17 | 88.00 |
| o1-2024-12-17 | 66.67 | o1-2024-12-17 | 53.62 | o3-mini-2025-01-31-high | 88.00 |
| DeepSeek-R1-Zero | 66.67 | DeepSeek-R1 | 53.62 | o3-mini-2025-01-31-medium | 88.00 |
| Doubao-1.5-pro-32k-250115 | 59.42 | Llama-3.1-405B-Instruct | 52.17 | o3-mini-2025-01-31-low | 88.00 |
| o3-mini-2025-01-31-high | 56.52 | DeepSeek-R1-Zero | 52.17 | DeepSeek-R1 | 88.00 |
| **Optical Fiber Communication** | | **Pathogen Biology** | | **Hydrogeology** | |
| o3-mini-2025-01-31-medium | 84.00 | o1-2024-12-17 | 59.57 | o1-2024-12-17 | 78.00 |
| DeepSeek-R1 | 82.00 | DeepSeek-R1 | 59.57 | DeepSeek-R1 | 76.00 |
| Doubao-1.5-pro-32k-250115 | 82.00 | DeepSeek-R1-Zero | 58.51 | Doubao-1.5-pro-32k-250115 | 76.00 |
| o3-mini-2025-01-31-high | 80.00 | Doubao-1.5-pro-32k-241225 | 54.26 | o3-mini-2025-01-31-high | 74.00 |
| o1-2024-12-17 | 76.00 | qwen-max-2025-01-25 | 53.19 | Doubao-1.5-pro-32k-241225 | 72.00 |
| **Signal and Information Processing** | | **Pathology and Pathophysiology** | | **Marine Biology** | |
| DeepSeek-R1 | 81.29 | o1-2024-12-17 | 66.09 | qwen-max-2025-01-25 | 90.00 |
| o3-mini-2025-01-31-high | 78.06 | Doubao-1.5-pro-32k-250115 | 64.35 | Yi-Lighting | 88.00 |
| DeepSeek-R1-Zero | 78.06 | Doubao-1.5-pro-32k-241225 | 60.87 | gpt-4o-2024-11-20 | 88.00 |
| o1-2024-12-17 | 74.19 | DeepSeek-R1 | 55.65 | Doubao-1.5-pro-32k-250115 | 86.00 |
| o3-mini-2025-01-31-medium | 72.26 | qwen-max-2025-01-25 | 54.78 | Llama-3.1-70B-Instruct | 86.00 |
| **Instrument Science and Technology** | | **Radiation Medicine** | | **Marine Chemistry** | |
| o1-2024-12-17 | 66.00 | o1-2024-12-17 | 86.00 | Doubao-1.5-pro-32k-250115 | 72.00 |
| DeepSeek-R1 | 66.00 | Doubao-1.5-pro-32k-241225 | 84.00 | claude-3-5-sonnet-20241022 | 70.00 |
| DeepSeek-R1-Zero | 60.00 | qwen-max-2025-01-25 | 84.00 | Llama-3.1-70B-Instruct | 70.00 |
| o3-mini-2025-01-31-low | 52.00 | DeepSeek-R1-Zero | 84.00 | MiniMax-Text-01 | 68.00 |
| qwen-max-2025-01-25 | 52.00 | Doubao-1.5-pro-32k-250115 | 82.00 | Mistral-Large-Instruct-2411 | 68.00 |
| **Materials Physics and Chemistry** | | **Anesthesiology** | | **Underwater Acoustics** | |
| DeepSeek-R1 | 65.33 | DeepSeek-R1 | 74.00 | o1-2024-12-17 | 78.00 |
| o3-mini-2025-01-31-high | 57.79 | DeepSeek-R1-Zero | 74.00 | DeepSeek-R1 | 76.00 |
| DeepSeek-R1-Zero | 57.29 | o1-2024-12-17 | 72.00 | o3-mini-2025-01-31-high | 70.00 |
| o1-2024-12-17 | 56.78 | o3-mini-2025-01-31-medium | 68.00 | DeepSeek-R1-Zero | 68.00 |
| o3-mini-2025-01-31-medium | 53.27 | o3-mini-2025-01-31-low | 68.00 | o3-mini-2025-01-31-medium | 64.00 |
| **Materials Processing Engineering** | | **Clinical Laboratory Diagnostics** | | **Physical Oceanography** | |
| DeepSeek-R1-Zero | 63.33 | o1-2024-12-17 | 70.00 | o1-2024-12-17 | 70.00 |
| DeepSeek-R1 | 62.22 | o3-mini-2025-01-31-high | 66.00 | o3-mini-2025-01-31-medium | 70.00 |
| Doubao-1.5-pro-32k-250115 | 61.11 | DeepSeek-R1 | 66.00 | DeepSeek-R1 | 68.00 |
| o1-2024-12-17 | 58.89 | o3-mini-2025-01-31-medium | 64.00 | DeepSeek-R1-Zero | 66.00 |
| Doubao-1.5-pro-32k-241225 | 55.56 | o3-mini-2025-01-31-low | 62.00 | claude-3-5-sonnet-20241022 | 64.00 |
| **Manufacturing Automation** | | **Dermatology and Venereology** | | **Acoustics** | |
| DeepSeek-R1 | 72.22 | o3-mini-2025-01-31-low | 74.24 | o3-mini-2025-01-31-high | 55.92 |

| | | | | | |
|---|---|---|---|---|---|
| Doubao-1.5-pro-32k-250115 | 70.63 | o1-2024-12-17 | 71.21 | DeepSeek-R1-Zero | 55.10 |
| DeepSeek-R1-Zero | 70.63 | o3-mini-2025-01-31-medium | 68.18 | DeepSeek-R1 | 51.84 |
| Doubao-1.5-pro-32k-241225 | 69.05 | DeepSeek-R1 | 68.18 | o1-2024-12-17 | 51.43 |
| o1-2024-12-17 | 68.25 | o1-mini-2024-09-12 | 66.67 | o3-mini-2025-01-31-medium | 47.76 |
| **Mechatronic Engineering** | | **Emergency Medicine** | | **Atomic and Molecular Physics** | |
| o1-2024-12-17 | 64.00 | o3-mini-2025-01-31-medium | 60.61 | DeepSeek-R1 | 56.19 |
| DeepSeek-R1 | 62.00 | DeepSeek-R1 | 60.61 | o1-2024-12-17 | 53.33 |
| Doubao-1.5-pro-32k-250115 | 62.00 | Doubao-1.5-pro-32k-250115 | 60.61 | o3-mini-2025-01-31-high | 52.86 |
| gemini-2.0-flash | 60.00 | o1-2024-12-17 | 59.09 | DeepSeek-R1-Zero | 51.90 |
| DeepSeek-R1-Zero | 60.00 | o3-mini-2025-01-31-high | 59.09 | o3-mini-2025-01-31-medium | 51.43 |
| **Fundamentals of Dynamics and Control** | | **Geriatric Medicine** | | **Electrodynamics** | |
| o3-mini-2025-01-31-medium | 47.55 | o1-2024-12-17 | 78.00 | DeepSeek-R1 | 58.83 |
| o3-mini-2025-01-31-high | 46.69 | o3-mini-2025-01-31-high | 74.00 | o3-mini-2025-01-31-high | 57.46 |
| DeepSeek-R1 | 45.82 | DeepSeek-R1 | 72.00 | o3-mini-2025-01-31-medium | 55.40 |
| Doubao-1.5-pro-32k-250115 | 45.82 | DeepSeek-R1-Zero | 72.00 | o1-2024-12-17 | 54.72 |
| o1-2024-12-17 | 45.53 | DeepSeek-V3 | 68.00 | DeepSeek-R1-Zero | 53.17 |
| **Rigid Body Mechanics** | | **Imaging and Nuclear Medicine** | | **Fluid Physics** | |
| o1-2024-12-17 | 52.60 | DeepSeek-R1 | 50.75 | o3-mini-2025-01-31-high | 59.70 |
| o3-mini-2025-01-31-high | 50.29 | o1-2024-12-17 | 47.76 | o1-2024-12-17 | 54.48 |
| DeepSeek-R1-Zero | 46.82 | o3-mini-2025-01-31-high | 47.76 | DeepSeek-R1 | 54.48 |
| DeepSeek-R1 | 46.24 | o3-mini-2025-01-31-medium | 47.76 | Doubao-1.5-pro-32k-250115 | 53.73 |
| o3-mini-2025-01-31-medium | 42.77 | Doubao-1.5-pro-32k-250115 | 46.27 | DeepSeek-R1-Zero | 53.73 |
| **Solid Mechanics** | | **Internal Medicine** | | **Particle and Nuclear Physics** | |
| DeepSeek-R1 | 61.29 | Doubao-1.5-pro-32k-250115 | 57.92 | o3-mini-2025-01-31-medium | 62.67 |
| o1-2024-12-17 | 59.14 | o1-2024-12-17 | 55.45 | DeepSeek-R1 | 62.22 |
| o3-mini-2025-01-31-medium | 59.14 | Doubao-1.5-pro-32k-241225 | 53.47 | o1-2024-12-17 | 61.33 |
| o3-mini-2025-01-31-high | 56.99 | qwen-max-2025-01-25 | 53.47 | o3-mini-2025-01-31-high | 60.00 |
| DeepSeek-R1-Zero | 55.91 | DeepSeek-R1-Zero | 52.97 | DeepSeek-R1-Zero | 59.11 |
| **Theoretical Fluid Mechanics** | | **Neurology** | | **Polymer Physics** | |
| DeepSeek-R1 | 77.86 | o1-2024-12-17 | 59.79 | DeepSeek-R1 | 69.72 |
| o1-2024-12-17 | 72.86 | DeepSeek-R1 | 59.79 | DeepSeek-R1-Zero | 61.47 |
| o3-mini-2025-01-31-high | 72.14 | DeepSeek-R1-Zero | 57.67 | o1-2024-12-17 | 60.55 |
| Doubao-1.5-pro-32k-250115 | 72.14 | gpt-4o-2024-05-13 | 57.14 | o3-mini-2025-01-31-high | 60.55 |
| DeepSeek-R1-Zero | 72.14 | Llama-3.3-70B-Instruct | 55.56 | o3-mini-2025-01-31-medium | 59.63 |
| **Theoretical Mechanics** | | **Nursing and Rehabilitation Medicine** | | **Quantum Mechanics** | |
| o3-mini-2025-01-31-high | 58.71 | qwen-max-2025-01-25 | 64.00 | o3-mini-2025-01-31-high | 55.83 |
| DeepSeek-R1 | 56.13 | DeepSeek-R1 | 64.00 | DeepSeek-R1 | 55.83 |
| DeepSeek-R1-Zero | 56.13 | Doubao-1.5-pro-32k-250115 | 64.00 | o1-2024-12-17 | 54.85 |
| o1-2024-12-17 | 53.55 | o1-2024-12-17 | 62.00 | o3-mini-2025-01-31-medium | 54.85 |
| o3-mini-2025-01-31-medium | 53.55 | claude-3-5-sonnet-20241022 | 58.00 | Doubao-1.5-pro-32k-250115 | 54.85 |
| **Iron and Steel Metallurgy** | | **Obstetrics and Gynecology** | | **Relativity** | |
| DeepSeek-R1 | 50.48 | DeepSeek-R1-Zero | 58.82 | o3-mini-2025-01-31-high | 72.15 |
| DeepSeek-R1-Zero | 49.52 | Doubao-1.5-pro-32k-241225 | 56.86 | DeepSeek-R1 | 70.89 |
| o3-mini-2025-01-31-high | 43.81 | Doubao-1.5-pro-32k-250115 | 56.86 | o1-2024-12-17 | 69.62 |
| Doubao-1.5-pro-32k-250115 | 42.86 | o1-2024-12-17 | 54.90 | Doubao-1.5-pro-32k-250115 | 69.62 |
| o1-2024-12-17 | 40.95 | qwen-max-2025-01-25 | 54.90 | o3-mini-2025-01-31-medium | 68.35 |
| **Non-ferrous Metallurgy** | | **Oncology** | | **Semiconductor Physics** | |
| o1-2024-12-17 | 60.00 | o1-2024-12-17 | 63.79 | DeepSeek-R1 | 60.66 |
| DeepSeek-R1-Zero | 60.00 | Doubao-1.5-pro-32k-250115 | 62.07 | o3-mini-2025-01-31-high | 55.74 |
| DeepSeek-R1 | 58.00 | Doubao-1.5-pro-32k-241225 | 58.62 | o1-2024-12-17 | 54.10 |
| Doubao-1.5-pro-32k-250115 | 56.00 | DeepSeek-R1-Zero | 58.62 | DeepSeek-R1-Zero | 54.10 |
| qwen-max-2025-01-25 | 52.00 | o3-mini-2025-01-31-high | 56.90 | o3-mini-2025-01-31-medium | 47.54 |
| **Physical Chemistry of Metallurgical Process** | | **Ophthalmology** | | **Solid State Physics** | |
| DeepSeek-R1-Zero | 68.00 | o3-mini-2025-01-31-medium | 68.00 | DeepSeek-R1 | 61.90 |
| o1-2024-12-17 | 64.00 | DeepSeek-R1-Zero | 62.00 | o1-2024-12-17 | 58.50 |
| DeepSeek-R1 | 64.00 | o3-mini-2025-01-31-high | 60.00 | o3-mini-2025-01-31-high | 57.82 |
| Doubao-1.5-pro-32k-250115 | 64.00 | gpt-4o-2024-11-20 | 60.00 | DeepSeek-R1-Zero | 56.46 |
| o3-mini-2025-01-31-high | 62.00 | gpt-4o-2024-05-13 | 58.00 | o3-mini-2025-01-31-medium | 53.74 |
| **Principles of Metallurgy** | | **Otorhinolaryngology** | | **Statistical Mechanics** | |
| Doubao-1.5-pro-32k-250115 | 76.00 | o1-2024-12-17 | 70.00 | DeepSeek-R1 | 84.00 |
| DeepSeek-R1 | 72.00 | o3-mini-2025-01-31-low | 68.00 | o3-mini-2025-01-31-high | 80.00 |
| o1-2024-12-17 | 70.00 | o3-mini-2025-01-31-high | 66.00 | o3-mini-2025-01-31-medium | 78.00 |

| gpt-4o-2024-11-20 | 70.00 | o3-mini-2025-01-31-medium | 66.00 | DeepSeek-R1-Zero | 76.00 |
| o1-mini-2024-09-12 | 68.00 | DeepSeek-R1-Zero | 66.00 | o1-2024-12-17 | 74.00 |
| **Mineral Processing Engineering** | | **Pediatrics** | | **Subatomic and Atomic Physics** | |
| qwen-max-2025-01-25 | 74.00 | o1-2024-12-17 | 72.00 | o3-mini-2025-01-31-high | 72.55 |
| Doubao-1.5-pro-32k-250115 | 74.00 | qwen-max-2025-01-25 | 72.00 | o3-mini-2025-01-31-medium | 72.55 |
| o1-2024-12-17 | 66.00 | Doubao-1.5-pro-32k-250115 | 72.00 | DeepSeek-R1 | 72.55 |
| DeepSeek-R1-Zero | 66.00 | Doubao-1.5-pro-32k-241225 | 70.00 | Doubao-1.5-pro-32k-250115 | 72.55 |
| Doubao-1.5-pro-32k-241225 | 62.00 | DeepSeek-R1 | 68.00 | DeepSeek-R1-Zero | 72.55 |
| **Mining and Safety Engineering** | | **Psychiatry and Mental Health** | | **Thermodynamics** | |
| DeepSeek-R1-Zero | 70.00 | qwen-max-2025-01-25 | 76.00 | o1-2024-12-17 | 56.74 |
| DeepSeek-R1 | 68.00 | gpt-4o-2024-08-06 | 72.00 | DeepSeek-R1 | 55.35 |
| Doubao-1.5-pro-32k-241225 | 66.00 | gpt-4o-2024-05-13 | 72.00 | o3-mini-2025-01-31-high | 53.95 |
| qwen-max-2025-01-25 | 66.00 | DeepSeek-R1-Zero | 72.00 | DeepSeek-R1-Zero | 52.56 |
| Doubao-1.5-pro-32k-250115 | 64.00 | claude-3-5-sonnet-20241022 | 70.00 | o3-mini-2025-01-31-medium | 50.70 |
| **Marine Engineering** | | **Surgery** | | **Thermodynamics and Statistical Physics** | |
| qwen-max-2025-01-25 | 52.00 | qwen-max-2025-01-25 | 54.41 | DeepSeek-R1 | 54.53 |
| DeepSeek-R1-Zero | 52.00 | o1-2024-12-17 | 52.94 | o3-mini-2025-01-31-high | 53.40 |
| o1-2024-12-17 | 50.00 | DeepSeek-R1-Zero | 52.94 | o1-2024-12-17 | 52.64 |
| DeepSeek-R1 | 50.00 | Doubao-1.5-pro-32k-250115 | 51.47 | DeepSeek-R1-Zero | 52.08 |
| MiniMax-Text-01 | 48.00 | DeepSeek-R1 | 50.00 | o3-mini-2025-01-31-medium | 51.89 |
| **Ship Mechanics and Design Principles** | | **Medicinal Chemistry** | | **Systems Science** | |
| qwen-max-2025-01-25 | 56.82 | o1-2024-12-17 | 64.00 | o1-2024-12-17 | 76.00 |
| o1-2024-12-17 | 53.41 | DeepSeek-R1 | 64.00 | DeepSeek-R1 | 72.00 |
| DeepSeek-R1-Zero | 52.27 | DeepSeek-R1-Zero | 64.00 | DeepSeek-R1-Zero | 72.00 |
| Doubao-1.5-pro-32k-250115 | 51.14 | qwen-max-2025-01-25 | 60.00 | o3-mini-2025-01-31-high | 70.00 |
| DeepSeek-R1 | 48.86 | o3-mini-2025-01-31-medium | 56.00 | o3-mini-2025-01-31-medium | 70.00 |
| **Nuclear Energy and Reactor Technology** | | **Microbiology and Biochemical Pharmacy** | | **Demography and Anthropology** | |
| DeepSeek-R1 | 70.18 | o1-2024-12-17 | 60.00 | DeepSeek-R1-Zero | 72.00 |
| o3-mini-2025-01-31-high | 63.16 | Doubao-1.5-pro-32k-250115 | 60.00 | o1-2024-12-17 | 70.00 |
| DeepSeek-R1-Zero | 63.16 | o3-mini-2025-01-31-high | 56.00 | claude-3-5-sonnet-20241022 | 70.00 |
| o1-2024-12-17 | 61.40 | o3-mini-2025-01-31-medium | 54.00 | DeepSeek-R1 | 70.00 |
| o3-mini-2025-01-31-medium | 59.65 | DeepSeek-R1 | 52.00 | qwen-max-2025-01-25 | 64.00 |
| **Radiation Protection and Nuclear Technology Applications** | | **Pharmaceutical Analysis** | | **Social and Folklore Studies** | |
| o1-2024-12-17 | 66.00 | DeepSeek-R1 | 68.00 | DeepSeek-R1 | 65.59 |
| o3-mini-2025-01-31-high | 64.00 | DeepSeek-R1-Zero | 68.00 | DeepSeek-R1-Zero | 64.52 |
| DeepSeek-R1 | 64.00 | claude-3-5-sonnet-20241022 | 66.00 | o1-2024-12-17 | 61.29 |
| DeepSeek-R1-Zero | 62.00 | Doubao-1.5-pro-32k-250115 | 66.00 | claude-3-5-sonnet-20241022 | 61.29 |
| claude-3-5-sonnet-20241022 | 60.00 | o1-2024-12-17 | 64.00 | DeepSeek-V3 | 53.76 |

**Back to Section Start** | **Back to Table of Contents**

