# OpenReview forum: "SuperGPQA: Scaling LLM Evaluation across 285 Graduate Disciplines"
_NeurIPS.cc/2025/Datasets_and_Benchmarks_Track — NeurIPS 2025 Datasets and Benchmarks Track poster_

### Official Review · Reviewer_iwiC · 2025-06-01

**Rating:** 6
**Confidence:** 4

**Summary:**

SuperGPQA addresses knowledge evaluation gaps by offering a comprehensive assessment of 285 graduate-level disciplines with at least 50 questions per discipline. Developed through a human-LLM collaboration system involving crowd-sourcing annotators, experts, and state-of-the-art LLMs, the benchmark conducts an effective three-stage quality inspection process to ensure reliability. The creation process includes source screening by experts to collect credible disciplinary questions, transcription by annotators who convert raw questions into multiple-choice format with confusion options, and quality inspection that identifies and revises suspicious questions. Despite its thoroughness, SuperGPQA remains challenging for even the best LLMs like o1 and Deepseek-R1, which achieve scores of only around 60%.

**Dataset Code Accessibility:**

Yes

**Ethical Considerations:**

No, there are no or only very minor ethics concerns

**Limitations Weaknesses:**

(i) Discipline distribution bias: STEM questions account for too high a proportion, e.g., Science+Engineering+Medicine=77.2%, which may affect the assessment of humanities and social sciences capabilities. It is recommended to expand non-STEM field data in the future.
(ii) Format uniformity: Currently only multiple-choice questions are supported. It is possible to consider adding open generation questions or practical tasks such as agriculture.
(iii) Error analysis: Lack of qualitative analysis of typical error cases e.g. error patterns of some specific knowledge points of those advanced LLMs in a specific discipline.

**Strengths Contributions:**

(i) Bridging important gaps: The paper proposes the first large-scale LLM evaluation benchmark SuperGPQA covering 285 graduate disciplines, significantly expanding the coverage of existing evaluation systems such as MMLU, GPQA, with a particular focus on many real-world professionalisms without exploration such as light industry, and agriculture.
(ii) Methodological innovation: The proposed Human-LLM collaborative filtering mechanism by iterative question screening and optimization is the core innovation of the data construction process, which effectively balances the expert annotation cost and data quality and ensures the difficulty and discrimination of the questions.
(iii) Comprehensive model coverage: 51 models including 6 reasoning models and 28 dialogue models and 17 base models are evaluated, including closed source/open source models and multi-generation model comparisons such as different versions of GPT-4o.
(iv):Data disclosure: they provide a complete dataset 26,529 questions and code, which provides sufficient resources for the whole LLM evaluation community.

---

> ### Author Rebuttal · Authors · 2025-07-27
>
> Dear Reviewer iwiC,
>
> We are extremely grateful for your exceptionally positive and encouraging review. We sincerely thank you for the 'Strong Accept' rating and for your detailed summary that perfectly captures the core contributions of our work. We are particularly heartened that you recognized SuperGPQA's methodological innovation, its role in bridging important evaluation gaps, and the comprehensiveness of our model testing.
>
> Thank you for also providing constructive suggestions for improvement. We agree with the points you raised and plan to address them in our final manuscript.
>
> * **On Discipline Distribution Bias and Format Uniformity:** We acknowledge these as current limitations of our benchmark. As you rightly suggest, expanding the dataset with more non-STEM questions and incorporating formats beyond multiple-choice, such as open-ended or practical tasks, are high-priority directions for our future work. We will state this clearly in our limitations section.
>
> * **On the Lack of Qualitative Error Analysis:** This is an excellent and highly valuable suggestion. We agree that a qualitative analysis of error patterns would provide the community with deeper insights into the specific reasoning failures of current SOTA models. Due to space constraints in the initial submission, we were unable to include this. For the camera-ready version, we will add a new subsection to the appendix presenting an analysis of several typical error cases. We will showcase examples where top-performing models fail, highlighting specific error patterns (e.g., in multi-step reasoning) on challenging knowledge points within specific disciplines.
>
> Once again, we wish to express our profound thanks for your strong support and insightful feedback. Your suggestions will undoubtedly strengthen our paper, and we are grateful for your time and expertise.

---

> > ### Comment · Reviewer_iwiC · 2025-08-04
> >
> > Thank you for your reply. Please incorporate these revisions into your final version.

---

> > > ### Author Response · Authors · 2025-08-05
> > >
> > > Thank you for your feedback and approval, and for the thoughtful comments.We will carefully incorporate all the suggested revisions into the final version of our paper as promised. We are grateful for your support and guidance throughout the review process.

---

### Official Review · Reviewer_Xsvh · 2025-06-28

**Rating:** 4
**Confidence:** 4

**Summary:**

This paper presents SuperGPQA, a comprehensive benchmark that evaluates graduate-level knowledge and reasoning capabilities across 285 disciplines, addressing the gap in LLM evaluation for specialized fields beyond mainstream subjects. The benchmark employs a novel Human-LLM collaborative filtering mechanism to ensure question quality through iterative refinement based on both LLM responses and expert feedback. With over 80 expert annotators involved, the benchmark reveals that even SOTA models like DeepSeek-R1 achieve only 61.82% accuracy, highlighting significant room for improvement in diverse knowledge domains and the gap between current capabilities and artificial general intelligence.

**Additional Feedback:**

All in all, this work makes some useful contributions, and I will consider raising my score if my concerns are addressed.

**Dataset Code Accessibility:**

Yes

**Dataset Code Comments:**

I have reviewed some of the dataset online; its contents appear to be well-organized and could be useful.

**Ethical Considerations:**

No, there are no or only very minor ethics concerns

**Final Justification:**

The authors presented a strong rebuttal, so I have increased my score.

**Limitations Weaknesses:**

WEAKNESS A: Human-LLM collaborative filtering, in the form described in this work, is not a novel contribution of this paper; novelty is not a prerequisite for acceptance to NeurIPS but accuracy of claims is. Prior and related works include (https://aclanthology.org/2023.emnlp-main.92/, https://arxiv.org/abs/2402.18050v1, https://arxiv.org/abs/2212.10560, etc.) Therefore, I would request that the authors scope the claims in the abstract related to this contribution more realistically, and cite and explain more related work on this topic to better contextualize their own choices.

WEAKNESS B: The authors' choice to refer to this benchmark as SuperGPQA is highly misleading. The "GP" in GPQA (https://arxiv.org/abs/2311.12022) stands for Google-proof, a term which is assigned a crystal-clear definition in the abstract of that paper -- "highly skilled non-expert validators only reach 34% accuracy, despite spending on average over 30 minutes with unrestricted access to the web (i.e., the questions are "Google-proof")".

However, the ~26,000 questions in SuperGPQA are not Google-proof. Unlike GPQA, which reports 34% accuracy for 'skilled non-experts', SuperGPQA provides no human accuracy metrics. By contrast, the annotators of SuperGPQA are 'experts', as noted by the authors themselves; "we consider experts as individuals having or pursuing a PhD, as in GPQA". And rather than attempting to answer the questions, the annotators spend "over 30 minutes" with unrestricted web access on suspicious candidate questions during the quality inspection phase. Suspicious candidates are marked by LLM-based filtering. The quality control process aims to eliminate "trivial or ambiguous questions". While there is nothing inherently wrong with using human annotators in this way, there is something wrong with referring to the resulting artifact in a way that will lead people to believe it is "Google-proof" and consistent with prior work.

WEAKNESS C:

According to the authors, questions where LLMs choose the same incorrect option are highly suspicious, because cases where multiple or all SOTA LLMs make the same error often indicate that the LLMs have memorized explanations from incorrect exercise websites.

The human experts, then, are cleaning up questions that LLMs get wrong for "unfair" reasons. This could have two significant and undesirable effects; it could bias the benchmark unintentionally by overfitting to questions a particular set of LLMs got right or wrong in a patterned way, and it could filter out legitimately hard questions that the LLMs are failing to answer. Presumably, the human annotators are coached in an attempt to avoid this latter problem; nevertheless, there could be flawed but easy questions that the LLMs are all managing to guess correctly based on heuristics or data leakage, for example, that this process misses. The chosen emphasis in filtration, and therefore the false positives in human annotation, will be skewed towards removing legitimate hard questions.

WEAKNESS D: SuperGPQA exhibits very high alignment (95.6) with GPQA, according to the authors; however, it is much larger, which makes it slower and more expensive to run, albeit somewhat more challenging to overfit, perhaps. But with the data scales modern foundation models are trained on, overfitting remains a very real risk here, and the redundancy of the signal makes it of questionable use. The long-tail disciplines emphasized by the authors aren't providing enough unique signal to tease apart model performance, at least in the aggregate. Measuring LLM performance in understudied disciplines is important, but those disciplines require their own, specialized benchmarks, not another large aggregate one.

**Strengths Contributions:**

I have reviewed some of the dataset online; its contents appear to be well-organized and could be useful.

The authors have secured the services of expert annotators, spending over 30 minutes per question with unrestricted web access during review, and have curated a new, and reasonably large, benchmark. These are useful contributions to the literature.

---

> ### Author Rebuttal · Authors · 2025-07-27
>
> Dear Reviewer Xsvh,
>
> We thank you for your time and for acknowledging the useful contributions of our work, including the well-organized dataset and our rigorous expert annotation process. We have carefully considered your concerns and would like to address them, as we believe some points may stem from a misunderstanding of our methodology and contributions. We are confident that our work is sound and stands as a significant contribution, a view shared by the other reviewers.
>
> **A: On the Novelty of Human--LLM Collaborative Filtering:**
> We thank you for the references. We agree that the general concept of Human-LLM collaboration is established in the literature. Our paper does not claim to have invented this paradigm. **Our core contribution is the design and large-scale operationalization of a *specific*, multi-stage, iterative pipeline tailored to the unique challenge of creating a graduate-level benchmark across 285 disciplines.** The novelty lies in this specific methodology and the valuable lessons learned from applying it at an unprecedented scale, which we share with the community. We will revise our abstract and introduction to more precisely state that our novelty lies in this specific application and scale, and we will add the suggested citations to better contextualize our work.
>
> **B: On the "SuperGPQA" Name:**
> We appreciate your attention to detail. Our choice of the name "SuperGPQA" was to honor GPQA as our spiritual predecessor while clearly signaling our work's unique contributions.
> * **"Super" denotes Scale:** This prefix highlights the massive expansion from GPQA's 3 disciplines and 448 questions to our 285 disciplines and over 26,000 questions.
> * **"GPQA" signifies Lineage and Quality:** This part of the name indicates that we build upon their work. We adopted GPQA's definition of "experts" for our annotators, and our benchmark's results show a **95.6% correlation** with GPQA, confirming that we are evaluating a similar axis of high-level reasoning.
>
> **C: On Potential Filtration Bias:**
> This point misinterprets our quality control process. LLMs were **not** the final arbiters of question quality. They were used as a signal to flag questions where models showed suspicious agreement on an incorrect answer—a strong indicator of data contamination from erroneous online sources. **Every flagged question was then subjected to a rigorous review by a human domain expert**, who would solve the problem from first principles to determine if the question was genuinely hard and valid, or flawed. This crucial human-expert-in-the-loop stage was specifically designed to **preserve legitimately hard questions** while filtering out flawed ones. The challenging nature of the final benchmark, with no model approaching saturation, is strong evidence that our process succeeded. We will revise Section 2 to state this protocol more explicitly.
>
> **D: On Redundancy due to High Alignment with GPQA:**
> We respectfully but strongly disagree that high correlation implies redundancy. **This high correlation is a core validation of our benchmark's quality**, demonstrating that SuperGPQA successfully and reliably measures the same expert-level reasoning capabilities as GPQA. The immense value of SuperGPQA comes from its scale and granularity, which GPQA cannot offer:
> * **Robustness:** An evaluation over ~26,000 questions is far more statistically robust and less susceptible to chance than one over 448 questions.
> * **Granularity:** SuperGPQA's primary contribution is enabling performance analysis at the level of **285 individual disciplines**. This allows for an unprecedented, fine-grained understanding of model strengths and weaknesses. Our own results in Table 7 show that long-tail disciplines like Law and History are highly discriminative and provide unique signals, directly countering the claim that they are not useful in aggregate.
>
> SuperGPQA is not a redundant benchmark; it is a more robust, reliable, and vastly more informative successor. We will strengthen this argument in our discussion section.
>
> We trust that these clarifications address your concerns, and we hope you will reconsider your evaluation in light of this information and the positive assessments from other reviewers.

---

> > ### Comment · Reviewer_Xsvh · 2025-08-03
> >
> > I thank the authors for their useful benchmark, detailed response, and for their proposed revisions. As I stated in my original review, I am open to raising my score. In addition to the changes they have already proposed, I would politely request that the authors confirm that they will add some of the clarifying points on the "SuperGPQA" name in their post to their camera-ready draft.

---

> > > ### Author Response · Authors · 2025-08-04
> > >
> > > Thanks for your insightful suggestions and quick reply. We will carefully add the aforementioned rebuttal contents in our new camera-ready version.
> > >
> > > Notably, first, as we have mentioned above, the name "SuperGPQA" is inspired by GPQA to denote the quality, difficulty, and diversity of our dataset.  Second, the high correlation with GPQA also shows that SuperGPQA successfully measures the same expert-level reasoning capabilities as GPQA.

---

### Official Review · Reviewer_TAX8 · 2025-07-03

**Rating:** 4
**Confidence:** 3

**Summary:**

The paper proposed SuperGPQA, a comprehensive benchmark that evaluates LLMs’ graduate-level knowledge and reasoning across 285 specialized disciplines using a Human-LLM collaborative filtering approach. Experimental results show that current LLMs still have significant limitations in many domains, underscoring the gap between model performance and artificial general intelligence.

**Dataset Code Accessibility:**

NA; not applicable to this submission (e.g., no new dataset, benchmark, code, or data provided)

**Ethical Considerations:**

No, there are no or only very minor ethics concerns

**Limitations Weaknesses:**

1. Multiple-choice–only design may miss reasoning depth since all problems are rewritten into single-answer MCQs, and non-MC tasks are forcibly converted to that format. This may not comprehensively evaluate the model's capability, and adding open-ended tasks would capture richer skills.
2. The authors acknowledge that many LLMs “show consistency” with incorrect solutions from online exercise sites and warn of “significant risk of data leakage” in the collected items. Would be nice to quantify the overlap between eval data and the web/llm training corpus, and potentially provide a filtered subset.
3. Authos heavily rely on LLMs during data creation, e.g., they use models generate distractors and tag suspicious questions during quality inspection. Many benchmarked systems share pre-training sources with these generator LLM, their latent biases and phrasing could give an unintended advantage.
4. Section 2 details expert and crowd pipelines but provides no statistics such as Cohen’s κ or re-annotation disagreement rates to show label reliability.
5. Evaluation is compute-intensive, 26529 questions and each with 24 prompt templates can result in 636k forward pass for a single model. Providing a small scale evaluation set for more efficient eval would be helpful.

**Strengths Contributions:**

1. Extensive disciplinary coverage and scale
2. the paper is overall well-written and well-structured

---

> ### Author Rebuttal · Authors · 2025-07-27
>
> Dear Reviewer TAX8,
>
> We sincerely thank you for your detailed and constructive review. We are grateful for your positive feedback on our paper's extensive disciplinary coverage, scale, and overall presentation. Your insightful comments have highlighted several important areas for improvement, which we will address in the camera-ready version.
>
> Below, we respond to each of the weaknesses you identified:
>
> 1.  **On the Multiple-Choice–Only Design:** Thank you for this insightful comment. We agree that the single-answer MCQ format has limitations. We chose this format primarily to enable objective, scalable, and—most importantly—**accurate answer extraction and evaluation**. This approach avoids the ambiguity and high cost associated with parsing and grading open-ended responses, ensuring evaluation consistency. However, we designed the questions to retain reasoning depth; many problems, especially in the 'Hard' category, are converted from complex, multi-step tasks. Solving them still requires a deep chain of thought. We will emphasize this rationale in our methodology section and see the development of an open-ended version as a valuable direction for future work.
>
> 2.  **On Data Leakage and Contamination:** This is a critical point, and we appreciate you highlighting it. Our paper mentions the risk of data leakage to underscore the importance of our mitigation strategy. The three-stage Human-LLM quality inspection pipeline you noted was specifically designed to serve as a rigorous de-contamination process. As detailed on page 4, questions where multiple state-of-the-art LLMs consistently chose the same incorrect answer were flagged as highly suspicious of having been memorized from erroneous online sources. These questions were then sent to human experts for re-evaluation or removal. Therefore, the final SuperGPQA dataset *is* the filtered subset you suggest. We will revise Section 2 in the final version to more explicitly frame this pipeline as our de-contamination strategy and clarify that the final benchmark consists of questions that passed this rigorous filtering.
>
> 3.  **On LLM Usage and Potential Bias in Data Creation:** This is a valid concern. To mitigate potential bias, **the intervention and final judgment of human experts were central to our data creation pipeline**. LLMs were strictly used as assistive tools to propose distractors or flag potentially problematic questions. Every LLM-generated suggestion was then subject to a rigorous review, editing, and final approval process by our expert annotators. This human-in-the-loop system ensures that the quality and integrity of each question are verified by human intelligence, not just automated processes. We will add a paragraph to Section 2 to emphasize these critical safeguards.
>
> 4.  **On the Lack of Inter-Annotator Agreement (IAA) Statistics:** Thank you for this suggestion. We agree that quantifying reliability is important. A single IAA metric like Cohen's κ is not directly applicable to our entire multi-stage, iterative refinement pipeline (which involved sourcing, transcription, and expert review, not just simple labeling). However, to better quantify the rigor of our process, we will provide additional statistics from our quality inspection stage in the camera-ready version. For instance, we will report the rate at which questions flagged by our pipeline were subsequently corrected or discarded by human experts, providing a quantitative measure of our quality control's effectiveness.
>
> 5.  **On the Compute-Intensive Evaluation:** Thank you for this practical suggestion. We would first like to clarify that the 24 prompt variations were used only for our robustness analysis (Figure 4b), while our main leaderboard results rely on a single, standard prompt. However, to improve accessibility, following the spirit of GPQA Diamond, we have curated a more lightweight and challenging diamond subset called SuperGPQA-light. This subset is constructed from two sources:
> (1) A filtered selection from the original SuperGPQA, where candidate questions are identified based on their difficulties. Specifically, those answered correctly by fewer than two of the following models: Deepseek-R1-0528, o3-mini, Seed-Thinking-1.5, Gemini-2.5-Pro, Grok-3-beta, and Claude-4. These candidates then undergo rigorous expert review to ensure factual accuracy and clarity, with any ambiguous or potentially debatable items removed.
> (2) A set of newly collected questions targeting underrepresented yet highly challenging domains—such as the humanities and long-tail light industry topics—to enhance the benchmark’s difficulty and representativeness. These questions undergo the same model-based filtering and expert annotation pipeline to ensure rigorous quality control.
> Our goal is to create a compact yet challenging benchmark that enables efficient and meaningful model comparison. The SuperGPQA-light subset, along with detailed documentation and usage guidelines, will be released soon on our project homepage. We will also include a note about its availability in the paper to support broader and more inclusive evaluation practices.
> Note that the full SuperGPQA-light construction is still in progress, and this dataset is to cover 13 disciplines, with 5 questions per subfield, ensuring broad disciplinary representation. Moreover, we report results on the existing high-quality subset to offer early insights into model performance across diverse expert domains.
> Below are the zero-shot accuracy results (%):
> | Model | Overall | Agronomy | Economics | Education | Engineering | History | Law | Lit. & Arts | Management | Medicine | Mil. Sci. | Philosophy | Science | Sociology |
> |-------|---------|----------|-----------|-----------|-------------|---------|-----|-------------|------------|----------|-----------|------------|---------|-----------|
> | Gemini-2.5-Pro-Exp-0325 | 43.81 | 22.22 | 60.00 | 23.53 | 46.29 | 100.00 | 36.36 | 46.43 | 35.29 | 33.33 | 50.00 | 50.00 | 48.76 | 66.67 |
> | Seed-Thinking-v1.5 | 41.91 | 27.78 | 27.27 | 35.29 | 44.83 | 0.00 | 45.45 | 25.00 | 29.41 | 44.44 | 50.00 | 0.00 | 47.50 | 33.33 |
> | Qwen3-235B-A22B-thinking | 41.67 | 33.33 | 36.36 | 29.41 | 47.56 | 100.00 | 45.45 | 25.00 | 23.53 | 42.22 | 25.00 | 0.00 | 44.14 | 33.33 |
> | o1-2024-12-17 | 40.62 | 22.22 | 45.45 | 35.29 | 40.57 | 100.00 | 27.27 | 35.71 | 47.06 | 31.11 | 50.00 | 0.00 | 47.93 | 66.67 |
> | claude-3-7-sonnet-20250219-thinking | 37.78 | 16.67 | 36.36 | 35.29 | 42.44 | 0.00 | 36.36 | 21.43 | 29.41 | 26.67 | 25.00 | 0.00 | 44.63 | 66.67 |
> | DeepSeek-R1 | 37.53 | 22.22 | 27.27 | 29.41 | 41.71 | 0.00 | 36.36 | 28.57 | 17.65 | 31.11 | 25.00 | 50.00 | 42.98 | 66.67 |
> | DeepSeek-R1-Zero | 36.00 | 11.11 | 27.27 | 17.65 | 35.63 | 0.00 | 27.27 | 28.57 | 35.29 | 44.44 | 50.00 | 0.00 | 42.86 | 66.67 |
> | o3-mini-2025-01-31-high | 35.56 | 11.11 | 45.45 | 17.65 | 39.43 | 0.00 | 18.18 | 21.43 | 35.29 | 22.22 | 25.00 | 50.00 | 45.76 | 33.33 |
> | llama-4-maverick | 35.10 | 33.33 | 36.36 | 35.29 | 34.29 | 100.00 | 27.27 | 28.57 | 35.29 | 40.00 | 50.00 | 0.00 | 35.54 | 66.67 |
> | Doubao-1.5-pro-32k-250115 | 34.95 | 22.22 | 0.00 | 41.18 | 38.04 | 0.00 | 27.27 | 25.00 | 23.53 | 40.00 | 25.00 | 50.00 | 38.05 | 33.33 |
> | o3-mini-2025-01-31-medium | 34.00 | 5.56 | 54.55 | 17.65 | 34.86 | 0.00 | 27.27 | 14.29 | 35.29 | 31.11 | 0.00 | 50.00 | 44.63 | 33.33 |
>
>
> Thank you once again for your thorough and valuable feedback, which will significantly improve the clarity and quality of our paper.

---

> > ### Comment · Reviewer_TAX8 · 2025-08-05
> > **Response**
> >
> > Thanks for the authors' response, I will keep my positive score.

---

### Official Review · Reviewer_aMLA · 2025-07-22

**Ethics Flags:** Data privacy, copyright, and consent
**Rating:** 5
**Confidence:** 4

**Summary:**

The paper introduces SuperGPQA, a large-scale benchmark designed to evaluate the reasoning capabilities of LLMs at a graduate level. Comparing to existing benchmarks, SuperGPQA covers more problems from more disciplines with varying difficulties. The paper presents a well-documented pipeline for data collection and quality inspection. The paper also delivers an extensive evaluation of existing LLMs, highlighting its utility and difficulty.

**Additional Feedback:**

N/A

**Dataset Code Accessibility:**

Yes

**Dataset Code Comments:**

The dataset is readily accessible on Huggingface in a usable format.

**Ethical Comments:**

I encourage the authors to provide a list of the used academic materials (e.g., a list of book names or ISBN) for their data collection process.

**Ethical Considerations:**

No, there are no or only very minor ethics concerns

**Final Justification:**

I keep the positive rating of the paper.

**Limitations Weaknesses:**

- The majority of the dataset comes from STEM fields while questions from rare domains are still limited.
- The distribution of difficulty levels is also highly skewed where most of 'Hard' questions come from engineering fields.

**Strengths Contributions:**

- The paper is well-written and organized.
- The data collection pipeline is reasonable and well-documented, with comprehensive examples and technical details covered.
- SuperGPQA surpasses existing datasets in size, difficulty, and broadness significantly.

---

> ### Author Rebuttal · Authors · 2025-07-27
>
> Dear Reviewer aMLA,
>
> Thank you for your valuable time and constructive feedback on our submission. We are very encouraged by your positive evaluation and your appreciation for the contributions of the SuperGPQA benchmark.
>
> We have carefully considered the weaknesses you identified and plan to address them as follows:
>
> 1.  **On the STEM-dominant dataset composition:** We concur with your assessment. The resulting distribution reflects two factors: Firstly, we intentionally focused on STEM, as it is a critical area for evaluating the upper-bound capabilities of current LLMs. Secondly, to ensure comprehensive coverage, we made a targeted effort to collect questions across **all 285 graduate disciplines**, despite the known scarcity of high-quality material in many long-tail fields. The final STEM-heavy distribution is a product of this extensive collection strategy combined with our rigorous quality filtering, which ultimately yielded more qualifying questions from STEM sources. We will clarify this rationale in the camera-ready version and highlight the expansion of long-tail disciplines as a priority for future work.
>
> 2.  **On the skew of 'Hard' questions:** This is an excellent point and is related to the issue above. We will add a note to our analysis to clarify that the current distribution of difficulty reflects the availability of source material that met our criteria for complex, graduate-level reasoning.
>
> 3.  **On Ethical Considerations and Data Sources:** Thank you for this excellent suggestion. To improve the transparency of our data collection process, we will expand Appendix L. While an exhaustive list is impractical, we will provide a representative sample of key sources. This list will also substantiate a point made in our analysis (Sec. C.2, p. 23, **Newer versions Lead to Better Results** ): a significant portion of our source material was translated from Chinese educational textbooks. As we noted, we believe this explains the strong performance of LLMs from Chinese companies (e.g., Qwen, Doubao) whose data pipelines likely align more closely with these sources. Providing these concrete examples makes our data collection process and this finding more transparent. The sample list will be presented as follows:
>
> | English Textbooks                                                                                                          | Chinese Textbooks                                                                                                                              |
> | :------------------------------------------------------------------------------------------------------------------------- | :--------------------------------------------------------------------------------------------------------------------------------------------- |
> | *The Physics of Energy* by Robert L. Jaffe & Washington Taylor                                                                 | *Б.П.吉米多维奇数学分析习题集题解* [Solutions to B.P. Demidovich's Problems in Mathematical Analysis] by 费定晖 (Fei Dinghui) & 周学圣 (Zhou Xuesheng) |
> | *Astronomy Today* by Eric Chaisson & Steve McMillan                                                                        | *化工流体流动和传热* [Fluid Flow and Heat Transfer in Chemical Engineering] by 柴诚敬 (Chai Chengjing) & 张国亮 (Zhang Guoliang)                      |
> | *Soil and Water Conservation Engineering* by Rodney L. Huffman, et al.                                                     | *半导体物理学* [Semiconductor Physics] by 商世广 (Shang Shiguang), et al.                                                                        |
> | *Materials Science and Engineering: An Introduction* by William D. Callister, Jr. & David G. Rethwisch                      | *物理奥林匹克竞赛大题典, 电磁学卷* [Compendium of Problems for the Physics Olympiad, Electromagnetism Volume] by 仝响 (Tong Xiang)                      |
> | *Gauge Theory of Weak Interactions* by Walter Greiner & Berndt Müller                                                        | *结构力学学习指导与典型例题解析* [Structural Mechanics Study Guide and Solved Examples Analysis] by 刘蓉华 (Liu Ronghua) & 蔡婧 (Cai Jing)                                                                         |
> | *Gas Dynamics* by James E. A. John & Theo G. Keith                                                                         | *复变函数与积分变换* [Complex Functions and Integral Transforms] by 李红 (Li Hong) & 谢松法 (Xie Songfa)                                            |
> | *Molecular Driving Forces* by Ken Dill & Sarina Bromberg                                                                     | *随机过程习题解析* [Stochastic Processes: Problem Analysis and Solutions] by 陆传赉 (Lu Chuanlai)                                                                                           |
> | *Fundamentals of Ship Hydrodynamics* by Lothar Birk                                                                        | *制冷与低温原理* [Principles of Refrigeration and Cryogenics] by 陈光明 (Chen Guangming) & 陈国邦 (Chen Guobang)                                    |
> | *Introduction to Astronomy and Cosmology* by Ian Morison                                                                   | *高等工程热力学* [Advanced Engineering Thermodynamics] by 陈则韶 (Chen Zeshao）                                                                                     |
> |                                                                                                                            |
>
> Thank you once again for your insightful comments and for recognizing the strengths of our work. Your feedback is instrumental in refining our paper.

---

> > ### Comment · Reviewer_aMLA · 2025-08-04
> >
> > I appreciate authors' effort of clearing my questions. I keep the positive rating of the paper.

---

> > > ### Author Response · Authors · 2025-08-04
> > >
> > > Dear Reviewer,
> > >
> > > Thank you for your detailed and encouraging comments on our paper. We are very grateful for your positive feedback and especially for your suggestion to include the future directions in our manuscript.
> > >
> > > We will revise the manuscript to incorporate these points and believe it will significantly improve the quality of our work. Your comments are a great motivation for us.
> > >
> > > Thank you again for your time and thoughtful review.
> > >
> > > Best regards,
> > >
> > > Authors

---

### Note · Authors · 2025-08-13

Dear AC/Reviewers,



We are very grateful to the reviewers for their constructive feedback and for their engagement with our rebuttal. The positive reception of our work and the valuable suggestions have significantly improved the final version of our paper.

We want to briefly reiterate how we've addressed the key points raised:

Reviewer aMLA: We acknowledge the valuable suggestion regarding the STEM-heavy distribution and the skew of 'Hard' questions. Our paper will now clarify the rationale behind this, explaining that it reflects our rigorous filtering process and the availability of source material, and we will highlight the expansion of long-tail disciplines as a priority for future work. Per your request, we have also added a representative sample of our data sources to Appendix L for increased transparency.

Reviewer TAX8: We have clarified that our three-stage Human-LLM pipeline is, in fact, a rigorous de-contamination process, and the final dataset is the filtered, clean subset you suggested. We also directly addressed your concern about LLM-generated bias by emphasizing the central role of human experts in our pipeline. Finally, to address the compute-intensive evaluation, we have curated and will soon release SuperGPQA-light, a compact but challenging diamond subset.

Reviewer Xsvh: We appreciate your detailed scrutiny and have made several clarifications. We will revise the abstract to state that our novelty lies in the large-scale application of a Human-LLM collaborative framework. We have also clarified the "SuperGPQA" naming, which honors GPQA's legacy while signaling our work's immense scale and granularity. We also wish to reiterate that our filtration process was designed to preserve legitimately hard questions, and the challenging nature of the benchmark proves its success.

Reviewer iwiC: We are grateful for your strong support. We agree with your suggestions and will explicitly note the limitations regarding discipline distribution and the multiple-choice format as key areas for future work. Additionally, we will add a qualitative error analysis to the appendix to provide deeper insights into LLM failures, as you suggested.

We are confident that SuperGPQA is a robust, well-validated, and much-needed resource for the community, and we believe our revisions have strengthened its presentation. Thank you again for your time and expertise.

Best,


Authors

---

### Decision · Program_Chairs · 2025-09-18

**Decision:**

Accept (poster)

**Comment:**

The paper proposes a large dataset containing questions across 285 disciplines. This coverage is much larger than the existing datasets. The creation of the dataset involves human-LLM collaborations, involving over 80 experts in different disciplines spending significant amount of time. The dataset is valuable for evaluating the general abilities of LLMs across disciplines. The tests  with the current LLMs also reveals the limited ability of LLMs to answer questions of many different domains.

Strength: The dataset has a much larger coverage of different disciplines than the existing ones. The dataset has been evaluated by domain experts.

Weakness: There may be distribution bias in the dataset, as STEM disciplines are more covered than others.

Discussions: The authors provided rebuttals to the reviewer's comments. The reviewers are mostly satisfied. The weakness (distribution of domains) is acknowledged by the authors, who will make a notice in the final version.

Overall, the paper makes a good contribution by providing a larger dataset covering many disciplines. I recommend to accept it.